# Transformation of hard pollen into soft matter

Teng-Fei Fan[1,4], Soohyun Park [1,4], Qian Shi[2], Xingyu Zhang[3], Qimin Liu[3], Yoohyun Song[1], Hokyun Chin[1], Mohammed Shahrudin Bin Ibrahim[1], Natalia Mokrzecka[1], Yun Yang[1], Hua Li[3✉], Juha Song [2✉], Subra Suresh[1✉] & Nam-Joon Cho [1,2✉]

Pollen's practically-indestructible shell structure has long inspired the biomimetic design of organic materials. However, there is limited understanding of how the mechanical, chemical, and adhesion properties of pollen are biologically controlled and whether strategies can be devised to manipulate pollen beyond natural performance limits. Here, we report a facile approach to transform pollen grains into soft microgel by remodeling pollen shells. Marked alterations to the pollen substructures led to environmental stimuli responsiveness, which reveal how the interplay of substructure-specific material properties dictates microgel swelling behavior. Our investigation of pollen grains from across the plant kingdom further showed that microgel formation occurs with tested pollen species from eudicot plants. Collectively, our experimental and computational results offer fundamental insights into how tuning pollen structure can cause dramatic alterations to material properties, and inspire future investigation into understanding how the material science of pollen might influence plant reproductive success.

[1] School of Materials Science and Engineering, Nanyang Technological University, 50 Nanyang Avenue, Singapore 639798, Singapore. [2] School of Chemical and Biomedical Engineering, Nanyang Technological University, 62 Nanyang Drive, Singapore 637459, Singapore. [3] School of Mechanical and Aerospace Engineering, Nanyang Technological University, 50 Nanyang Avenue, Singapore 639798, Singapore. [4] These authors contributed equally: Teng-Fei Fan, Soohyun Park. ✉email: LIHUA@ntu.edu.sg; songjuha@ntu.edu.sg; ssuresh@ntu.edu.sg; njcho@ntu.edu.sg

Pollen is a remarkable natural material that plays a critical role in plant reproduction and transfers viable cellular material (i.e., male gametes or sperm cells) between different reproductive parts of plants[1,2]. Regarded as practically indestructible[3,4], pollen grains are highly dynamic microscale structures that possess unique material characteristics[5–7]. When pollen grains are released from anthers, they become dehydrated and individual grains fold onto themselves—a structurally intricate process termed "harmomegathy"[8,9]. In addition, pollen grains undergo architectural remodeling during pollen tube growth, which is guided by an organized sequence of enzymatically controlled reactions[10,11]. Expanding upon the biochemical mechanisms that plants use to control the chemorheological properties of pollen grains holds untapped potential for engineering pollen-inspired materials with high-performance capabilities.

Common features of pollen grains across various plant species include a microcapsule structure, function-driven shape, and ornamental architecture[12]. We selected pollen grains from sunflower plants (*Helianthus annuus*), which have spiky appendages and a tripartite structure (Fig. 1a and Supplementary Fig. 1). The outermost layer ("exine") is made up of sporopollenin, which is a strong, crosslinked biopolymer[7], while the inner layer ("intine") is composed of elastic, load-bearing cellulose/hemicellulose microfibrils and pectin[13,14] (Fig. 1b). The aperture gap in the exine layer is integral to pollen tube growth and is neighbored by pollen wall regions with distinct material properties. As part of remodeling processes, pectin methylesterase (PME) enzyme plays a key role in controlling wall elasticity by converting pectin into pectate[15], exposing carboxylic acid functional groups and imparting greater surface charge density that modulates the intine structural

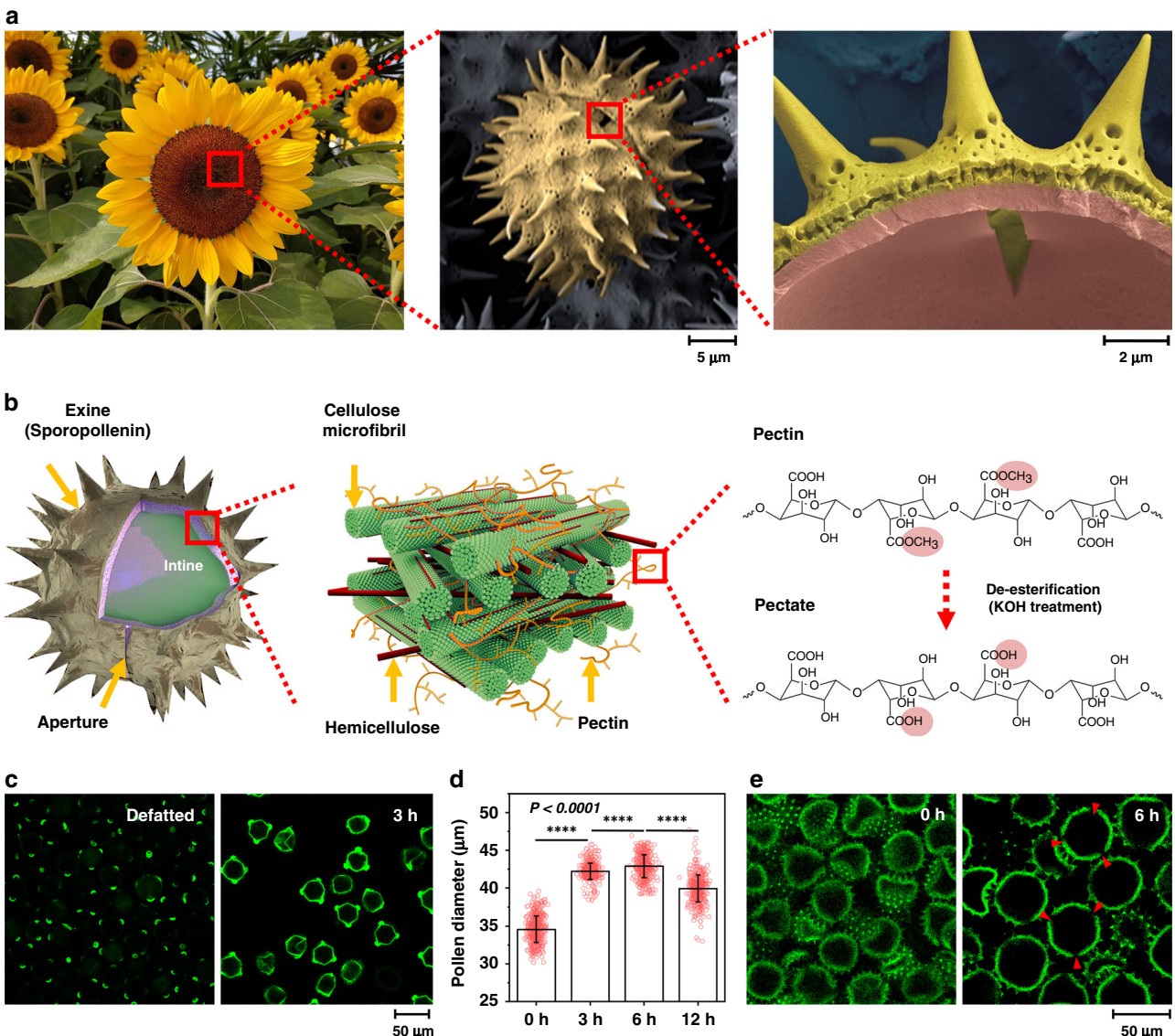

**Fig. 1 Engineering dynamic responsiveness in pollen particles. a** Steps involved in the extraction of pollen grains from sunflower plants, shown here with cross-sectional SEM images. The pollen structures are pseudo-colored. **b** Schematic illustration of stimuli-responsive pollen capsule behavior at multiple length scales. Left: Pollen structural components. Middle: Cellulose microfibrils organized by hemicellulose and pectin in the intine layer. Right: Hydrolytic conversion of pectin to pectate within the intine layer. **c** Immunofluorescence microscopy detection of de-esterified pectin within pollen shells using JIM5 as the primary antibody. **d** Size characterization of treated pollen particles as a function of KOH incubation time, as measured by DIPA. Data are analyzed by Student's *t*-test and are presented as mean ± s.d. ($n = 500$ per condition). ****$p < 0.0001$. Source data are provided as a Source Data file. **e** Cross-sectional CLSM images of pollen-derived microgel particles. Red arrows indicate aperture openings.

arrangement[15]. Biologically, this enzymatic activity is spatially controlled within the pollen wall structure[16,17], and inspires biomimetic strategies to engineer pollen structures with tunable material properties.

Motivated by these intricate biological features, we developed a nature-inspired strategy to de-esterify pectin molecules throughout the entire pollen wall structure. Our strategy involved two stages focused on liberating pollen epitopes, followed by chemical hydrolysis. In order to expose pectin epitopes, the defatted pollen was first treated with 10% (wt/vol) KOH at 80 °C for 2 h under stirring to efficiently remove cytoplasmic components. Afterwards, the pollen was incubated without stirring in fresh 10% (wt/vol) KOH at 80 °C for an extended period of up to 12 h before centrifugation was performed and this combination of processing steps facilitated microgel formation (Supplementary Fig. 2).

## Results

**Characterization of de-esterified pollen particles.** To verify de-esterification, we conducted immunofluorescence microscopy experiments using the monoclonal JIM5 antibody that recognizes weakly esterified epitopes of pectin[18] (Fig. 1c). The results confirmed successful processing as indicated by increased de-esterification after 3 h incubation while longer incubation periods resulted in more extensive de-esterification of pectin molecules[19] (Supplementary Fig. 3a). Dynamic image particle analysis (DIPA) further revealed that the treated particles swelled from approximately 35 to 43 µm in diameter after 6 h incubation, which was judged to be the optimal condition based on the greatest extent of particle swelling (Fig. 1d). These results were complemented by immunolabeling experiments with JIM7 antibody, which recognizes highly esterified epitopes of pectin[18] (Supplementary Fig. 3b). In addition, the pollen dispersions exhibited more gel-like character, as indicated by increased resistance to gravity due to de-esterification of pectin (Supplementary Fig. 3c). Coinciding with an increase in gel-like properties, confocal laser scanning microscopy (CLSM) experiments revealed that chemical processing caused the pollen particles to swell and join together while individual particles remained structurally intact[20] (Fig. 1e). Fourier transform infrared (FTIR) spectroscopy experiments also verified that all KOH-treated pollen grains appear nearly identical, irrespectively of the treatment time, whereas only the defatted sample exhibited more complex and convoluted spectral features along with peaks corresponding to pectin molecules due to residual cytoplasmic contents (Supplementary Fig. 4). Taken together, the findings reveal that incubating pollen in well-controlled alkaline conditions for extended periods of time (replicating classical soapmaking[21]) can result in the transformation of hard pollen grains into pliable, soft microgel particles.

**Chemomechanical responsiveness of pollen microgel particles.** To further evaluate how de-esterification affects the chemomechanical responsiveness of individual pollen particles, we developed a tethering strategy to immobilize pollen particles on a glass surface. While the carboxylic acid functional groups on the de-esterified pectin molecules are expected to be protonated under acidic pH conditions, the same functional groups would exhibit a higher degree of deprotonation, and hence greater anionic surface charge, under increasingly basic pH conditions (Fig. 2a). Thus, we exposed the immobilized particles to rapid changes in solution pH and observed morphological changes by time-lapse optical microscopy (Fig. 2b). When the solution pH was changed from 2 to 12, the microgel particles underwent extensive swelling and this behavior was reversible when the pH titration was conducted in the opposite direction (Supplementary Figs. 5–7 and Supplementary Movie 1). The apertures also became swollen (Supplementary Fig. 8). In marked contrast,

when the solution pH was changed from 12 to 2, the microgel particles underwent relatively faster de-swelling or compression while defatted, unprocessed pollen grains exhibited negligible pH-responsive behavior in both directions (Fig. 2c). The pH-dependent swelling response is analogous to that of ionizable polymer networks whose time scales for de-swelling occur similarly within seconds[22,23]. The observed behavior also appears to be distinct from harmomegathy because it is triggered by a different stimulus and involves more uniform morphological responses. DIPA experiments confirmed similar pH-dependent effects for pollen microgel particles in bulk solution (Supplementary Fig. 9).

We also evaluated the effect of different ions in altering the swelling/de-swelling properties of pollen microgel particles. Under neutral pH conditions, the carboxylic acid functional groups in the pectin molecules are predominately deprotonated[24] and thus sensitive to divalent cations[25] (Fig. 2d). Time-lapse optical microscopy experiments on immobilized pollen particles provided direct evidence of the dynamic response to calcium ions (Fig. 2e). Similar effects were observed with other cations as well (Fig. 2f and Supplementary Movies 2–6). In all cases, the addition of 10 mM ethylenediaminetetraacetic acid (EDTA)—a known cation chelator—caused re-swelling of the microgel particles.

We also investigated how microgel formation affects the structural and mechanical properties of the pollen substructure layers. Scanning electron microscopy (SEM) imaging confirmed the gradual removal of pollen cement from the exine layer—a process that also occurs upon the natural hydration of pollen during germination—with increasing KOH incubation time[26] (Supplementary Fig. 10). Moreover, atomic force microscopy (AFM) measurements revealed that prolonged alkaline treatment caused a significant decrease in the Young's modulus of the exine layer (Supplementary Figs. 11 and 12). In turn, the ratio of the Young's modulus values of the exine and intine layers ($M_{E/I}$) decreased from ~3 (defatted pollen) to ~1.5 (12 h KOH-treated pollen).

**Deformation mechanism of bi-layered pollen microparticles.** To further evaluate how changes in the chemomechanical properties of pollen substructure layers affect the swelling/de-swelling behavior of pollen microgel particles, we also conducted computational simulations based on a multiphysics model that incorporated finite element analysis (FEA)[27–29]. Specifically, the model simulated the extent of pollen intine swelling in different ionic solutions in line with the aforementioned experiments and accounted for the effects of chemo-electro-mechanical coupled fields on inflation/deflation of the exine layer (Supplementary Fig. 13). The pollen intine swelling pressure ($P_{i,\text{swelling}}$) was analytically calculated as a function of changes in environmental conditions (i.e., ion types and concentrations) and the computed values used as boundary conditions for analyzing inflation of the exine layer (i.e., exine inflation pressure, $P_{e,\text{inflation}}$) to evaluate the swelling/deswelling behavior of pollen microgels, and we systematically studied $M_{E/I}$ ratios from 0.15 to 8 (Supplementary Table 1). A stiffer exine ($M_{E/I} > 2$) would be expected to impose a rigid boundary condition upon the hydrated intine, thereby reducing the potential for intine swelling (Fig. 3a). On the other hand, when $M_{E/I} < 2$, both experiments and simulations demonstrated a steep increase in particle diameter ($M_{E/I} = 1.6$, Supplementary Movie 7). Moreover, the de-swelling of pollen microgel particles that was experimentally observed in the presence of multivalent cations (cf. Fig. 2) was also captured in the simulations for $M_{E/I} < 1$, and arises from the intine becoming stiffer than the exine due to the chelation reaction as shown in Fig. 2d

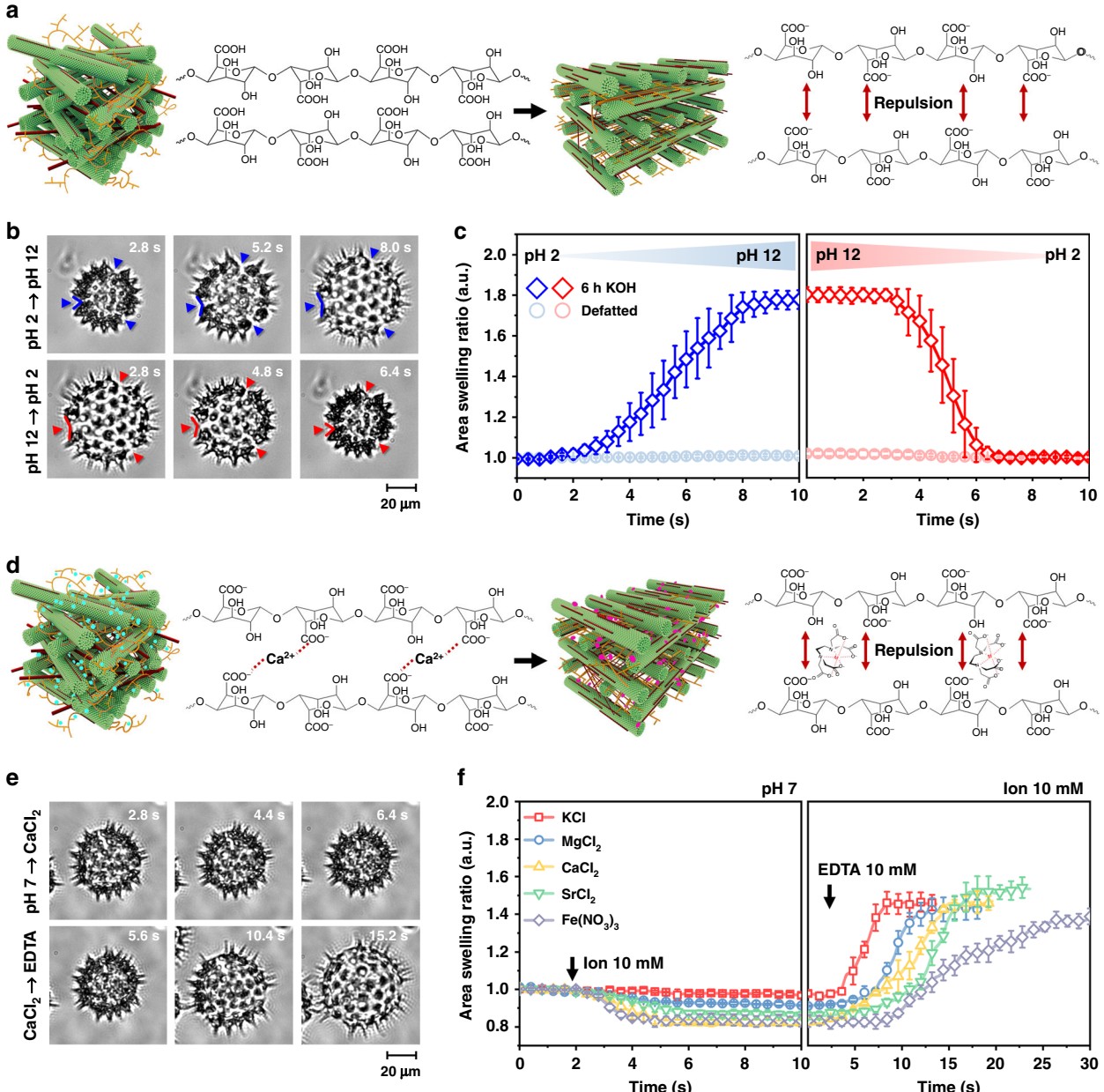

**Fig. 2 Tunable pectate interactions enable rapid, stimuli-responsive material properties. a** Schematic illustration of pH-dependent effects on pectin structure and corresponding intermolecular repulsion events. **b** Time-lapse optical micrographs of tethered pollen particles. The solution pH was changed from pH 2 to 12 (top series), and from pH 12 to 2 (bottom series). **c** Quantitative comparison of pH-induced pollen swelling and de-swelling behavior. **d** Schematic illustration of cation-induced attraction and EDTA chelating agent-induced repulsion between pectin molecules. **e** Time-lapse optical micrographs of tethered pollen particles. Calcium ions were added (top), followed by EDTA chelating agent (bottom). **f** Quantitative comparison of ion-induced pollen de-swelling and swelling behaviors. Mean ± s.d. are reported ($n = 5$ particles) in **c** and **f**, and the area swelling ratio was normalized by the initial area at pH 2 and pH 7 in **c** and **f**, respectively. Source data are provided as a Source Data file.

($M_{E/I} = 0.7$, Supplementary Movie 8). Together with the gelation of de-esterified pectin molecules within the intine layer, these findings reveal that a softened exine layer plays an important role in modulating the mechanical properties of microgel particles by allowing greater swelling of the hydrated intine. Thus, the interplay of mechanical responses in the exine and intine layers dictates the morphological behavior of the microgel particles.

Taking into account the mechanical behavior of an individual microgel particle, its representative volume element (RVE) in the FEA was defined as a structure that consisted of a hyperelastic exine layer only with one aperture and one-third symmetry (Supplementary Fig. 13). We captured the large elastic

deformation and strain energy density of a microgel particle by using FEA to simulate pollen exine inflation. Numerical simulations also revealed the morphological evolution of a microgel particle during its structural expansion (Fig. 3b, c and Supplementary Figs. 14 and 15). We defined three key swelling ratios of pollen-derived microgel particles: $\lambda_0$, $\lambda_{open}$, and $\lambda_{max}$. $\lambda_0$ is the initial swelling ratio ($\lambda_0 = 1$) when the total volume of the microgel particle is minimized. Once pectin hydration begins, rapid aperture opening occurs and intine swelling-induced pressure and resulting strain are highly localized at the tips of the three apertures. $\lambda_{open}$ is the critical swelling ratio of a pollen microgel particle, at which these chemomechanical shifts trigger

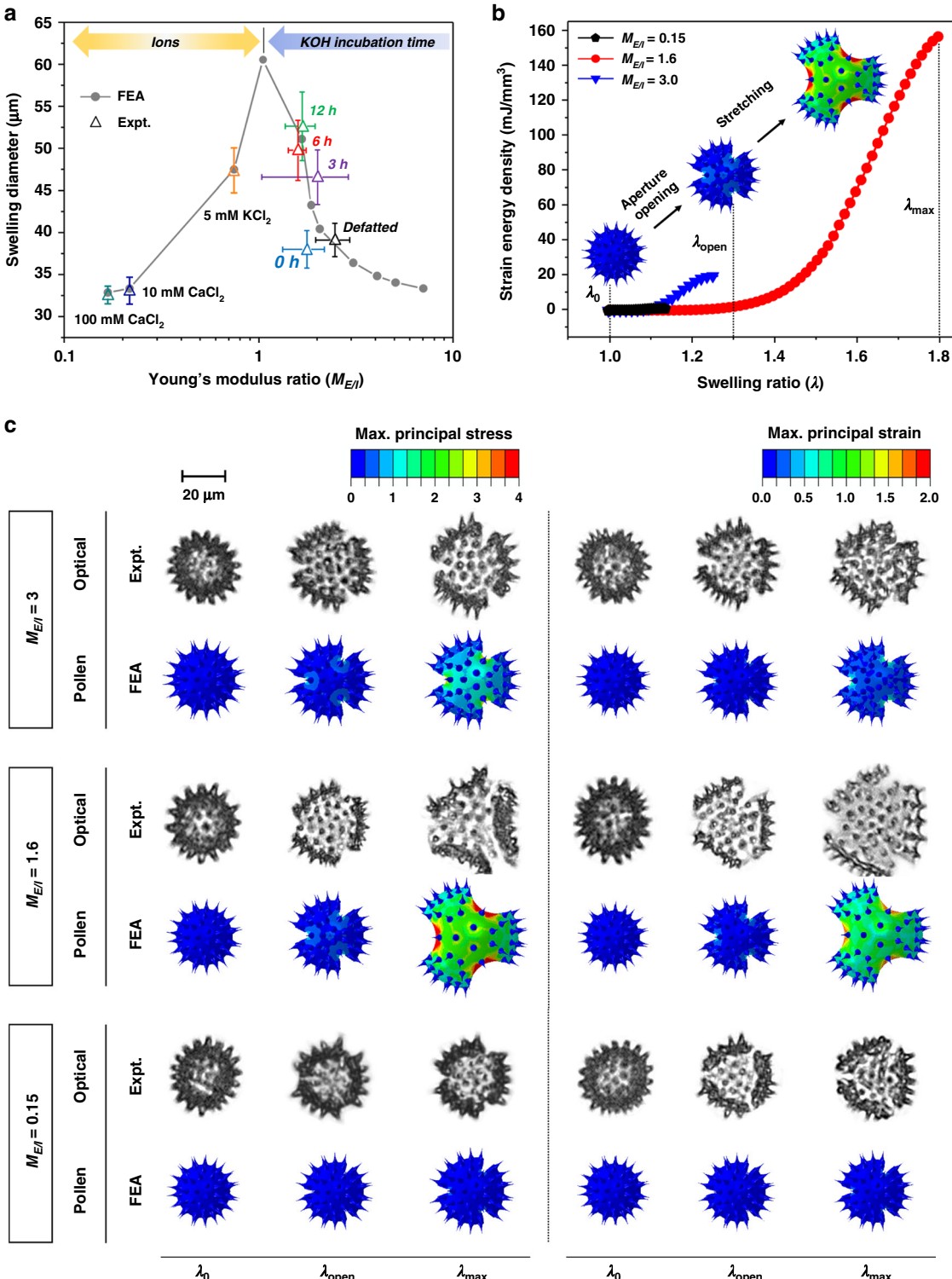

**Fig. 3 Mechanical response of pollen microgel particles. a** Swelling diameter of pollen microgel particle as a function of the ratio of the Young's modulus values of the exine to intine layers, $M_{E/I}$. **b** Predicted evolution of the strain energy density during pollen expansion as a function of the swelling ratio ($\lambda$) for three typical Young's modulus ratio values ($M_{E/I} = 0.15$, 1.6, and 3). **c** Maximum principal stress and maximum principal strain contours of the pollen microgel particles ($M_{E/I} = 0.15$, 1.6, and 3) for three critical swelling ratios ($\lambda_0$, $\lambda_{open}$, and $\lambda_{max}$) (labeled FEA), along with representative optical micrographs of pollen microgel particles in various chemical environments (i.e., ionic changes) that triggered similar morphological evolutions. $M_{E/I} = 0.15$ corresponds to pollen microgel particles immersed in 100 mM $CaCl_2$; $M_{E/I} = 1.6$ to 6 h KOH-treated pollen microgels incubated in pH 10 solution; $M_{E/I} = 3$ to defatted pollen grains. Source data are provided as a Source Data file.

opening of the three apertures. When the critical swelling ratio exceeds $\lambda_{open}$, the deformation mode of a microgel particle transfers from aperture opening to stretching. Further expansion of the microgel particle induces large deformation of both the intine and exine layers until the maximum swelling ratio, $\lambda_{max}$, is reached. The aperture opening does not require significant strain energy with highly localized deformation around the apertures, whereas subsequent stretching is an energy-consuming process that involves large-scale global deformation throughout the pollen shell. Stress and strain contours of 6 h KOH-treated pollen microgel particles immersed in pH 10 solution ($M_{E/I} = 1.6$) indicate that the three apertures are initially opened, accompanied by a small volume change until the critical swelling ratio $\lambda_{open}$ is reached, followed by a dramatic change in microgel particle diameter at the maximum swelling ratio, $\lambda_{max}$ (Supplementary Movie 7). In the case of small expansion ($M_{E/I} < 0.5$ or $M_{E/I} > 2$), the stress and strain of a microgel particle are localized around the aperture tips, indicating that either a stiffer exine or intine alone does not accommodate stretching of a particle, thereby constraining its expansion and easily recovering its dry condition with minimum energy loss.

**Microgel formation among different plant kingdom.** To extend our findings, we tested pollen grains and spores from other clades and discovered that those from flowering monocots (cattail) and gymnosperms (pine), as well as spore-bearing lycophytes (lycopodium) did not form microgel particles (Fig. 4). We also tested pollen grains from baccharis and camellia plants, which belong to the same eudicot clade as sunflower plants (Fig. 4a and Supplementary Fig. 16). We discovered that pollen grains from tested eudicot plants are transformed into microgel particles (Fig. 4b, Supplementary Figs. 17–28 and Supplementary Movie 9). Taken

together, these results support that pollen grains from the tested eudicot plants have suitable material properties to facilitate microgel conversion using our nature-inspired strategy. The observed variation in gelling propensity among pollen from different plant species may also relate to structural variations in microcapsule architecture that give rise to distinct chemomechanical responses.

Microgel formation among pollen species of eudicot plants lends strong support to the important role of chemomechanical responses in controlling pollen-related biological functions. Indeed, our strategy involving de-esterification of pectin molecules within the intine layer mimics the activity of key enzymes involved in pollen tube growth, and our findings demonstrate how subtle, chemically programmed variations in the mechanical properties of both the exine and intine layers can cause the swelling of pollen particles akin to the shape transformations that occur during the orchestrated sequence of pollen hydration, germination, and tube growth[30]. While experimental and computational studies of dehydration-induced harmomegathy have long recognized that the exine and intine layers of natural pollen grains have distinct mechanical properties[8], we discovered that this distinction alone is insufficient to explain hydration-induced pollen swelling. Strikingly, through direct experimental investigation with SEM and AFM and computational models that verified changes in the chemomechanical properties of pollen substructure layers drive the stimuli-responsive behavior of microgel particles, our findings reveal that the interplay of exine and intine layers plays a key role in dictating pollen swelling. The ratio of the Young's modulus values of the chemically tuned exine and intine layers must be within a certain, optimal range to trigger significant swelling and volume expansion of pollen-derived microgel particles in our experiments. This chemomechanical balance appears to be carefully regulated and suggests

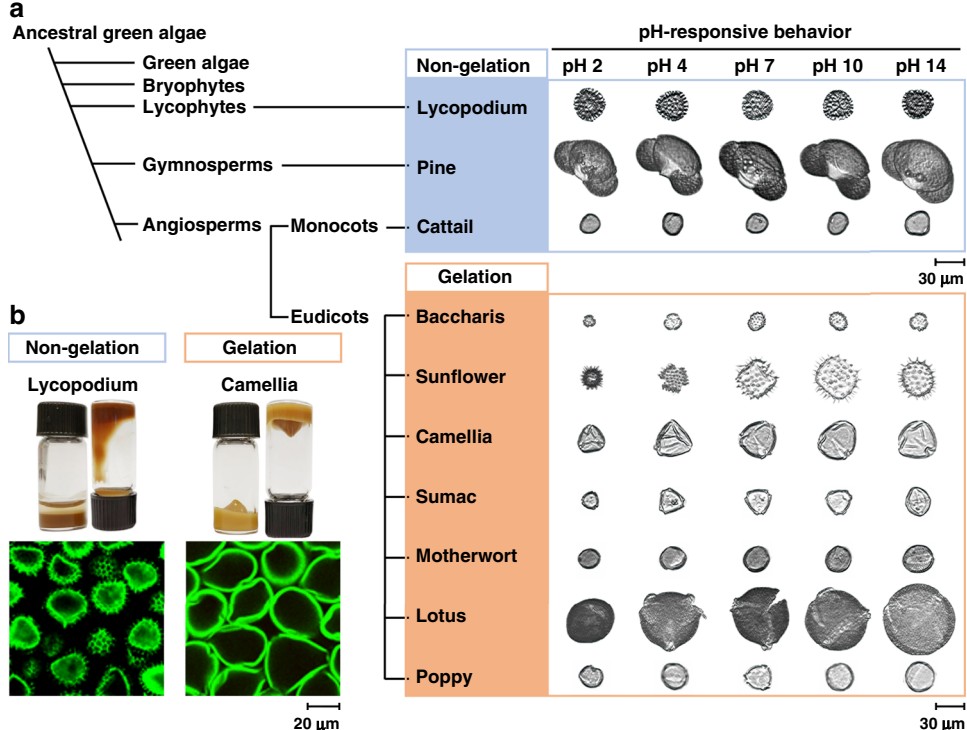

**Fig. 4 Plant kingdom sampling to identify other types of chemically tractable pollen. a** Evolutionary cladogram of plant pollen sources including lycophytes, gymnosperms and angiosperms with corresponding DIPA images as a function of pH response. **b** Top: Photographs of pollen dispersion in upright and reversed vials, showing the response of pollen gels to gravity. Bottom: Cross-sectional CLSM images of treated particles in cases where particles interact to form a microgel and where particles remain as discrete, individual entities.

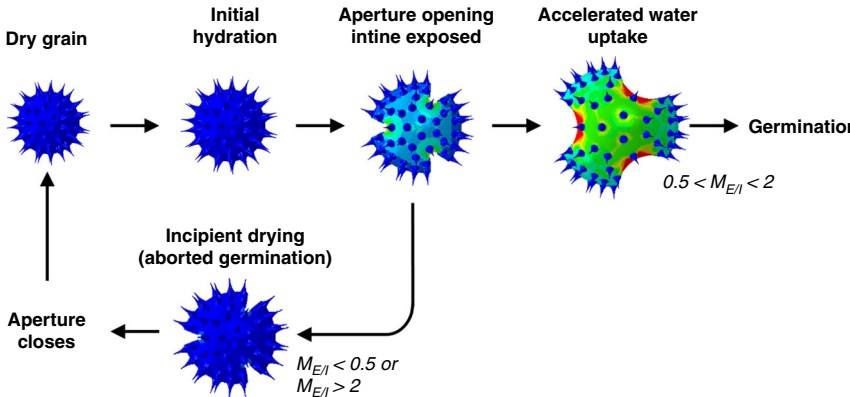

**Fig. 5 Pollen hydration/dehydration cycle of angiosperms with corresponding FEA simulation images.** The top sequence shows a dry pollen grain undergoing the water uptake process during germination. In situations where complete hydration is not possible, germination might be aborted, and the apertures close again as shown in the bottom sequence[13].

there is a chemorheological regulatory pathway at the individual particle level underpinning whether pollen hydration leads to successful germination or incipient drying[10] (Fig. 5).

## Discussion

In addition to the foregoing mechanistic implications, our results also have potential significance for pollen-inspired design of materials for a variety of possible applications. For example, hard pollen particles have been explored as natural hollow microcapsules owing to a number of factors including their abundant and renewable supply, low cost, robust shell structure, chemical stability, biocompatibility, high monodispersity, quality control, and species-specific particle size and ornamental features[31,32]. The present findings broadly extend this potential by demonstrating how a simple chemical process akin to classical soap-making can convert ultratough pollen grains into stimuli-responsive microgel particles.

Understanding and tuning the biomechanics of natural materials holds promise for materials design and application. These findings are relevant to understanding how the morphological evolution of microgel particles might play a role in ensuring plant reproduction in nature[10]. The ease of aperture opening in pollen particles induces rapid water uptake from the stigma surface. Meanwhile, the subsequent stretching and expansion of pollen grains require significant water uptake and considerable structural change with high energy consumption, to promote germination. As a result, insufficient hydration of pollen due to the inappropriate stigma conditions ceases germination, leading to harmomegathy with minimum energy loss (Fig. 5). Thus, our findings provide insight into the design principles of pollen grains in the context of harmomegathy (dehydration) and germination (hydration), while explaining how our microgel formation and tuning approach can exceed the performance limits of these natural processes. Furthermore, the process offers new pathways to produce highly uniform microgel particles from pollen sources. Such particles could be useful in a wide range of applications where excellent quality control is essential as, for example, in high-performance sensors and actuators, as well as in vivo drug delivery.

## Methods

**Pollen materials.** Defatted pollen grains from the sunflower (*Helianthus annuus* L.), pine (*Pinus taeda*), and daisy (*Baccharis halimifolia* L.) families were purchased from Greer Laboratories, Inc. (Lenoir, NC, USA). Natural lycopodium (*Lycopodium clavatum*) spores (S-type) and the cattail (*Typhae angustfolia*) pollen grains were purchased from Sigma-Aldrich (St. Louis, MO, USA), and camellia (*Camellia Sinensis* L.) bee pollen granules were purchased from Yuensun Biological

Technology Co., Ltd. (Xi'an, Shaanxi, China). Sumac (*Rhus chinensis*) bee pollen was purchased from Fengzhixiang Apiculture Co. Ltd. (Zhengzhou, Henan, China). Motherwort (*Leonurus cardiaca*) bee pollen was purchased from GTL Biotech Co., Ltd. (Xi'an, Shaanxi, China). Lotus (*Nelumbo nucifera*) bee pollen and poppy (*Papaver rhoeas*) pollen were purchased from Peffer Industrial Co., Ltd. (Zhengzhou, Henan, China).

**Microgel preparation.** Pollen microgel suspensions were prepared following a three-step process:

(1) Defatting: Natural pollen grains or bee pollen granules were defatted to remove pollenkitt. For this purpose, bee pollen granules (250 g) were refluxed in acetone (500 mL) for 3 h in a round-bottom flask under magnetic stirring (50 °C, 220 rpm). Then, acetone was decanted and deionized water (1 L, 50 °C) was added to the sample, mixed, and stirred for 1 h. The pollen suspension was passed through a nylon mesh (with 300 μm diameter pores) to remove any contaminating particulate matter. The resulting filtrate was subsequently passed through filter paper (with 6 μm diameter pores) in a vacuum filtration system. Next, the pollen sample was hydrated with deionized water (1 L, 50 °C) with magnetic stirring for 30 min and then vacuum filtered. The resulting pollen particles were refluxed in acetone (500 mL, 50 °C) with magnetic stirring (400 rpm) for 3 h. Afterward, acetone was removed by vacuum filtration and pollen samples were transferred to a glass petri dish and left to dry in a fume hood (12 h). After defatting with acetone, a dried pollen sample (20 g) was dispersed in diethyl ether (250 mL) with magnetic stirring (25 °C, 400 rpm, 2 h) in order to defat the samples. This diethyl ether defatting step was done twice and fresh diethyl ether was used in each cycle. Removal of diethyl ether was done by vacuum filtration. After washing with diethyl ether twice, the pollen sample was dispersed in fresh diethyl ether and left to stir overnight (25 °C, 400 rpm, 12 h). Diethyl ether was then removed by vacuum filtration and the pollen sample was transferred to a petri dish and air-dried in a fume hood (12 h).

(2) Cytoplasmic removal (1st KOH treatment step): Defatted pollen (2 g) was mixed with aqueous 10% (wt/vol) KOH (20 mL) in a 100 mL polytetrafluoroethylene (PTFE) round-bottom flask under magnetic stirring at 400 rpm. The suspension was refluxed for 2 h at 80 °C with stirring at 200 rpm. The suspension was transferred to a 50 mL conical centrifuge tube, and then centrifuged at 5000 × g for 5 min. The supernatant was removed and the sample was topped up to a total volume of 40 mL with fresh 10% (wt/vol) KOH. The mixture was vortexed at high speed for 2 min, followed by centrifugation at 5000 × g for 5 min. The KOH washing step was repeated a total of 5 times. Finally, the supernatant was decanted and the pollen suspension was left in the tube for the next treatment step.

(3) Microgel formation (2nd KOH treatment step): Fresh 10% (wt/vol) KOH was added to the pollen sample up to a total volume of 40 mL, followed by vortexing at high speed for 2 min. The sample was left to sit in a hot plate oven set to 80 °C for a specific period of time (3, 6, or 12 h). After that, the pollen particles were separated by centrifugation (5000 × g, 5 min). The supernatant was removed and the 50 mL conical centrifuge tube was topped up with deionized water (up to a total volume of 40 mL), followed by vortexing at high speed (2500 rpm) for 2 min. The resulting pollen suspension was centrifuged again (5000 × g, 5 min) and the supernatant was removed to measure its pH level using pH-indicator strips (MilliporeSigma, Burlington, MA, USA). These water-washing steps were repeated until the pH level reached 7.5. The resulting pollen microgel suspension was collected by the last step of centrifugation followed by the supernatant removal, then stored at 4 °C for further characterization. While

alkaline processing conditions can be used to convert hard pollen grains into soft microgel particles, not all alkaline processing protocols work and different results such as particle fragmentation (see, e.g., ref. [33]) can occur depending on processing parameters such as incubation time, static or stirred incubation conditions, washing solvents, and drying procedures.

**Dynamic image particle analysis.** Dynamic image particle analysis was performed using a benchtop Fluid Imaging FlowCAM system (Fluid Imaging Technologies, Scarborough, ME, USA) with a 200 μm flow cell and a ×10 optical lens. To optimize the focus and other measurement settings, the system was calibrated with 50 μm diameter polystyrene beads (Thermo Fisher Scientific, Waltham, MA, USA). The liquid sample was injected into the flow chamber by using a syringe pump at a speed of 0.25 mL min$^{-1}$. When a pollen particle passes through the laser light and causes significant light scatter that exceeds a certain threshold, the camera captures time-lapse images in the corresponding field of view at 15 images per second. The optical settings were: brightness 100, contrast 40, sharpness 6, and gain 400. Based on image data and analysis conducted using the Visual Spreadsheet software (Fluid Imaging Technologies), particle characteristics such as number concentration, edge gradient, and circularity were computed.

For pH-dependent studies, five different pH conditions were tested: 2, 4, 7, 10, and 14, and the solutions were prepared by the addition of 2 M HCl or 2 M KOH to deionized water. For each pH condition, 3 μL of pollen microgel sample was mixed thoroughly with 700 μL of the corresponding pH solution. After an incubation period of 5 min, the pollen particles were dispersed by pipetting and then 200 μL of the pollen suspension was passed through the flow cell.

For ion-dependent studies, aqueous solutions of KCl, MgCl$_2$, CaCl$_2$, SrCl$_2$, or Fe (NO$_3$)$_3$ in deionized water were used. The solutions were prepared at the following ion concentrations: 5, 10, 20, 50, and 100 mM. For each condition, 200 μL of pollen microgel was mixed thoroughly with 3 mL of the ion solution in a 10 mL conical centrifuge tube and incubated for 5 min. For control, deionized water was used as the dispersion medium in the absence of added ions. After an incubation period of 5 min, 200 μL of the pollen microgel suspension was passed through the flow cell. The rest of the mixture was centrifuged at 5000 × g for 5 min. The resulting supernatant was discarded and 600 μL of an aqueous solution containing an equivalent molar concentration of EDTA was added. After vortexing and an incubation period of 5 min, 200 μL of the EDTA solution was passed through the flow cell. All the experiments were conducted in triplicate and 500 particles were used for analysis.

**Fourier-transform infrared (FTIR) spectroscopy.** Before experiment, the pollen gel samples were frozen at −20 °C for 24 h and then lyophilized in a freeze dryer (Labconco, Kansas City, MO, USA) under 0.008 mbar vacuum pressure for 2 days. FTIR measurements were performed on the freeze-dried samples by using a PerkinElmer spectrometer (PerkinElmer, Waltham, MA, USA) with a diamond cell attenuated total reflection (ATR) accessory module. Reflectance infrared spectra were collected at 4000–650 cm$^{-1}$, with 16 scans per measurement and 3 replicate measurements per sample. Background spectra were collected prior to sample readings and automatically subtracted from each measurement. A baseline correction procedure was carried out using the Spectrum 10 software (PerkinElmer). Following baseline correction, each spectrum was standardized as previously reported[34].

**Scanning electron microscopy (SEM).** For the defatted samples, the pollen particles were dried in a freeze dryer (Labconco) under 0.008 mbar vacuum pressure for two days. For the microgel samples, 3 μL of the sample was dispersed in 200 μL of the appropriate medium in a 1.5 mL microcentrifuge tube and then frozen with liquid nitrogen for 2 min, followed by drying in a freeze dryer for two days. The dried samples were spread and immobilized on a sample holder with copper tape and sputter-coated with a 20-nm thick gold film using a JFC-1600 Auto Fine Coater (JEOL, Tokyo, Japan; operating settings, 20 mA for 80 s). For cross-sectional observation, the dried samples were adhered onto a piece of double-sided copper tape (2 cm × 1.3 cm) and dipped into liquid nitrogen for 5 min. Then, multiple cuts were conducted across the frozen sample with a surgical blade (B. Braun Melsungen AG, Melsungen, Germany). Finally, the pollen-adhered copper tape was dried in a freeze dryer for two days. Field-emission SEM imaging was performed using a JSM-7600F Schottky field-emission scanning electron microscope (JEOL) at an accelerating voltage of 5.00 kV under various magnification levels (between ×1500 and ×15000).

**EDC/NHS activation of pollen particles.** The carboxyl acid functional groups on the surface of pollen particles were activated by treatment with1-ethyl-3-(3-dimethylaminopropyl) carbodiimide hydrochloride (EDC, Sigma-Aldrich) and N-hydroxysuccinimide (NHS, Sigma-Aldrich). First, pollen samples (20 mg) were dispersed in 0.5 mL of 100 mM 2-(N-morpholino) ethanesulfonic acid (MES, Sigma-Aldrich) buffer (0.5 M NaCl, pH 6.0). Then, EDC (5 mg) and NHS (15 mg) were added to the pollen suspension and quickly dispersed by vortexing. After an incubation period of 30 min on a rocking platform shaker, the suspension was centrifuged (555 × g, 5 min) in order to remove the supernatant containing unreacted EDC and NHS. Then, 0.5 mL of PBS (pH 7.4) was added to re-suspend

the pollen. Immediately after resuspension, the pollen samples were either used immediately or frozen by liquid nitrogen and placed in a freeze-dryer under 0.008 mbar vacuum pressure for 4–6 h. Freeze-dried pollen samples were stored in a dry cabinet until further use.

**Immobilization of pollen particles.** Glass coverslips (25 mm × 75 mm, ibidi GmbH, Planegg, Germany) were sequentially rinsed with water and ethanol, dried with a stream of nitrogen gas, and treated with oxygen plasma for 1 min using an Expanded Plasma Cleaner (PDC-002, Harrick Plasma, Ithaca, NY, USA). After cleaning, the glass coverslips were immediately soaked in a 2% (vol/vol) solution of (3-Aminopropyl) triethoxysilane (APTES, Sigma-Aldrich) in 95% ethanol to functionalize the surface. After 30 min, the coverslips were sequentially rinsed with deionized water and 95% ethanol for a total of 3 times before drying with a stream of nitrogen gas. As-prepared APTES-treated coverslips were kept in a dry cabinet and used within 2 weeks. Then, activated pollen particles were dispersed in PBS buffer and injected onto an APTES-treated glass coverslip that was enclosed within a microfluidic flow-through chamber (sticky-Slide VI 0.4, ibidi GmbH) at a flow rate of 300 μL min$^{-1}$, as controlled by an Ismatec Reglo Digital peristaltic pump (Cole-Parmer GmbH, Wertheim, Germany). After checking that the density of attached pollen particles was within a sufficient range, the chamber was thoroughly washed by flowing PBS buffer at a flow rate of 3 mL min$^{-1}$.

**Confocal laser scanning microscopy (CLSM).** Pollen gel samples in suspension (~20 μL) were pipetted onto a thin glass slide (24 mm × 75 mm, CellPath Ltd, Newton, UK). CLSM imaging was performed using a Zeiss LSM710 microscope (Carl Zeiss, Oberkochen, Germany) equipped with three spectral reflection/fluorescence detection channels, six laser lines (405/458/488/514/561/633 nm), and connected to a Z1 inverted microscope (Carl Zeiss). A ×20 optical lens was used for imaging and at least two images were obtained per sample. Imaging was performed under the laser excitation channel at 488 nm (12.0%). Fluorescence signals from the samples were collected in photomultiplier tubes equipped with emission filters at 495–550 nm. Plane mode scanning was performed with a pixel dwell of 12.6 μs. Optimized imaging conditions were used for other types of pollen samples as follows: For immunolabeled-pollen particles, the samples were gently suspended by hand-shaking and then pipetted (~20 μL) onto a thin glass slide. Laser excitation channels were set at 488 nm (3.5%) and the emission filter was set at 493–634 nm. Plane mode scanning was performed with a pixel dwell of 1.0 μs. For the immunolabeling studies, all imaging conditions were identical in order to compare the fluorescence intensity of each sample. For tethered pollen samples, imaging was performed with two excitation channels, such as 405 nm and 488 nm, with two respective emission filters: 416–477 nm and 430–740 nm. Plane mode scanning was performed with a pixel dwell of 1.0 μs. Image processing was performed with Fiji software (available at http://fiji.sc/ [Accessed February 21, 2019]).

**Time-Lapse microscopy imaging.** Microscopy experiments were conducted using an Eclipse Ti-E inverted microscope (Nikon, Tokyo, Japan) with a CFI Super Plan Fluor ELWD ×20 or ×40 (NA = 0.45 or 0.60) objective lens (Nikon), and images were collected with an iXon3 512 pixel × 512 pixel EMCCD camera (Andor Technology, Belfast, UK). A halogen lamp, light power supply (TI-PS100W, Nikon) connected to TI-DH diascopic pillar illuminator (Nikon), was used to illuminate the pollen samples. As described above, the pollen particles were immobilized on an APTES-treated glass coverslip and the coated coverslip was then enclosed within a microfluidic flow-through chamber (25.5 mm × 75.5 mm) with a ~200 μL volume channel. Liquid samples were introduced at a flow rate of 300 μL min$^{-1}$, as controlled by a peristaltic pump (Cole-Parmer GmbH). During the swelling/de-swelling experiments, the bulk solution in the measurement chamber was exchanged and time-lapse images were captured in 0.4-s time intervals. Data processing was performed in MATLAB (MathWorks, Natick, MA, USA), ImageJ (National Institutes of Health, Bethesda, MD, USA) and OriginPro 8.5 (OriginLab, Northampton, MA, USA) software programs (Supplementary Fig. 29).

**Immunolabeling studies.** In order to detect the presence and esterification state of pectin molecules within the intine layer, JIM 5 and JIM 7 antibodies were used according to previously defined protocols[17,35]. Defatted pollen grain (10 mg) or as-prepared pollen microgel (10 μL) samples were washed with phosphate-buffered saline (PBS, 1 mL) and then incubated in 2% bovine serum albumin (BSA) in PBS (1 mL) for 30 min. The pollen particles were then incubated overnight in a shaker (Gaia Science Pte Ltd, Singapore) with a rotation speed of 200 rpm at 4 °C in the presence of a primary antibody (JIM 5 or JIM 7) (PlantProbes, Leeds, UK, Cat. No. JIM5 and Cat. No. JIM7) at 1:20 dilutions in PBS containing 0.2% BSA. After washing twice with fresh PBS, the samples were incubated with Alexa Fluor 488 AffiniPure Goat Anti-Rat IgG (H + L) secondary antibody (Jackson ImmunoResearch Laboratories, West Grove, PA, USA, Cat. No. 112-545-003), diluted 1:100 in PBS containing 0.2% BSA, for 1 h in a dark environment. Antibody-labeled pollen particles were examined immediately with a confocal laser scanning microscope (Zeiss LSM 710) and without any antifade reagents. Pollen grains without primary and/or secondary antibodies were used as controls.

**Force-distance measurements for mechanical characterization**. The mechanical properties of defatted pollen and microgel particles were characterized by conducting AFM force-distance (or load versus displacement) measurements. This depth-sensing AFM indentation approach enables the quantitative determination of the Young's modulus of the shell material[36,37]. For wet samples, pollen microgel samples (5 μL) were dispersed in the appropriate medium (50 μL), and the mixture was spread onto a petri dish (Nunc, Roskilde, Denmark) by pipette. Then, excess liquid was aspirated and the partially hydrated samples remained on the glass slide for experimental characterization. For dry specimens, the pollen sample was directly spread onto the surface of a petri dish before measurement. Intact pollen particles were chosen for extracting the Young's modulus of the exine layer and broken pollen particles, which exposed the inner layer, were chosen to determine the Young's modulus of the intine layer. In order to avoid the substrate effect, we followed the 10% depth rule where the indentation depth should be less than 1/10th of the layer thickness[38]. The average thickness of the exine layer was ~0.6 μm for defatted pollen particles and 0 h KOH-treated pollen specimens, and became thinner with increasing KOH treatment time (~0.5 μm). Thus, the indentation depths for exine measurements were set in the range of 20–60 nm under indentation loads of 3–6 μN. On the other hand, the intine layer was swollen in wet conditions, with a thickness range from 0.5 to 1.5 μm. Thus, for dry samples, the indentation depths of intine measurements were carefully limited up to ~60 nm, while for wet samples, the indentation depths were in the range of 50 to 100 nm. For all measurements, the NX-10 AFM instrument (Park Systems, Suwon, South Korea) was used with two AFM probes: (i) an aluminum reflex-coated silicon cantilever PPP-NCHR (Nanosensors, Neuchâtel, Switzerland) with a typical spring constant of ~42 N m$^{-1}$ and a tip end radius of 5 nm; and (ii) a diamond cantilever TD26135 (Micro Star Technology, Huntsville, TX) with a spring constant of ~150 N m$^{-1}$ and a tip end radius of 5 nm. The tips were shaped as polygon-based pyramid with a half-cone angle of 20°. We confirmed that both AFM probes provided almost identical values of Young's moduli for exine and intine layers regardless of the various indenting parameters such as maximum contact forces, contact time or approach speed (Supplementary Fig. 30). Thus, measurements were conducted at various positions (more than 16 data points) in a 5 μm × 5 μm area at an approach speed of 0.8 μm s$^{-1}$ with a maximum loading force of 4.8 μN and zero contact time. Before experiments, the AFM cantilever was rinsed with water and ethanol, and treated with a UV light cleaner for 30 min in order to remove organic contaminants. The spring constant and sensitivity of the deflection signal were also calibrated by recourse to the thermal vibration of the AFM cantilever[39] by employing the commercial software (XEP, Park Systems). The force-versus-displacement curves were corrected by subtracting the deflection distance of the AFM probe from the total displacement.

For data analysis, we assumed that both the exine and intine layers were isotropic. Also, we used an AFM probe with a tip end radius of 5 nm, and thus the indentation depths (20–100 nm) were significantly larger than the tip end radius. Thus, the classical Hertz model was used for data analysis[37,40]. The Hertz model[36] was fitted to the force-versus-displacement curves using a commercial software analysis program (XEI, Park Systems) and the Python script language (Supplementary Fig. 30). We treated the geometry of the AFM tip as a parabolic model whereby it has a tip radius of $R_c$, so that the force ($F$) can be expressed as:

$$F = \frac{4\sqrt{R_c}}{3}\frac{E}{1-\nu^2}\delta^{3/2},\qquad(1)$$

where $E$ is Young's modulus, $\nu$ is the Poisson's ratio, and $\delta$ is the indentation depth. The Poisson's ratio $\nu$ was set at 0.5, which is typical for natural materials.

**Numerical modeling for swelling/deswelling of pollen microgel particles**. For pollen grains, the intine layer is naturally hydrophobic due to highly esterified pectin, whereas the exine layer consists of sporopollenin that contains lipids and pollenkitt (pollen cement). Thus, the pollen shell is inherently hydrophobic and not swellable in its natural form. After the chemical treatment, the highly esterified pectin becomes de-esterified and hydrophilic while the exine layer becomes porous (due to removal of pollen cement). In addition, due to chemical treatment, the cytoplasm is also fully removed and thus the internal cavity is fully empty (Supplementary Fig. 2). Also, since the cytoplasmic contents, including cells and cellular debris, have to be released through the three apertures, the once-continuous intine layer is also ruptured. Thus, in the modeling, the two-layer pollen shell was fully discretized through the entire layer thickness at the apertures as shown in Supplementary Fig. 2. As a result, a pollen microgel particle has a hollow shell structure with three apertures, consisting of the outer exine and inner intine layers. Thus, the volumetric increase in a pollen microgel particle is caused by the interplay of the swollen intine layer and relatively stiffer exine layer. In particular, we carefully defined the swelling/deswelling behavior of the hydrogel-like intine layer due to water absorption and desorption depending on the osmotic pressure and inflation/deflation of the hyperelastic exine due to swelling-induced mechanical pressure of the underlying intine layer. Moreover, the three apertures are open, which allows rapid water intake into the internal cavity of the hollow particle that is concomitant with inflation of the pollen exine. Taken together, two key factors affect the swelling/deswelling behavior of the pollen microgel particle system: (1) intine uptakes ionic solution from the surrounding medium to generate a considerable osmotic pressure, which is the driving force for inflating the outer exine; and (2)

the stiffer exine along with the three apertures exerts an inhomogeneous constraint on intine swelling. Therefore, the swelling-induced mechanical pressure exerted at the outer surface of the intine layer ($P_{i,\text{swelling}}$) due to osmotic pressure decreases as swelling proceeds. The intine swelling pressure ($P_{i,\text{swelling}}$) equals the equilibrium pressure that is required to inflate the exine layer ($P_{e,\text{inflation}}$). The swelling ratio of the intine, $\lambda_i$, is equal to the inflation ratio of exine, $\lambda_e$ since the intine layer is tightly bound to the exine layer (Supplementary Fig. 13a). Details of the overall approach behind the multiphysics model for intine layer and finite element analysis (FEA)[27–29] for exine layer are described in Supplementary Methods.

**Statistical analysis**. Statistical analysis was performed using the Origin 2018 software package (OriginLab). Unpaired Student's $t$-test and one-way ANOVA with Tukey's multiple comparison tests were used to calculate the statistical significance of data sets. A $P$ value of less than 0.05 was considered statistically significant. Data are presented as the mean ± standard deviation, wherever appropriate.

## Data availability

All the data supporting the findings of this study are available from the corresponding authors upon reasonable request. The source data underlying Figs. 1d, 2c, f, and 3a, b, Supplementary Figs. 4, 7, 8a, b, 9b, c, 11, 12, 19–28, and 30a–e, and Supplementary Table 1 are provided as a Source Data file.

## Code availability

The computer code and algorithm that support the findings of this study are available from the corresponding author upon reasonable request.

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

## Acknowledgements

This work was supported by the National Research Foundation of Singapore through a Competitive Research Programme grant (NRF-CRP10-2012-07) and by the Ministry of Education of Singapore through a AcRF Tier 1 grant 2017-T1-001-246 (RG51/17).

## Author contributions

J.S., S.S., and N.-J.C. designed the study. T.-F.F., S.P., Y.S., H.C., M.S.I., N.M., and Y.Y. carried out experiments and collected data. Q.S., X.Z., Q.L., and H.L. contributed to numerical modeling and theoretical analysis. N.-J.C. initiated the idea. All authors contributed to data analysis and writing the paper.

## Competing interests

N.-J.C. is a co-inventor on patent no. WO2019147190A1. The remaining authors declare no competing interests.
