## [Peer Review File · Nature Communications]

Reviewers' Comments:

Reviewer #1:

Remarks to the Author:

The manuscript by Fan and coworkers reports observations on the chemical hydrolysis of sunflower pollen grains with KOH which allows to control the swelling of the grains with changes pH and ion concentration. These experiments are followed by a mechanical analysis of the exine and intine. While some results are interesting, many aspects of the manuscript are confusing and a number of claims seem to extend beyond what the authors have unequivocally established.

1) The main result of this work is the discovery that treatment of defatted pollen grains with KOH substitutes the methyl groups for hydroxyl groups in the pectins of the intine wall. I was left with several questions about this process and the data that were presented.

(a) The KOH treatment appears to replicate fairly closely the treatment reported by the same group a few years ago (Mundargi, R. C., Potroz, M. G., Park, J. H., Seo, J., Lee, J. H., & Cho, N. J. (2016). Extraction of sporopollenin exine capsules from sunflower pollen grains. *RSC Advances*, 6(20), 16533-16539). In the earlier work, sunflower pollen grains are treated with 6% KOH for 12hrs while in the current manuscript the treatment is with 10% KOH first for 2hrs and then for up to 12hrs. To my surprise, the conclusion of the 2016 paper, where a weaker solution of KOH was used, is: "The SECs isolated from alkaline lysis are completely damaged and lose their unique spiky microstructure". Scanning electron micrographs of the resulting material (Fig. 1b in the 2016 paper) show an amorphous mass of material. In contrast, the conclusion of the current manuscript is: "Taken together, the findings reveal that incubating pollen in alkaline conditions for extended periods of time (replicating classical soapmaking²²) results in the transformation of hard pollen grains into pliable, soft microgel particles". Although the authors talked about potential degradation of the pollen grains after long exposure to KOH, Fig. 1d of the manuscript shows that there are still pollen grains after 12hr since they have measured their diameter. Also, the so-called SEC shown in the current manuscript have the "spiky microstructure" of untreated pollen grains. I have trouble understanding why the results are so different and why the authors have not cited their own paper and explained the discrepancies.

(b) The experimental protocol used in this manuscript involves two successive treatments with KOH, one for 2hrs and a subsequent one for 0 to 12hrs. All other experimental conditions seem identical. Despite the similarity between the two treatments, the authors claim that their final outcomes are different. The first 2hr treatment would extract the shell (i.e. remove the pollen content and "expose pectin epitopes") while the second treatment of variable duration would substitute the methyl groups for OH groups ("extensive demethylation of exposed pectin molecules"). Supplementary Fig. 2 makes the same claim graphically. It is not clear why the authors make such a stark distinction between the net result of two identical and successive treatments with KOH. Why take such a categorical view of the experimental protocol? Is it not possible that substitution (demethylation) of pectins begins with the first KOH treatment? Is there not indication of that in Suppl. Fig 3 (0h)?

(c) It is not clear what is presented in Fig. 2c. At first, I interpreted Fig. 2c as statistics on MANY pollen grains that were treated with KOH in the same way and then exposed to rapid changes in pH. I also assumed that the area of pollen grains was normalized by the area at pH=2 which would explain why the standard deviation is zero for the Area Swelling Ratio of 1. However, I note now that the standard deviation also tends to zero at pH = 12 (Area Swelling Ratio of about 1.8). This result would certainly not be expected if MANY pollen grains are used to produce this curve and the normalization is done only with the area at pH=2. Specifically, it is extremely unlikely that many pollen grains would swell to exactly the same area swelling ratio. This curve must be explained better and ideally replaced with one that has proper statistics on a population of pollen grains.

(d) Suppl Fig. 3 shows JIM5 labelling of pollen grains treated with KOH for different time durations. Why is JIM5 labelling nearly absent at 6 and 12 hrs if the pollen grains are still able to swell after such exposure? Where are the de-esterified pectins in those treatments? Also note that the 0hr treatment shows extensive JIM5 fluorescence suggesting that the 2hr KOH pretreatment already started the de-esterification process.

(e) Finally, some controls seem necessary. For example, have the authors treated their pollen grains with PME to remove the methyl groups enzymatically? This simple control experiment would help solidify the claims made in the first part of the manuscript (Figs. 1 and 2). Also, I would have liked to see in Fig. 2c the effect of pH on defatted pollen grains that have NOT been treated with KOH. It seems the authors have done this experiment since they show pollen grains with this treatment in Fig. 3c $M_E/I = 3$. The pollen grains do show substantial swelling with pH although not as much as the KOH-treated one. Fig. 2c must include proper statistics for the pH swelling of KOH-treated and non-treated pollen grains.

2) A mechanical analysis follows the swelling analysis of KOH-treated pollen grains. At this point, the manuscript becomes more obscure and I am concerned that the authors have not captured properly the mechanical behavior of the pollen grains.

(a) I am concerned about the validity of the protocol used to infer the elastic modulus of the exine and intine. According to the authors, intact pollen grains were used to measure the modulus of the exine. In this configuration, it is not clear how the specific modulus of the exine can be disentangled from the large scale deformation of the entire pollen grain under the force applied by the AFM. On the other hand, the modulus of the intine was measured on open fragments. Presumably the intine is still bound to the exine. How can the authors claim to have measured the elastic modulus of the intine if they are probing a thin layer of material that is still mechanically connected to the exine? Moreover, the intine wall is probably anisotropic in its structure.

(b) I could not find what was the tip radius of the AFM probe, nor the final displacement of the probe. In fact, the authors do not present the force-displacement curves that are normally used to infer the elastic properties of materials. The authors only mention that "the approach speed of the AFM cantilever was fixed to 0.8 $\mu\text{m}/\text{sec}$ with a maximum loading force of 4.8 μN ." Measuring material properties of soft elastic materials with an AFM is not a simple task when working with complex structures such as the pollen grain. The authors present their results as if they were routine measurements. A better description of the AFM measurements and some controls are necessary to validate the moduli reported in the manuscript.

(c) During harmomegathy, the volume of the pollen grain increases substantially but this volume increase is achieved in many species not by stretching the exine but by unfolding or unrolling it. In other words, swelling of pollen grains during harmomegathy does not necessarily involve stretching of the exine. I believe some unfolding or unrolling of the exine is seen in the pollen grains used to illustrate the simulations in Fig. 3c, as indicated by the change in curvature of the exine segments. The case of $M_E/I = 1.6$ in Fig. 3c is particularly striking because the normally convex exine segments become flat or even concave as the pollen swells. Thus the exine shell is unrolling to accommodate the increase in pollen grain volume. The finite element simulations are presented and implemented as if the exine is simply a hard shell that must be STRETCHED to allow swelling of the pollen grain. This is not a fair assumption. In fact, close inspection of the simulations for $M_E/I = 1.6$ reveals that the simulations are not reproducing faithfully the deformation of the pollen grains. If the swelling during harmomegathy is achieved without much softening or stretching of the exine, is it not possible that KOH-treated pollen grains can also swell, at least to some extent, without stretching the exine? If this is the case, the resistance offered by the exine is not a simple stretching deformation but one that involves also bending of the exine shell.

(d) The authors should define more clearly what they mean by stiffness. Are they talking about the

Young's modulus of the material or do they also include the thickness of the exine and intine shells as part of their definition. Also, at some point in the manuscript, the authors say: "Moreover, atomic force microscopy (AFM) measurements revealed that prolonged alkaline treatment caused significant loss in mechanical strength of the exine layer (Extended Data Figs. 16-17)." The word "strength" in mechanics refers to the maximal stress that can be supported by a material. The authors have inferred, at best, the elastic modulus of the material not its strength.

Based on these comments, I believe the claim that "these findings reveal that a softened exine layer plays an important role in modulating the swelling behavior of the microgel particles by allowing greater swelling of a hydrated intine." has not been established.

3) The authors must be more careful with the claims they are making and the references or results they are citing in support of those claims. I'm listing some examples below.

(a) "This chemomechanical approach, inspired by classical soapmaking²², mimics key aspects of angiosperm reproduction pathways while achieving material properties not found in nature." It is not clear how the KOH treatment mimics key aspects of angiosperm reproduction. I suppose the authors are referring to the enzyme pectin methylesterase involved in remodeling the pectin structure of growing pollen tubes. If the mimicking is limited to this, the parallel is quite remote and their results, although interesting on their own, do not inform us in any concrete way about angiosperm reproduction.

(b) "Regarded as practically indestructible, pollen grains are highly dynamic microscale structures that possess unique material characteristic^{6,7}". The references provided are:

6 Huang, C., Wang, Z., Quinn, D., Suresh, S. & Hsia, K. J. Differential growth and shape formation in plant organs. *Proc. Natl. Acad. Sci. U. S. A.* 115, 12359-12364 (2018).

7 Radja, A., Horsley, E. M., Lavrentovich, M. O. & Sweeney, A. M. Pollen cell wall patterns form from modulated phases. *Cell* 176, 856-868 (2019).

I do not see how a paper that does not mention pollen (ref 6) and a mostly theoretical paper (ref 7) support the claim.

(c) "The outermost layer ("exine") is made up of sporopollenin, which is a strong, crosslinked biopolymer¹³, while the inner layer ("intine") is composed of elastic, load-bearing cellulose/hemicellulose tubule structures and pectin^{14,15} (Fig. 1b)". The references provided are:

14 Baskin, T. I. Anisotropic expansion of the plant cell wall. *Annu. Rev. Cell Dev. Biol.* 21, 203-222 (2005).

15 Rojas, E. R., Hotton, S. & Dumais, J. Chemically mediated mechanical expansion of the pollen tube cell wall. *Biophys. J.* 101, 1844-1853 (2011).

The papers do not support the claim made. Also, cellulose microfibrils do not form "tubules". Their representation in Fig. 1b and Fig. 2a,d has open tubes is misleading.

(d) "Biologically, this enzymatic activity is spatially controlled within the pollen wall structure, confounding prior efforts to engineer pollen structures with tunable material properties^{17,18}." The references given are:

17 Wolf, S., Hématy, K. & Höfte, H. Growth control and cell wall signaling in plants. *Annu. Rev. Plant Biol.* 63, 381-407 (2012).

18 Vieira, A. M. & Feijó, J. A. Hydrogel control of water uptake by pectins during in vitro pollen hydration of *Eucalyptus globulus*. *Am. J. Bot.* 103, 437-451 (2016).

The references cited are not attempts to "engineer pollen structures with tunable material properties".

(e) "Under neutral pH conditions, the carboxylic acid functional groups in the pectin molecules are predominately deprotonated (Fig. 2d)²³". The reference give is:

23 Michard, E., Simon, A. A., Tavares, B., Wudick, M. M. & Feijó, J. A. Signaling with ions: The keystone for apical cell growth and morphogenesis in pollen tubes. *Plant Physiol.* 173, 91-111

(2017).

Figure 2d does not establish this critical claim and, as far as I can tell, the reference cited does not either.

(f) A reference is needed for this statement "The pH-dependent swelling response is analogous to that of ionizable polymer networks whose time scales for de-swelling occur similarly within seconds."

(g) "Our investigation of pollen grains and spores from across the plant kingdom reveals that microgel formation is restricted to pollen from eudicot plants." This statement is made on the basis of only three non-eudicot pollen grains or spores: Lycopodium, pine, and cattail. This is not enough to exclude ALL non-eudicots. Moreover, the cattail pollen shows significant swelling (Extended data Figure 30). The area increase would be 50% given that the diameter increases from 20 μ m to 25 μ m.

(h) "Collectively, our experimental and computational results offer fundamental insights into how tuning pollen structure can cause dramatic alterations to material properties, elucidating how the materials science of pollen influences plant reproductive success." It is not clear how the work presented in the manuscript relates to the "reproductive success" of plants.

Reviewer #2:

Remarks to the Author:

Dear Editor

The authors reports a extensive study on production of a stimuli responsive (mono- and divalent cations, pH) sporopollenin microgels from various pollen grains. The manuscript is well-designed and it was written in a reader-friendly manner. The findings of the study was intersting and worth spreading. The discussions are relevant and very informative. The findings of the work should reach wider audience by publication.

I think that the submission can be accepted for publication in the present form.

My recommedation is ACCEPT for publication.

With my best regards,

Dr Idris Sargin

Selcuk University, Konya, Turkey.

Response to Reviewers for NCOMMS-19-11769

Reviewer 1 Comments

The manuscript by Fan and co-workers reports observations on the chemical hydrolysis of sunflower pollen grains with KOH which allows to control the swelling of the grains with changes pH and ion concentration. These experiments are followed by a mechanical analysis of the exine and intine. While some results are interesting, many aspects of the manuscript are confusing and a number of claims seem to extend beyond what the authors have unequivocally established.

Reply: We thank the Reviewer for taking the time to carefully review our manuscript and for providing many excellent comments to help us revise and improve the manuscript. First of all, we are very encouraged by the Reviewer's comment that the results are "interesting." We also thank the Reviewer for providing constructive feedback about points he/she found to be "confusing." Upon reading the Reviewer's comments, we identified several points to clarify in order to better explain the novelty of our work and how it is distinct from past work reported by ourselves and others. We have also performed suggested control experiments where applicable. Below, we provide detailed responses to each point raised by the Reviewer and explain how we have improved the revised manuscript accordingly. Where applicable, we have also provided additional Review-Only Material (ROM) to explain the novelty of our approach and findings.

1) The main result of this work is the discovery that treatment of defatted pollen grains with KOH substitutes the methyl groups for hydroxyl groups in the pectins of the intine wall. I was left with several questions about this process and the data that were presented.

(a) The KOH treatment appears to replicate fairly closely the treatment reported by the same group a few years ago (Mundargi, R. C., Potroz, M. G., Park, J. H., Seo, J., Lee, J. H., & Cho, N. J. (2016). Extraction of sporopollenin exine capsules from sunflower pollen grains. *RSC Advances*, 6(20), 16533-16539). In the earlier work, sunflower pollen grains are treated with 6% KOH for 12hrs while in the current manuscript the treatment is with 10% KOH first for 2hrs and then for up to 12hrs. To my surprise, the conclusion of the 2016 paper, where a weaker solution of KOH was used, is: "The SECs isolated from alkaline lysis are completely damaged and lose their unique spiky microstructure". Scanning electron micrographs of the resulting material (Fig. 1b in the 2016 paper) show an amorphous mass of material. In contrast, the conclusion of the current manuscript is: "Taken together, the findings reveal that incubating pollen in alkaline conditions for extended periods of time (replicating classical soapmaking²²) results in the transformation of hard pollen grains into pliable, soft microgel particles". Although the authors talked about potential degradation of the pollen grains after long exposure to KOH, Fig. 1d of the manuscript shows that there are still pollen grains after 12hr since they have measured their diameter. Also, the so-called SEC shown in the current manuscript have the "spiky microstructure" of untreated pollen grains. I have trouble understanding why the results are so different and why the authors have not cited their own paper and explained the discrepancies.

Reply: We thank the Reviewer for this critical remark and it is a great question. We would like to clarify that the two alkaline processing methods are technically different and yield distinct results in a highly reproducible manner. In earlier work by our group, we had focused our pollen research on developing chemical processing methods to extract sporopollenin exine capsules (SECs), and the paper (Ref. 1, Mundargi, R. C. *et al.*) quoted in the Reviewer's comment dealt with SEC production. It was mainly focused on acid processing methods and the initial alkaline processing method (included as a control experiment in Ref.1) was unsuccessful for extracting intact SECs as discussed in Ref. 1. We have recently reproduced those alkaline processing results again and confirmed their validity.

On the other hand, in a broader context beyond just SEC production, we have carefully studied how different experimental parameters, such as incubation time, static or stirred incubation conditions, washing solvents, centrifugation, and drying procedures, affect the outcome of alkaline processing methods. Within this scope, we discovered how the alkaline processing conditions used in the protocol of the current manuscript yielded intact, soft pollen-derived particles that exhibit microgel-like behavior. A detailed comparison of the alkaline processing methods from Ref. 1 and from the current manuscript under review are presented in **ROM Fig. 1** below and highlight how different alkaline processing conditions can yield damaged pollen particles or intact, microgel-like particles.

ROM Fig. 1. Comparison of two alkaline processing methods and microscopic characterization of the processed pollen samples. Key differences are stirred vs. static incubation, incubation time, washing solvents, centrifugation, and drying conditions.

For example, in additional control experiments, we verified that KOH incubation in static vs. stirred conditions had a significant effect on particle morphology, as indicated in **ROM Fig. 2**. Pollen incubation in 10% KOH at 80 °C for 12 h *under stirred conditions* yielded damaged, fragmented particles while pollen incubation in 10% KOH at 80 °C for 12 h *under static conditions* yielded intact particles without morphological damage.

ROM Fig. 2. Effect of KOH incubation in stirred vs. static conditions on sunflower pollen morphology. Pollen grains were incubated in 10% KOH at 80 °C for 12 h.

On a related note, we agree with the Reviewer that it is useful to cite our past work that tangentially describes another alkaline processing method (we overlooked its citation because the previous work was focused on SEC production and yielded fragmented particles) in order to provide context to the readership and also emphasize that the specific details of the processing protocol are important to control the processing outcome. In the revised manuscript (pg. 16, lines 349-353), these points have been added as follows:

“While alkaline processing conditions can be used to convert hard pollen grains into soft microgel particles, not all alkaline processing protocols work and different results such as particle fragmentation (see, e.g., Ref. 32) can occur depending on processing parameters such as incubation time, static or stirred incubation conditions, washing solvents, and drying procedures.”

Cited Reference

32 Mundargi, R. C. et al. Extraction of sporopollenin exine capsules from sunflower pollen grains. RSC Advances 6, 16533-16539 (2016).

(b) The experimental protocol used in this manuscript involves two successive treatments with KOH, one for 2hrs and a subsequent one for 0 to 12hrs. All other experimental

conditions seem identical. Despite the similarity between the two treatments, the authors claim that their final outcomes are different. The first 2hr treatment would extract the shell (i.e. remove the pollen content and “expose pectin epitopes”) while the second treatment of variable duration would substitute the methyl groups for OH groups (“extensive demethylation of exposed pectin molecules”). Supplementary Fig. 2 makes the same claim graphically. It is not clear why the authors make such a stark distinction between the net result of two identical and successive treatments with KOH. Why take such a categorical view of the experimental protocol? Is it not possible that substitution (demethylation) of pectins begins with the first KOH treatment? Is there not indication of that in Suppl. Fig 3 (0h)?

Reply: We sincerely thank the Reviewer for this comment. First, we would like to address the Reviewer’s comment that the “experimental conditions seem identical” aside from the incubation time format. As mentioned in our reply to the previous comment, there are other significant and important differences between the current protocol such as the static vs. stirred incubation which can affect the processing outcomes (*cf.* ROM Figure 2) along with washing steps and drying protocol. Thus, while the protocols may appear similar at first glance, they lead to different processing outcomes in a highly reproducible manner.

As for the Reviewer’s critical remark about the “categorical view of the experimental protocol,” we appreciate the Reviewer’s suggestion and agree that the description could be rephrased, especially as it pertains to the two KOH incubation processing steps. As the Reviewer points out, we fully agree that substitution (demethylation) of pectin molecules can begin during the first KOH treatment step (as indicated in Supplementary Figure 3). We apologize for any confusion about this point and have clarified the schematic illustration in Supplementary Figure 2 to more clearly represent the two KOH incubation steps, including describing the intine layer as being constituted of ‘weakly’ and ‘highly’ de-esterified intine rather than giving the impression of fully esterified or de-esterified intine. The revised figure is presented below:

Supplementary Figure 2 | Schematic illustration of pollen microgel fabrication process. Microgel fabrication process involves the following four processing steps (as indicated by arrows): **(1)** Treatment with organic solvent to remove excess lipid coating; **(2)** Pollen incubation in alkaline conditions for 2 h at 80 °C **under stirring condition** (“Pollen shell extraction”); **(3)** Extended incubation in alkaline conditions at 80 °C for up to 12 h (“Pollen microgel formation”); and **(4)** Gelation of pollen microgels induced by water rinsing.

Along this line, we verified that the JIM5 and JIM7 antibody labeling experiments support that de-esterified pectin is already present in the pollen shells after the first 2 h KOH treatment (expressed as “0 h” results) while longer incubation times can result in more extensive de-esterification (see **Supplementary Fig. 3** below from revised manuscript). More detailed analysis of the antibody labeling, especially with respect to the weakened intensity of the labeling markers with greater incubation time (from 6 h onward) are explained in our response to comment (d) below.

Supplementary Figure 3 | Immunofluorescence microscopy experiments for detection of de-esterified pectin within processed pollen shells. Prior to experiments, the pollen microgel samples were incubated with **a**, JIM5 and **b**, JIM7 monoclonal antibodies, which recognize weakly and highly esterified pectin molecules, respectively, and both antibodies cannot detect fully de-esterified pectin samples⁵⁸. The data are amended from the original Supplementary Figures 3 and 5, and have been added as Supplementary Figure 3 in the revised manuscript.

Also, related statements in the manuscript have been edited as follows:

Pg. 3, lines 55-59

“In order to expose pectin epitopes, the defatted pollen was first treated with 10 w/v% KOH at 80 °C for 2 h under stirring to efficiently remove cytoplasmic components. Afterwards, the pollen was incubated without stirring in fresh 10 w/v% KOH at 80 °C for an extended period of up to 12 h before centrifugation was performed and this combination of processing steps facilitated microgel formation (Supplementary Fig. 2).”

Pg. 3-4, lines 73-76

“Taken together, the findings reveal that incubating pollen in well-controlled alkaline conditions for extended periods of time (replicating classical soapmaking⁵) can result in the transformation of hard pollen grains into pliable, soft microgel particles.”

Pg. 15, lines 330-332

“(2) **Cytoplasmic removal (1st KOH treatment step)**: Defatted pollen (2 g) was mixed with aqueous 10 w/v% KOH (20 mL) in a 100 mL polytetrafluoroethylene (PTFE) round-bottom flask under magnetic stirring at 400 rpm....”

Pg. 16, lines 339-340

“(3) **Microgel formation (2nd KOH treatment step)**: Fresh 10 w/v% KOH was added to the pollen sample up to a total volume of 40 mL, followed by vortexing at high speed for 2 min....”

(c) It is not clear what is presented in Fig. 2c. At first, I interpreted Fig. 2c as statistics on MANY pollen grains that were treated with KOH in the same way and then exposed to rapid changes in pH. I also assumed that the area of pollen grains was normalized by the area at pH=2 which would explain why the standard deviation is zero for the Area Swelling Ratio of 1. However, I note now that the standard deviation also tends to zero a pH = 12 (Area Swelling Ratio of about 1.8). This result would certainly not be expected if MANY pollen grains are used to produce this curve and the normalization is done only with the area at pH=2. Specifically, it is extremely unlikely that many pollen grains would swell to exactly the same area swelling ratio. This curve must be explained better and ideally replaced with one that has proper statistics on a population of pollen grains.

Reply: We thank the Reviewer for this excellent comment and apologize for the lack of clarity. The data in Figure 2c are time-lapsed optical micrographs of individual pollen particles tethered onto a functionalized surface and these experiments were conducted to measure the *time scale* of microgel particle swelling behavior. For these experiments, the mean \pm s.d. are reported from $n = 5$ particles, as described in the figure legend of the original article. This particle number corresponds to one field of view and we could only focus on one field of view per experiment because we used the highest imaging resolution on our instrument. We repeated the experiment numerous times for each condition and, qualitatively, the results across different repeats agree with the quantitative trend across all experiments. We focused our quantitative analysis on a single image frame ($n = 5$ particles) because the image processing is very time-intensive and the particle size must be determined manually for each particle across each image frame. Since even $n = 5$ particles yielded sufficiently clear data and the experiment was repeated many times, we felt that it was sufficient to present these data for understanding the *kinetics* of the swelling behavior. At the same time, we would like to emphasize that the time-lapsed optical microscopy experiments complemented the main DIPA experiments, which measured the pH-dependent swelling behavior of suspended pollen particles in bulk solution and were conducted on $n = 500$ particles per condition.

We also understand and appreciate the Reviewer's critique about the similar swelling ratio, however, we would like to emphasize that the pollen particles, as produced by nature, are remarkably monodisperse and have uniform material properties (viz. composition and architecture). For example, DIPA measurements showed that the defatted sunflower pollen grains without KOH treatment have a relatively narrow size distribution, *i.e.*, diameters of 36.8 ± 1.7 and 39.5 ± 1.9 μm at pH 2 and pH 10 conditions, respectively. Also, the 6 h KOH-treated microgel particles have similar levels of uniformity, *i.e.*, diameters of 28.3 ± 1.6 and 49.6 ± 3.4 μm at pH 2

and 10 conditions, respectively (**ROM Table 1** in Part 2). As such, the relatively small (but still apparent) measurement errors reported in the time-lapsed optical microscopy experiments after pH equilibration (swelling/de-swelling, first and last 2 s in Fig. 2c) are consistent with the DIPA measurement results.

(d) Suppl Fig. 3 shows JIM5 labelling of pollen grains treated with KOH for different time durations. Why is JIM5 labelling nearly absent at 6 and 12 hrs if the pollen grains are still able to swell after such exposure? Where are the de-esterified pectins in those treatments? Also note that the 0hr treatment shows extensive JIM5 fluorescence suggesting that the 2hr KOH pretreatment already started the de-esterification process.

Reply: We thank the Reviewer for this excellent question. The JIM5 and JIM7 antibodies are commonly used to detect *partially* methyl-esterified residues, whereby JIM5 and JIM7 recognize weakly and highly esterified pectin molecules, respectively, and both antibodies cannot detect fully de-esterified samples^{2,3}.

Thus, the JIM5 signal increases for samples up to around 3 h due to moderate de-esterification compared to the defatted control sample, but then becomes negligible at 6 h once the pectin molecules become fully de-esterified. By contrast, the JIM7 signal is seen for the defatted control in which case the intine is composed of highly esterified pectin molecules, but the signal diminishes due to increasing levels of pectin de-esterification in KOH-treated samples.

Along with the amendments in Supplementary Fig. 3, related statements in the manuscript have been also edited as follows:

Pg. 3, lines 60-68

“To verify de-esterification, we conducted immunofluorescence microscopy experiments using the monoclonal JIM5 antibody that recognizes weakly esterified epitopes of pectin²¹ (Fig. 1c). The results confirmed successful processing as indicated by increased de-esterification after 3 h incubation while longer incubation periods resulted in more extensive de-esterification of pectin molecules²² (Supplementary Fig. 3a). Dynamic image particle analysis (DIPA) further revealed that the treated particles swelled from approximately 35 to 43 μm in diameter after 6 h incubation, which was judged to be the optimal condition based on the greatest extent of particle swelling (Fig. 1d). These results were complemented by immunolabeling experiments with JIM7 antibody, which recognizes highly esterified epitopes of pectin²¹ (Supplementary Fig. 3b).”

(e) Finally, some controls seem necessary. For example, have the authors treated their pollen grains with PME to remove the methyl groups enzymatically? This simple control experiment would help solidify the claims made in the first part of the manuscript (Figs. 1 and 2). Also, I would have liked to see in Fig. 2c the effect of pH on defatted pollen grains that have NOT been treated with KOH. It seems the authors have done this experiment since they show pollen grains with this treatment in Fig. 3c ME/I = 3. The pollen grains do show substantial swelling with pH although not as much as the KOH-treated one. Fig. 2c must include proper statistics for the pH swelling of KOH-treated and non-treated pollen grains.

Reply: We thank the Reviewer for these excellent suggestions and have performed the requested control experiments:

1. We tested the effect of solution pH on defatted pollen grains that have NOT been treated with KOH. The DIPA measurements ($n = 500$ particles) and time-lapsed optical microscopy ($n = 5$ particles) confirmed that the defatted pollen grains are not sensitive to changes in solution pH. This information has been included in the revised manuscript as **Supplementary Figure 9** (DIPA measurements) and incorporated in **Figure 2c** (time-lapsed optical microscopy measurements). The updated figures are presented below along with text revisions.
2. We have also enzymatically treated processed pollen microgel particles with pectinase – which catalyzes the removal of pectin molecules – in order to verify that pectin molecules play a key role in controlling the pollen’s mechanical behavior. Upon pectinase treatment, the microparticles had negligible pH-dependent swelling. The data are presented below as **ROM Fig. 3**.
3. We also considered performing pectin methylesterase (PME) experiments but decided that, among enzymatic processing control options, the pectinase option was more appropriate for our study design and data analysis strategy. In the pectinase experiments, the pollen particles could be treated using our exact protocol (including the initial 2 h KOH incubation step which is necessary for extracting the pollen shells) and then the pectinase enzyme treatment to remove pectin before conducting the pH-dependent swelling experiments. By contrast, the PME experiments would ideally be conducted whereby PME is the only component that is capable of causing pollen de-esterification but the initial 2 h KOH incubation step is necessary for extracting the pollen shells (thus, there would be pectin de-esterification from a combination of enzymatic and chemical factors) so we decided that the pectinase experiment was a more useful enzymatic control and it verified the importance of pectin molecules in dictating the stimuli-responsive microgel behavior.

Control experiments with defatted pollen grains

Pg. 4, lines 87-90

“In marked contrast, when the solution pH was changed from 12 to 2, the microgel particles underwent relatively faster de-swelling or compression while defatted, unprocessed pollen grains exhibited negligible pH-responsive behavior in both directions (Fig. 2c).”

Supplementary Figure 9 | Characterization of the pH-dependent swelling behavior of defatted sunflower pollen without alkaline processing treatment. **a**, Time-lapsed optical micrographs of defatted sunflower pollen particles tethered on a functionalized surface. The solution pH was changed from pH 2 to 12 (top series) and from pH 12 to 2 (bottom series). **b**, Top: Diameter of defatted sunflower pollen particles that did not undergo the alkaline processing protocol, as measured by DIPA ($n=500$ particles per condition). Bottom: Optical micrographs of representative, individual sunflower pollen grains in pH 2, 4, 7, 10, and 14 conditions at 10 \times magnification. **c**, Corresponding data for defatted sunflower pollen particles that underwent the alkaline processing protocol.

Figure 2 | Tunable pectate interactions enable rapid, stimuli-responsive material properties.
 c, Quantitative comparison of pH-induced pollen swelling and de-swelling behavior....

Control experiment with enzymatically treated pollen grains

ROM Figure 3. Swelling characterization of pectinase-treated pollen microgel particles. pH-dependent swelling behavior of pollen microgel particles without or with pectinase treatment was characterized by DIPA measurements. Data are reported as mean \pm s.d. ($n = 500$ particles per condition).

2) A mechanical analysis follows the swelling analysis of KOH-treated pollen grains. At this point, the manuscript becomes more obscure and I am concerned that the authors have not captured properly the mechanical behavior of the pollen grains.

Reply: We thank the Reviewer for very thoughtful and valuable feedback about the mechanical analysis section of the paper. In the revised manuscript, we have carefully provided more details about the AFM measurement methods along with control experimental results for validation of our measurement approach. We also have clarified our findings on the mechanical behavior of the pollen grains, particularly regarding the stretching mechanism of pollen shells as part of the extensive swelling of pollen microgels.

(a) I am concerned about the validity of the protocol used to infer the elastic modulus of the exine and intine. According to the authors, intact pollen grains were used to measure the modulus of the exine. In this configuration, it is not clear how the specific modulus of the exine can be disentangled from the large scale deformation of the entire pollen grain under the force applied by the AFM. On the other hand, the modulus of the intine was measured on open fragments. Presumably the intine is still bound to the exine. How can the authors claim to have measured the elastic modulus of the intine if they are probing a thin layer of material that is still mechanically connected to the exine? Moreover, the intine wall is probably anisotropic in its structure.

Reply: We thank the Reviewer for this valuable comment. Our AFM stiffness measurement protocol is based on a recently published literature reference⁵ from a top nanomechanical group working on fundamental characterization of pollen grains, and we followed a similar approach while setting the testing parameters according to the following criteria: (1) 10% depth rule; and (2) assumption of isotropic exine/intine layers. We explain each point below:

1. In order to avoid the substrate effect, we followed the 10% depth rule whereby the indentation depth should be less than $1/10^{\text{th}}$ of the layer thickness. This rule has been well-known for various film stiffness measurements by AFM or nanoindentation⁶. In the dry condition, the average thickness of the exine and intine layers was $0.58\ \mu\text{m}$ and $0.54\ \mu\text{m}$, respectively. In the wet condition, the pollen microgel particles swell, where only the intine layer was significantly increased to $\sim 1.5\ \mu\text{m}$ while the exine remained stable (**ROM Fig. 4**). Likewise, we carefully chose the indentation depth, which could only induce local deformation without large-scale deformation. Thus, we selected to analyze data where the indentation depths were in the range of $20\text{--}60\ \text{nm}$ (within 10% of $0.58\ \mu\text{m}$) for dry and wet exine, $<50\ \text{nm}$ for dry intine (within 10% of $0.54\ \mu\text{m}$), and $<100\ \text{nm}$ for wet intine (within 10% of $1.5\ \mu\text{m}$).

ROM Figure 4. Schematic illustration of AFM measurements with pollen microgel particles.

a, Nanoindentation experiments on the whole pollen shell consisting of two layers, exine and intine, for the stiffness measurement of exine with indentation depths of 20–60 nm under indentation loads of 3–6 μN . The pollen samples were prepared in both dry and wet conditions. **b**, The AFM tip can approach either the exine or intine layer depending on the orientational direction of the specimens, which were prepared from pollen shell fragments. The average thickness of the exine layer was $\sim 0.6 \mu\text{m}$ for defatted and 0 h KOH-treated pollen specimens, and became thinner ($\sim 0.5 \mu\text{m}$) with increasing KOH incubation time. Therefore, the indentation depths for measuring the exine of fragmented specimens were kept the same as those for the entire sample across both dry and wet conditions. By contrast, the thickness of the intine layer had significant variation between dry and wet conditions (0.5–1.5 μm), and thus the indentation depths were carefully controlled to be below 60 nm for dry samples and 50–100 nm for wet samples.

Additionally, we also checked all the data outside of those ranges, and confirmed that the retrieved stiffness, or modulus, of the exine or intine layers was very similar to the retrieved data from within the ranges of the indentation depths. This is because we set our indentation depth ranges following the 10% depth rule of the layer thickness in a very conservative manner. Biological samples typically have inhomogeneous thickness distributions, and thus sometimes larger indentation depths can provide valid data. We also tested rate-dependency and tip stiffness effects during the AFM measurements. Regardless of the different measurement conditions, all the values obtained from the tests showed no statistically significant difference (**Supplementary Figure 31**). Finally, we performed more than 20 indentations per sample for at least five pollen samples.

Supplementary Figure 31 | Effects of various AFM parameters on stiffness measurement of exine and intine layers of dry, defatted pollen grains. a, Representative force-displacement

curves from AFM measurements using two types of AFM probes, an aluminum reflex-coated silicon cantilever PPP-NCHR (Nanosensors, Neuchâtel, Switzerland) with a typical spring constant of ~ 42 N/m and a tip end radius of 5 nm and a diamond cantilever TD26135 (Micro Star Technology, Texas, USA) with a spring constant of ~ 150 N/m and a tip end radius of ~ 5 nm. **b** and **c**, Young's moduli of exine (b) and intine (c) retrieved from force-displacement curves after AFM measurements using two AFM probes, NCHR and diamond probes at a maximum contact force of 20 μ N with setpoint of 6 μ N with zero contact time and an approach speed of 0.8 μ N/s. AFM measurements were performed using the NCHR probe, varying the maximum contact force (10 to 30 μ N), contact time (0 to 1, 3, and 5 s), and approach speed (0.5 to 1.2 μ N/s). NS denotes non-significant differences among all measurement conditions. **d**, Representative force-displacement curves of exine layer measurements in the wet condition. **e**, Representative force-displacement curves of intine layer measurements in the wet condition.

In the revised manuscript, this information has been added as follows:

Pg. 19-20, lines 500-532

“The mechanical properties of defatted pollen grains and microgel particles were characterized by conducting AFM force-distance measurements in order to determine the Young's modulus of the material^{35,36}. For wet samples, pollen microgel samples (5 μ L) were dispersed in the appropriate medium (50 μ L), and the mixture was spread onto a petri dish (Nunc, Roskilde, Denmark) by pipette. Then, excess liquid was aspirated and the partially hydrated samples remained on the glass slide, followed by immediate experiment. For dry samples, the pollen sample was directly spread onto the surface of a petri dish before measurement. Intact pollen particles were chosen to measure the Young's modulus of the exine layer and fragmented pollen particles, which exposed the inner layer, were chosen to measure the Young's modulus of the intine layer. In order to avoid the substrate effect, we followed the 10% depth rule where the indentation depth should be less than $1/10^{\text{th}}$ of the layer thickness³⁷. The average thickness of the exine layer was ~ 0.6 μ m for defatted pollen particles and 0 h KOH-treated pollen specimens, and became thinner with increasing KOH treatment time (~ 0.5 μ m). Thus, the indentation depths for exine measurements were set in the range of 20–60 nm under indentation loads of 3–6 μ N. On the other hand, the intine layer was swollen in wet conditions, with a thickness range from 0.5 μ m to 1.5 μ m. Thus, for dry samples, the indentation depths of intine measurements were carefully limited up to ~ 60 nm, while for wet samples, the indentation depths were in the range of 50 to 100 nm. For all measurements, the NX-10 AFM instrument (Park Systems, Suwon, South Korea) was used with two AFM probes, i) an aluminum reflex-coated silicon cantilever PPP-NCHR (Nanosensors, Neuchâtel, Switzerland) with a typical spring constant of ~ 42 N/m, and a tip end radius of 5 nm, and ii) a diamond cantilever TD26135 (Micro Star Technology, Texas, USA) with a spring constant of ~ 150 N/m and a tip end radius of ~ 5 nm. We confirmed that both AFM probes provided almost identical values of Young's moduli for exine and intine layers regardless of the various indenting parameters such as maximum contact forces, contact time or approach speed (Supplementary Fig. 31). Thus, measurements were conducted at various positions (more than 16 data points) in a 5 μ m \times 5 μ m area at an approach speed of 0.8 μ m/s with the maximum loading force of 4.8 μ N and zero contact time. Before experiments, the AFM cantilever was rinsed with water and ethanol, and treated with a UV light cleaner for 30 min in order to remove organic contaminants. The spring constant and sensitivity of the deflection signal were also calibrated by

recourse to the thermal vibration of the AFM cantilever³⁸ by employing the commercial software (XEP, Park Systems). *The force-versus-displacement curves were corrected by subtracting the deflection distance of the AFM probe from the total displacement.*”

2. We assumed that both the exine and intine layers are isotropic in this experiment since we focused on the differences in stiffness between the exine and intine. Also, we used an AFM probe with a tip end radius of 5 nm, and thus the indentation depths were significantly larger than this tip end radius. Thus, the classical Hertz model was used for data analysis, which is consistent with another recent nanomechanical report on pollen grains.

In the revised manuscript, more detailed information about these aspects has been added as follows:

Pg. 20, lines 534-542

“For data analysis, we assumed that both the exine and intine layers were isotropic. Also, we used an AFM probe with a tip end radius of 5 nm, and thus the indentation depths (20–100 nm) were significantly larger than the tip end radius. Thus, the classical Hertz model was used for data analysis^{36,39}. The Hertz model³⁶ was fitted to the force-versus-displacement curves using a commercial software analysis program (XEI, Park Systems) and the Python script language (Supplementary Fig. 31). We treated the geometry of the AFM tip as a parabolic model whereby it has a tip radius of R_c , so that the force () can be expressed as:

$$F = \frac{4\sqrt{3}}{3} \frac{E\delta^{3/2}}{1-\nu^2}, \quad (1)$$

where E is Young’s modulus, ν is the Poisson’s ratio, and δ is the indentation depth. The Poisson’s ratio ν was set at 0.5, which is typical for natural materials.”

(b) I could not find what was the tip radius of the AFM probe, nor the final displacement of the probe. In fact, the authors do not present the force-displacement curves that are normally used to infer the elastic properties of materials. The authors only mention that “the approach speed of the AFM cantilever was fixed to 0.8 $\mu\text{m}/\text{sec}$ with a maximum loading force of 4.8 μN .” Measuring material properties of soft elastic materials with an AFM is not a simple task when working with complex structures such as the pollen grain. The authors present their results as if they were routine measurements. A better description of the AFM measurements and some controls are necessary to validate the moduli reported in the manuscript.

Reply: We thank the Reviewer for this suggestion. Since one paper was already published regarding the AFM measurement of pollen exine⁵, and our method was mostly consistent with those methods and results, we did not provide the details of our measurement methods and validations in our original manuscript. Nevertheless, we fully agree that we should provide a better description of the AFM measurements and have done so in the revised manuscript. Additional methods descriptions for experiments and data analysis were added too, as described in detail in our response to the previous comment.

(c) During harmomegathy, the volume of the pollen grain increases substantially but this volume increase is achieved in many species not by stretching the exine but by unfolding or unrolling it. In other words, swelling of pollen grains during harmomegathy does not necessarily involve stretching of the exine. I believe some unfolding or unrolling of the exine is seen in the pollen grains used to illustrate the simulations in Fig. 3c, as indicated by the change in curvature of the exine segments. The case of $M_E/I = 1.6$ in Fig. 3c is particularly striking because the normally convex exine segments become flat or even concave as the pollen swells. Thus the exine shell is unrolling to accommodate the increase in pollen grain volume. The finite element simulations are presented and implemented as if the exine is simply a hard shell that must be STRETCHED to allow swelling of the pollen grain. This is not a fair assumption. In fact, close inspection of the simulations for $M_E/I = 1.6$ reveals that the simulations are not reproducing faithfully the deformation of the pollen grains. If the swelling during harmomegathy is achieved without much softening or stretching of the exine, is it not possible that KOH-treated pollen grains can also swell, at least to some extent, without stretching the exine? If this is the case, the resistance offered by the exine is not a simple stretching deformation but one that involves also bending of the exine shell.

Reply: We thank the Reviewer for these valuable comments. We agree that, during harmomegathy, the volume decrease of the pollen shell should occur through closing its apertures or folding its shell structure. Also, during harmomegathy, hydration and dehydration occur depending on the environmental conditions around a pollen grain^{10,11}. As Katifori *et al.* reported in their paper, the unfolding/folding of pollen shells can accommodate this harmomegathy process with minimal energy cost, with which we strongly agree¹¹. However, in our current study, we focused on the subsequent “germination” process after harmomegathy is halted. During this germination process, lipid components from the outer layer of pollen grains are gradually dispersed, allowing increased water uptake^{12,13}. Through this process, apertures are opened, which accelerates water uptake, leading to germination with significant swelling of pollen grains¹³. Meanwhile, in the exine layer, pollen cements, which occupy the cavity of the sporopollenin network in the exine, are released. Along these lines, we found that our processed pollen microgel samples exhibit key similarities to germinated pollen grains as follows:

1. During germination, the maximum swelling volume of pollen grains should be 1.5–3 times¹³, which cannot be reached by just opening the apertures of pollen grains based on the pollen structure. Particularly, sunflower pollen has spiky shells without any foldable shell design (*e.g.*, wrinkles or folded structure) except for apertures. For instance, 0 h KOH-treated sunflower pollen showed a small change in the equatorial diameter ($R_{pH 10} / R_{pH 2} \approx 1.2$) from pH 2 to pH 10, whereas 6 h KOH-treated sunflower pollen showed a more significant change in the equatorial diameter ($R_{pH 10} / R_{pH 2} \approx 1.8$), as shown in **ROM Table 1**. Also, the spikes restrict extensive folding of the pollen shell during harmomegathy due to structural hindrance. As indicated in **Supplementary Figure 9**, the areal change in the pollen microgel particle increases with longer KOH incubation time. Thus, the increased volume of pollen grains can be achieved by further expansion of pollen through stretching of the pollen shell.

ROM Table 1. Average equatorial diameters of defatted pollen grains and pollen grains after treatment with 10% KOH for various incubation times at pH 2 and pH 10.

Type of Pollen grains	Average equatorial diameter at pH 2	Average equatorial diameter at pH 10	Ratio ($R_{pH\ 10} / R_{pH\ 2}$)
Defatted pollen	$36.78 \pm 1.74 \mu\text{m}$	$39.48 \pm 1.90 \mu\text{m}$	1.07
0 h KOH-treated pollen	$32.35 \pm 1.62 \mu\text{m}$	$38.41 \pm 2.12 \mu\text{m}$	1.19
3 h KOH-treated pollen	$30.47 \pm 2.08 \mu\text{m}$	$46.56 \pm 3.07 \mu\text{m}$	1.53
6 h KOH-treated pollen	$28.32 \pm 1.62 \mu\text{m}$	$49.58 \pm 3.41 \mu\text{m}$	1.75
12 h KOH-treated pollen	$26.93 \pm 1.91 \mu\text{m}$	$52.31 \pm 3.88 \mu\text{m}$	1.94

2. We also found that opening the apertures of pollen microgel particles is a very fast process, but increasing the swelling volume does require some additional time to reach its equilibrium. Thus, in the supplementary figures and videos, during the cyclical pH changes, initial expansion from pH 2 to pH 12 or final compression from pH 12 to pH 2 is relatively faster. We indicated that this aperture opening or unfolding process as reversible harmomegathy when the stigma condition is not sufficient and pollen grains need to restore harmomegathy. Then, after the germination process is initiated, a significant volume change accompanied by structural changes in the exine and intine layers will lead to pollen tube growth¹³.
3. Since pollen grains are already open to the external environment due to the defatting process and pollen shell extraction, the hydrated intine can facilitate hydration of pollen grains even though the exine is still rigid (**Supplementary Figure 9 and ROM Table 1**). In this case, the exine only allows aperture opening or unfolding of pollen with minimal volume change (*e.g.*, 0 h KOH-treated pollen). The swelling of intine or water uptake of pollen is significantly constrained by the rigid exine layer. Thus, germination is efficiently ceased or suspended until pollen allows more hydration from the environment (**Supplementary Figure 29**).

Supplementary Figure 29 | Pollen hydration/dehydration cycle of angiosperms with corresponding FEA simulation images. The top sequence shows a dry pollen grain undergoing the water uptake process during germination. In situations where complete hydration is not possible, germination might be aborted, and the apertures close again as shown in the bottom sequence¹³.

4. Through the second KOH treatment, the exine layer exhibited significant structural changes, becoming more porous (**Supplementary Figure 10**). During germination, the exine also releases pollen cement or lipid components, increasing cavity size and hydrophilicity in order to accommodate fast water uptake¹³. Moreover, through the AFM measurements, we confirmed that the exine layer became less stiff (**Supplementary Figure 31d,e**).

Supplementary Figure 10 | Scanning electron microscopy (SEM) images of defatted pollen grains and pollen shells after treatment with 10 w/v% KOH for various incubation times. Top: SEM image of entire pollen microgels after defatting without or followed by 10 w/v% KOH incubation (2nd KOH treatment) for 0–12 h. Longer KOH treatment resulted in greater opening of the apertures. Bottom: Surface morphology of exine layers at higher magnification. Exines of the defatted pollen grain exhibited a dense and smooth surface morphology with a few microscale pores around the spikes, while the exine surfaces of KOH-treated pollen are rough and porous with exposed sporopollenin skeleton. The increased porosity of the exine surface was attributed to the release of pollen cement (“pollenkitt”) due to KOH treatment^{13,28}.

5. The Reviewer pointed out that “the normally convex exine segments become flat or even concave as the pollen swells.” We agree that concave or flat pollen shell segments were clearly observed in some cases. We speculate that inhomogeneous material properties or thickness across the pollen shell may induce abnormal shapes during swelling. In future work, we intend to investigate this abnormal swelling mechanism of pollen microgel particles by characterizing the spatial distribution of structural and mechanical measurements of pollen shells from both normal and abnormal swelling pollen specimens.

(d) The authors should define more clearly what they mean by stiffness. Are they talking about the Young’s modulus of the material or do they also include the thickness of the exine and intine shells as part of their definition. Also, at some point in the manuscript, the authors say: ‘Moreover, atomic force microscopy (AFM) measurements revealed that prolonged alkaline treatment caused significant loss in mechanical strength of the exine layer (Supplementary Figs. 16-17).’ The word “strength” in mechanics refers to the maximal

stress that can be supported by a material. The authors have inferred, at best, the elastic modulus of the material not its strength.

Reply: We thank the Reviewer for the thoughtful comment. We defined stiffness as the modulus value of the exine or intine layer. The thickness ratio of those two layers, exine and intine, is important, and in this study, we treated their apparent dry thicknesses as almost the same based on experimental observations. Also, we agree that “strength” is not a proper description in the sentence. Therefore, we reworded the description of the stiffnesses of the exine and intine layers to instead discuss the moduli of the exine and intine layers throughout the whole manuscript and supplementary information as follows:

Pg. 5, lines 107-109

“Moreover, atomic force microscopy (AFM) measurements revealed that prolonged alkaline treatment caused a significant decrease in the elastic modulus of the exine layer (Supplementary Figs. 11-12).”

(e) Based on these comments, I believe the claim that “these findings reveal that a softened exine layer plays an important role in modulating the swelling behavior of the microgel particles by allowing greater swelling of a hydrated intine.” has not been established.

Reply: We thank the Reviewer for this comment. As we discuss in our responses to previous comments, the results support that a rigid exine layer significantly constrained the swelling of the intine layer, thus suppressing the volume change of pollen microgel particles as indicated in **Supplementary Figure 9** and **ROM Table 1**. We also provided experimental evidence showing the structural change of the exine layer, including increased porosity, in addition to AFM nanomechanical measurements on the exine layer (**Supplementary Figure 10**). Thus, we believe that our findings on the interplay between the exine and intine layers is still one of the key factors to modulate the swelling behavior of pollen microgel particles.

3) The authors must be more careful with the claims they are making and the references or results they are citing in support of those claims. I’m listing some examples below.

Reply: We thank the Reviewer for this important advice and we have carefully checked the references accordingly. In some cases, we tried to incorporate conceptually related references in order to emphasize what inspired a statement based on work in related fields since pollen materials science is a new and emerging field. We appreciate the Reviewer’s efforts to keep the paper citations grounded and we have added more technical references to support specific facts and to complement the conceptually related references.

(a) “This chemomechanical approach, inspired by classical soapmaking²², mimics key aspects of angiosperm reproduction pathways while achieving material properties not found in nature”. It is not clear how the KOH treatment mimics key aspects of angiosperm reproduction. I suppose the authors are referring to the enzyme pectin methylesterase involved in remodeling the pectin structure of growing pollen tubes. If the mimicking is

limited to this, the parallel is quite remote and their results, although interesting on their own, do not inform us in any concrete way about angiosperm reproduction.

Reply: We thank the Reviewer for his/her critical feedback and we were indeed referring to how the pectin methylesterase enzyme is involved in remodeling the pectin structure of growing pollen tubes. We appreciate that the Reviewer thinks our findings are interesting and also understand that these findings can stand on their own merit while the potential link to angiosperm reproduction is limited at present and not directly tested in this paper. As such, in the revised manuscript, we have decided to rephrase the sentence as follows:

Pg. 1, lines 21-24

“This chemomechanical approach, inspired by classical soapmaking⁵, mimics and expands on the function of plant enzymes in order to remodel pollen shells and impart materials properties not found in nature.”

(b) ‘Regarded as practically indestructible, pollen grains are highly dynamic microscale structures that possess unique material characteristic^{6, 7}’. The references provided are: 6 Huang, C., Wang, Z., Quinn, D., Suresh, S. & Hsia, K. J. Differential growth and shape formation in plant organs. *Proc. Natl. Acad. Sci. U. S. A.* 115, 12359-12364 (2018).7 Radja, A., Horsley, E. M., Lavrentovich, M. O. & Sweeney, A. M. Pollen cell wall patterns form from modulated phases. *Cell* 176, 856-868 (2019). I do not see how a paper that does not mention pollen (ref 6) and a mostly theoretical paper (ref 7) support the claim.

Reply: We agree with the Reviewer that better references should be identified for this statement. We have decided to keep Ref. 7 because it is an important, cutting-edge reference about the origin of dynamic microscale structures in pollen, and also added new supporting references from reviews/book chapters and original works as follows:

Pg. 2, lines 34-36

“Regarded as practically indestructible^{1,2}, pollen grains are highly dynamic microscale structures that possess unique material characteristic⁸⁻¹⁰.”

Cited References

- 1 Birks, H. J. B., Birks, H. H. & Ammann, B. The fourth dimension of vegetation. *Science* **354**, 412-413 (2016).
- 2 de Miranda Chaves, S. A. Pollen Grains, Landscapes, and Paleoenvironments. *Foundations of Paleoparasitology*, 205.
- 8 Blackmore, S., Wortley, A. H., Skvarla, J. J. & Rowley, J. R. Pollen wall development in flowering plants. *New Phytol.* **174**, 483-498 (2007).
- 9 Radja, A., Horsley, E. M., Lavrentovich, M. O. & Sweeney, A. M. Pollen Cell Wall Patterns Form from Modulated Phases. *Cell* **176**, 856-868. e810 (2019).
- 10 Mackenzie, G., Boa, A. N., Diego-Taboada, A., Atkin, S. L. & Sathyapalan, T. Sporopollenin, the least known yet toughest natural biopolymer. *Front. Mater.* **2**, 66 (2015).

(c) “The outermost layer (“exine”) is made up of sporopollenin, which is a strong, crosslinked biopolymer¹³, while the inner layer (“intine”) is composed of elastic, load-bearing cellulose/hemicellulose tubule structures and pectin^{14,15} (Fig. 1b)”. The references provided are: 14 Baskin, T. I. Anisotropic expansion of the plant cell wall. *Annu. Rev. Cell Dev. Biol.* **21**, 203-222 (2005). 15 Rojas, E. R., Hotton, S. & Dumais, J. Chemically mediated mechanical expansion of the pollen tube cell wall. *Biophys. J.* **101**, 1844-1853 (2011). The papers do not support the claim made. Also, cellulose microfibrils do not form “tubules”. Their representation in Fig. 1b and Fig. 2a, d has open tubes is misleading.

Reply: We agree with the Reviewer that the cellulose-hemicellulose structure is better described as a “microfibril” network rather than a “tubule” structure because the intine layer shows various nanoscale structural features that are entangled laterally to form ribbons, strands, or cables¹⁴. In the original manuscript, we represented the cellulose microfibril in a rod shape to distinguish it from other components, but we agree that the open tubes should be improved. Thus, in the revised manuscript, we replaced the “tubule” structures with “microfibril” network structures with appropriate literature citations, and modified the relevant figures (Figs. 1b, 2a, and 2d) as follows (additions in yellow highlight):

Pg. 2, lines 45-48

“The outermost layer (“exine”) is made up of sporopollenin, which is a strong, crosslinked biopolymer¹⁰, while the inner layer (“intine”) is composed of elastic, load-bearing cellulose/hemicellulose **microfibrils** and pectin^{16,17} (Fig. 1b).”

Cited References

- 10 Mackenzie, G., Boa, A. N., Diego-Taboada, A., Atkin, S. L. & Sathyapalan, T. Sporopollenin, the least known yet toughest natural biopolymer. *Front. Mater.* **2**, 66 (2015).
- 16 Rojas, E. R., Hotton, S. & Dumais, J. Chemically mediated mechanical expansion of the pollen tube cell wall. *Biophys. J.* **101**, 1844-1853 (2011).
- 17 Heslop-Harrison, Y. & Heslop-Harrison, J. The Microfibrillar Component of the Pollen Intine Some Structural Features. *Ann. Bot.* **50**, 831-842 (1982).

Figure 1 | ...**b**, Schematic illustration of stimuli-responsive pollen capsule behavior at multiple length scales. Left: Pollen structural components. Middle: Cellulose microfibrils organized by hemicellulose and pectin in the intine layer. Right: Hydrolytic conversion of pectin to pectate within the intine layer. **c**, ...

Figure 2 | ... **a**, Schematic illustration of pH-induced changes in pectin structure. **b**, ... **d**, Schematic illustration of ion-induced changes in pectin structure. **e**,...

(d) ‘Biologically, this enzymatic activity is spatially controlled within the pollen wall structure, confounding prior efforts to engineer pollen structures with tunable material properties^{17, 18}.’ The references given are: 17 Wolf, S., Hématy, K. & Höfte, H. Growth control and cell wall signaling in plants. *Annu. Rev. Plant Biol.* **63**, 381-407 (2012). 18 Vieira, A. M. & Feijó, J. A. Hydrogel control of water uptake by pectins during in vitro pollen hydration of *Eucalyptus globulus*. *Am. J. Bot.* **103**, 437-451 (2016). The references cited are not attempts to “engineer pollen structures with tunable material properties”.

Reply: We agree with the Reviewer that this statement should be adjusted (especially the reference location) and, in the revised manuscript, have revised the statement as follows (additions in yellow highlight):

Pg. 2, lines 50-52

“Biologically, this enzymatic activity is spatially controlled within the pollen wall structure^{19,20}, and inspires biomimetic strategies to engineer pollen structures with tunable material properties.”

19 Vieira, A. M. & Feijó, J. A. Hydrogel control of water uptake by pectins during in vitro pollen hydration of *Eucalyptus globulus*. *Am. J. Bot.* **103**, 437-451 (2016).

20 Knox, J. P., Linstead, P. J., King, J., Cooper, C. & Roberts, K. Pectin esterification is spatially regulated both within cell walls and between developing tissues of root apices. *Planta* **181**, 512-521 (1990).

(e) ‘Under neutral pH conditions, the carboxylic acid functional groups in the pectin molecules are predominately deprotonated (Fig. 2d)²³’. The reference give is: 23 Michard,

E., Simon, A. A., Tavares, B., Wudick, M. M. & Feijó, J. A. Signaling with ions: The keystone for apical cell growth and morphogenesis in pollen tubes. *Plant Physiol.* 173, 91-111 (2017). Figure 2d does not establish this critical claim and, as far as I can tell, the reference cited does not either.

Reply: We apologize for the confusion on this point. The reference we cited is about the divalent cation effect (and its relationship to charged functional groups), and in the revised manuscript, we added an additional reference about the apparent acid dissociation constant value of pectin (pH value at which there is 50% dissociation). In the revised manuscript, the sentence has been amended as follows:

Pg. 4-5, lines 96-99

“Under neutral pH conditions, the carboxylic acid functional groups in the pectin molecules are predominately deprotonated²⁶ and thus sensitive to divalent cations²⁷(Fig. 2d). Time-lapsed optical microscopy experiments on immobilized pollen particles provided direct evidence of the dynamic response to calcium ions (Fig. 2e).”

Cited References

- 26 Michard, E., Simon, A. A., Tavares, B., Wudick, M. M. & Feijó, J. A. Signaling with ions: the keystone for apical cell growth and morphogenesis in pollen tubes. *Plant Physiology*, pp. 01561.02016 (2016).
- 27 Pacinia, E. & Hesseb, M. Pollenkitt – Its composition, forms and functions. *Flora* **200**, 399–415 (2005).

(f) A reference is needed for this statement “The pH-dependent swelling response is analogous to that of ionizable polymer networks whose time scales for de-swelling occur similarly within seconds.”

Reply: We thank the Reviewer for this great suggestion. In the revised manuscript, we provided detailed references about the pH-responsive swelling/de-swelling behaviors observed in polymer networks as follows (additions in yellow highlight):

Pg. 4, lines 90-91

“The pH-dependent swelling response is analogous to that of ionizable polymer networks whose time scales for de-swelling occur similarly within seconds^{24,25}.”

Cited References

- 24 Zhao, B. & Moore, J. S. Fast pH-and ionic strength-responsive hydrogels in microchannels. *Langmuir* **17**, 4758-4763 (2001).
- 25 BeMiller, J. N. An introduction to pectins: structure and properties. *Chemistry and function of pectins* **310**, 2-12 (1986).

(g) “Our investigation of pollen grains and spores from across the plant kingdom reveals that microgel formation is restricted to pollen from eudicot plants.” This statement is made on the basis of only three non-eudicot pollen grains or spores: Lycopodium, pine, and cattail. This is not enough to exclude ALL non-eudicots. Moreover, the cattail pollen shows significant swelling (Supplementary Figure 30). The area increase would be 50% given that the diameter increases from 20µm to 25µm.

Reply: We thank the Reviewer for this comment and agree that the statement of “microgel formation is restricted to pollen from eudicot plants” should be restricted to “among the tested species” and we have rephrased our conclusions accordingly. As the Reviewer points out, for some other pollen species, there is modest swelling observed at the highest tested pH condition (pH 14) but pollen species in that category do not form microgels. This is reported in our study and also consistent with ongoing studies of additional pollen species (see **ROM Fig. 6**). Among the tested pollen species, the only species that formed microgel particles exhibited pH-dependent swelling/de-swelling behavior across the pH spectrum, including across pH 2 to pH 12. In the revised manuscript, we have revised our commentary as follows (additions in yellow highlight):

Pg. 1, lines 26-29

“Our investigation of pollen grains and spores from across the plant kingdom further showed that microgel formation occurred with tested pollen species from eudicot plants while tested pollen species from non-eudicot plants did not form microgels.”

Pg. 7, lines 146-154

“We also tested pollen grains from Baccharis and Camellia plants, which belong to the same eudicot clade as sunflower plants (Fig. 4a and Supplementary Fig. 16). We discovered that pollen grains from tested eudicot plants are transformed into microgel particles (Fig. 4b, Supplementary Figs. 17-28 and Supplementary Video 11). Taken together, these results support that pollen grains from eudicot plants have unique material properties that facilitate microgel conversion using our nature-inspired strategy. The observed variation in gelling propensity among pollen from different plant species may also relate to structural variations in microcapsule architecture that give rise to distinct chemomechanical responses.”

[REDACTED]

Editorial Note: Figure redacted at authors request

(h) “Collectively, our experimental and computational results offer fundamental insights into how tuning pollen structure can cause dramatic alterations to material properties, elucidating how the materials science of pollen influences plant reproductive success.” It is not clear how the work presented in the manuscript relates to the “reproductive success” of plants.

Reply: We agree with the Reviewer and, in the revised manuscript, we revised this statement as follows (additions in yellow highlight):

Pg. 1-2, lines 29-32

“Collectively, our experimental and computational results offer fundamental insights into how tuning pollen structure can cause dramatic alterations to material properties, and inspire future investigation into understanding how the materials science of pollen might influence plant reproductive success.”

Pg. 8-9, lines 188-193

On the other hand, insufficient hydration of pollen can cease germination, leading to harmomegathy with minimal energy loss. Thus, our findings provide insight into the design principles of pollen grains in the context of harmomegathy (dehydration) and germination (hydration), while facilitating the development of a broadly effective strategy to prepare pollen-derived microgel particles with high-performance material properties that go beyond the natural synthetic capabilities of plant reproductive machinery.”

Reviewer 2 Comments

The authors reports a extensive study on production of a stimuli responsive (mono- and divalent cations, pH) sporopollenin microgels from various pollen grains. The manuscript is well-designed and it was written in a reader-friendly manner. The findings of the study was intersting and worth spreading. The discussions are relevant and very informative. The findings of the work should reach wider audience by publication.

I think that the submission can be accepted for publication in the present form.

My recommedation is ACCEPT for publication.

Reply: We thank the Reviewer for positive endorsement of our work and are grateful that they recommend publication of the work. We have carefully revised the manuscript according to the suggestions of Reviewer 1 too and hope that the revised manuscript is deemed worthy of publication and will hopefully reach a wide audience and inspire future readers to explore this exciting field.

References

- 1 Mundargi, R. C. *et al.* Extraction of sporopollenin exine capsules from sunflower pollen grains. *RSC Advances* **6**, 16533-16539 (2016).
- 2 Willats, W. G. *et al.* Analysis of pectic epitopes recognised by hybridoma and phage display monoclonal antibodies using defined oligosaccharides, polysaccharides, and enzymatic degradation. *Carbohydrate Research* **327**, 309-320 (2000).
- 3 Bárány, I., Fadón, B., Risueño, M. C. & Testillano, P. S. Cell wall components and pectin esterification levels as markers of proliferation and differentiation events during pollen development and pollen embryogenesis in *Capsicum annuum* L. *Journal of Experimental Botany* **61**, 1159-1175 (2010).
- 4 Knox, J. P., Linstead, P. J., King, J., Cooper, C. & Roberts, K. Pectin esterification is spatially regulated both within cell walls and between developing tissues of root apices. *Planta* **181**, 512-521 (1990).
- 5 Qu, Z. & Meredith, J. C. The atypically high modulus of pollen exine. *J. R. Soc., Interface* **15**, 20180533 (2018).
- 6 Zeng, Z. & Tan, J.-C. AFM Nanoindentation To Quantify Mechanical Properties of Nano- and Micron-Sized Crystals of a Metal–Organic Framework Material. *ACS Appl. Mater. Interfaces* **9**, 39839-39854 (2017).
- 7 Parre, E. & Geitmann, A. More Than a Leak Sealant. The Mechanical Properties of Callose in Pollen Tubes. *Plant Physiology* **137**, 274–286, doi:0.1104/pp.104.050773 (2005).
- 8 Qu, Z. & Meredith, J. C. The atypically high modulus of pollen exine. *J R Soc Interface* **15**, doi:10.1098/rsif.2018.0533 (2018).
- 9 Hutter, J. L. & Bechhoefer, J. Calibration of atomic-force microscope tips. *Review of Scientific Instruments* **64**, 1868-1873 (1993).
- 10 Payne, W. W. Observations of harmomegathy in pollen of Anthophyta. *Grana* **12**, 93-98 (1972).
- 11 Katifori, E., Alben, S., Cerda, E., Nelson, D. R. & Dumais, J. Foldable structures and the natural design of pollen grains. *Proc. Natl. Acad. Sci. U. S. A.* **107**, 7635-7639 (2010).
- 12 Hesse, M. Pollenkitt and viscin threads: their role in cementing pollen grains. *Grana* **20**, 145-152 (1981).
- 13 Heslop-Harrison, J. Pollen germination and pollen-tube growth in *International review of cytology*. Vol. **107** 1-78 (Elsevier, 1987).
- 14 Heslop-Harrison, Y. & Heslop-Harrison, J. The Microfibrillar Component of the Pollen Intine Some Structural Features. *Annals of Botany* **50**, 831-842 (1982).

Reviewers' Comments:

Reviewer #1:

Remarks to the Author:

I thank the authors for their careful response to my comments. They have responded to the key concerns I had. I would only ask that they rephrase the legend of Suppl. Fig. 19. For the bottom part of the figure, the legend states: "Bottom: Pollen species from algae and green plants showed minimal response to changes in solution pH." The statement is factually wrong and, again, goes far beyond what authors have demonstrated. The words "algae" and "green plants" should be changed since algae do not have pollen or spores, and "green plants" is meaningless. In the case of Lycopodium, spores were tested not pollen. Finally, the authors DO NOT have the data to justify the sweeping statement that all non-eudicots pollen and spores cannot form gels.

Jacques Dumais

Reviewer #3:

Remarks to the Author:

The authors induce pectin de-esterification in pollen grains to "mimic the activity of key enzymes involved in pollen tube growth". The authors conclude that "subtle, chemically programmed variations in the mechanical properties of both the exine and intine layers can cause the swelling of pollen particles akin to the shape transformations that occur during the orchestrated sequence of pollen hydration, germination, and tube growth". The authors observed that as a result of changes in the cell wall, and specifically pectin de-esterification, the cell wall swells by pH and ionic treatment, a process that was seen to be reversible.

While there is in principle nothing wrong with a purely curiosity driven approach, the authors do suggest that they 1. address a biological knowledge gap with their study, and 2. provide the basics for a possible application (micro-gel particles). In my eyes, the present manuscript does not provide satisfactory information for either, and it fails to even properly identify the knowledge gap, which leaves the pollen biologist wanting.

As for the biochemical aspect of the manuscript, pectin gel swelling is not entirely novel since changes in mechanical properties and swelling of cell walls upon changes in pectin esterification is already well known. I am not an expert in biochemistry, however, and may have overlooked important aspects that warrant this study to be of high profile with relation to the biochemistry.

Unfortunately, I haven't been convinced yet by the computational model approach either. The main reason is that the justification for the conditions used are inadequately presented and consequently I may simply not have understood (despite having experience in FE modeling). Concretely, the modeling approach and the boundary conditions used would require additional explanations to reassure me that the interpretations are of relevance to the biological system. This is crucial since the authors proceed to draw important conclusions from their finite element model such as the one stated in the beginning of page 8:

"...that the swelling of pollen particles occurs when the ratio of the mechanical strength of the chemically tuned exine and intine layers is within a certain, optimal range. This chemomechanical balance appears to be carefully regulated and suggests that there might be a chemorheological regulatory pathway at the individual particle level underpinning whether pollen hydration leads to successful germination or incipient drying".

In the following, detailed concerns are listed which the authors may wish to consider. (Annoyingly, the text didn't have line numbers and hence locations are approximate)

Major concerns:

1. Finite element model: Even though some similarities between experimental and finite element data are drawn through matching a number of data points, I am not convinced that the working physics of the phenomena in model and experiments are actually the same. If a direct relation exists, is not clear enough in the present version of the manuscript.

The study aims at investigating the contribution of the mechanical properties of the exine and intine layers of pollen grain in relation to the swelling and de-swelling of the grain. In the finite element model used to simulate the swelling process, the authors implement an internal pressure in the lumen of the grain to simulate pollen grain 'swelling', similar to inflating a balloon. The authors mention that "hydration and swelling of the hyperelastic pollen shells associated with osmotic pressure effects were simulated based on a hypothetical internal pressure". The pollen grain's increase in size in the model does therefore seem to occur due to a pressure-induced stretching of both enveloping layers, the intine and the exine, and which essentially results in an increase of the volume of the internal cavity. In the biological sample this would presumably be achieved through osmotically driven water uptake into the cytoplasm and/or vacuole. The chemical treatment that is used in the experiments, on the other hand, is not cited to change the internal turgor pressure or volume of the internal cavity, but to cause a swelling of the wall material composing the inner layer of the envelope, the intine. It is unclear how the approach of inflating the structure by increasing the pressure in the cavity simulates the process of wall swelling in the inner of the two enveloping layers. Shouldn't this chemical process lead to a volume increase in the intine?

To justify their modeling approach, the authors refer to a publication by van der Sman (2015, reference 47). However, it is not exactly clear to me how this works and more detailed explanation is needed. van der Sman considered the hydration of the cellular tissue from two different sources: the vacuole water resulting in the turgor and the water retained in the hydrated cell wall (van der Sman considered the hydrated shell to consist of both the cell wall and the cytoplasm). According to the paper by van der Sman, "a theoretical analysis of the contributions of both the turgor and the hydrated cell wall to the total water holding capacity of cellular tissue" was performed. In this publication, the turgor was not a "hypothetical" element presumed but a real pressure applied in the cell cavity due to osmotic pressure of the solutes in the vacuole (i.e., turgor). As van der Sman mentions, membrane integrity is required and results in water retention inside the cell (symplastic water) and turgor. In addition, in formulation of the stress in the shell, a term for osmotic pressure is also added to account for the hydration of the cell wall material (e.g., eq. 3 of van der Sman). van der Sman further concludes that "This implies that even for thin cell walls, with RWC E 5%, it will still be important to include the water retention in the apoplast for correct prediction of mechanical behavior". Unless I overlooked it, no such term accounting for water content of the wall material is mentioned in the present manuscript. Instead, the authors seem to base their model on an inflation of the spherical shell by a ("hypothetical") turgor and seem to suggest that this is effectively the same as swelling of the shell wall. I am unable to confirm that this can be concluded from van der Sman's paper, or that it reflects the physics of the present experiments (swelling of the wall material).

One source of confusion may be the use of terminology. On page 22 the authors write "To simplify simulations, we disregarded ... volume changes in the intine due to swelling effects. Instead, intine swelling was imposed by tensile circumferential stresses ($\sigma_{\theta\theta}$, $\sigma_{\varphi\varphi}$) due to the osmotic pressure across its thin gel-like matrix." In this two-sentence passage, the term 'swelling' seems to refer to two different processes. In the first sentence the authors seem to refer to the increase in volume of the intine material due to the molecular repulsion (as in Figure 2), whereas in the second sentence the 'swelling' seems to refer to the increase in size of the entire pollen grain due to pressure from the internal cavity and resulting tension of the pollen envelope. If I interpret this correctly, I urge the authors to use two different terms for the two different phenomena, for example 'inflation' or 'turgor-driven expansion' for the latter. I could have this wrong and better explanations (and a clearer image in Supp Figure 13d) would be helpful. Further, when the authors mention that "intine swelling was imposed by tensile circumferential stresses", it initially gives the

impression that these stresses were input as initial conditions in the model where in reality they seem to result from the stretch of the material due to turgor.

Crucially, the authors do not mention how the thickness of the wall is affected in this experimental process. As can be seen from Fig. 1 of van der Sman, the wall can get thinner when stretched by an internal turgor. It needs to be clarified how the changes in pollen wall thickness upon wall swelling are related between the model and the experiment and I strongly suggest that the authors add transmission electron micrographs of cross-sections of rapidly frozen and fixed pollen grains to demonstrate the actual volume and thickness change in the intine resulting from the chemical treatment.

Finally, given the authors' intention to correlate their finding with biological processes, a discussion of the actual germination process would be worthwhile (e.g. Heslop-Harrison Y, Heslop-Harrison J (1992) Germination of monocot angiosperm pollen: evolution of actin cytoskeleton and wall during hydration, activation and tube emergence. *Ann Bot* 69:385–394; Márquez J, Seoane-Camba JA, Suárez-Cervera M (1997) Allergenic and antigenic proteins released in the apertural sporoderm during the activation process in grass pollen grains. *Sex Plant Reprod* 10:269–278)

2. In definition of the finite element boundary conditions: Why wasn't Abaqus/Standard used instead of /Explicit? Abaqus/Explicit is generally reserved for addressing problems of highly dynamic nature (e.g., impact). Here, the swelling process can be safely regarded as a quasi-static process and could be modeled as such using a Static step in Abaqus/Standard. While Abaqus/Explicit can also be used to solve such problems, the choice should really be justified. The reason is that ensuring the correctness of the values and patterns resulting from Explicit Dynamics can be challenging. In the same vein, how does use of such small time steps (in order of microseconds) correspond to the quasi static process of swelling?

3. Page 3: How can FTIR confirm that 'structural integrity' of the wall is maintained? What does structural integrity mean in this context? That polymers remain polymers or that they maintain a given network structure (i.e. linkages)? What exactly does FTIR allow to conclude? Please provide a bit of information to the non-expert.

4. Page 4: "can result in the transformation of hard pollen grains into pliable, soft microgel particles". It would be helpful to define microgel particle. The authors seem to repeatedly suggest that, upon treatment, pollen grains from some species form "microgel" particles while others do not. No clear definition is provided what properties are required for a particle to count as a microgel particle. The pollen grains clearly still have a lumen and a stiff outer wall (exine). What makes them microgel particles that might be useful for applications in any manner?

5. Methods section, page 16: In the "microgel formation" section it is mentioned that the sample was left at 80°Celsius. Heat is known to alter the plant cell wall material. Did the authors verify how temperature affects their results?

Minor comments

6. Page 2, line 4: 'transfers viable cellular material' should be more specific. The transfer consists in the gametes, or sperm cells, from the male gametophyte (=pollen) to the female gametophyte.

7. Page 2, bottom half. 'hollow' does not seem to express that the pollen grain is filled by a liquid (not a gas). Maybe simply leave 'hollow' out of the sentence? Similarly, further down, 'holes' may leave the wrong impression of actual empty spaces which is not the case since an aperture still has an intine. I suggest replacing by 'gap in the exine'.

8. Page 3: "To address this challenge": what challenge? No challenge is specified as of yet except

for an allusion to potential of getting inspired by design of pollen grain.

9. Page 3, bottom half. 'Coinciding with an increase in gel-like properties...'. Until here, the manuscript has not mentioned any proof for gel-like properties to actually arise, hence it is somewhat misplaced to casually mention them here.

10. Page 4, first line: 'soft microgel'. Before this line, there has been no mention of or proof for the fact that the particles have become soft or pliable (although this is presented later in the manuscript). I am wondering whether reorganizing the narrative would help the reader to follow. "The aperture opening dissipates little energy, whereas stretching is an energy-consuming process.": not clear based on what this statement is made. Is it based on elastic strain energy variations?

11. Page 6: What is the "initial swelling ratio, λ_{open} "? Please define.

12. Page 7, third line: you may wish to add '... did not form microgel particles UNDER THE CONDITIONS USED HERE'.

13. Page 7: "we discovered that this distinction alone is insufficient to explain hydration-induced pollen swelling." This claim requires further discussion.

14. Page 7, last line: "mechanical strength" or stiffness?

15. Figure 1C and E: the legend must indicate which primary antibody was used. What is "hydrostatic tensile stress"? Is it principal stress due to hydrostatic pressure?

16. Page 19: Was AFM used or another indentation device? I am asking since the text mentions "instrumented indentation". This term is generally used for cantilever-less indentation systems.

17. What was the AFM tip shape?

18. Page 19: Was the immunolabel not preceded by a chemical fixation step? The protocol speaks of 'labeled sections' which would mean there would be a resin embedding step as well. There doesn't seem to be any mention of that (unless it is elsewhere in the methods). Please provide exhaustive description of the method.

19. Page 19: To compare fluorescence intensity between samples it is crucial that they were prepared in parallel, that the microscope settings were identical for acquisition and that image processing was done in identical manner. Please confirm that this was the case.

20. Legend of Figure 1 and elsewhere: Please specify that the SEM images are pseudo-colored.

21. Why does particle size decrease between 6-12 hours (Figure 1D)?

22. Figure 2: I realize that the drawings are conceptual and simplified. However, pectin is a highly branched molecule (rather than a single strand as indicated). Maybe this could be symbolized at least conceptually?

23. Figure 2a: The figure legend should specify whether the repulsion is made due to increase or decrease in pH.

24. Figure 2c: How is "area swelling ratio" determined?

25. Figure 2d: The legend is not descriptive of the events shown in figure.

26. Figure 3 and FEA model: Did the presence of spikes actually influence the modeling results? Did they serve any other purpose than looking prettier when put side-by-side with micrographs? (This is fine, but it should be mentioned somewhere whether or not they have an effect or are purely meant to make the model look similar to the biological sample).

27. Supplementary Figure 2: I understand that these are conceptual drawings, but filling the entire pollen grain lumen with 'genetic material' simply ignores the cellular structure. Not the entire protoplast consists of genetic material! To not change the drawing, the light green material would need to be identified as 'Protoplast of the vegetative cell and generative cell (or sperm cells - depending on species)'. To draw this more accurately, at least the nucleus of the vegetative cell as well as the generative cell or sperm cells should be drawn.

28. Supplementary Figure 3: Is this epifluorescence or confocal laser scanning microscopy? If the latter, are these optical sections or max projections? Please confirm that imaging was done under identical conditions for all samples.

29. Supplementary Figure 5: Please specify that the gray tone images are brightfield micrographs and identify the optics used (DIC?)

Response to Reviewers for NCOMMS-19-11769A-Z

Reviewer 1 Comments

I thank the authors for their careful response to my comments. They have responded to the key concerns I had.

Reply: We sincerely thank the Reviewer for taking the time and expert effort to carefully review our manuscript and for providing many excellent comments to help us revise and improve the manuscript. In the revised manuscript, we have added/edited the relevant remarks to fully incorporate the Reviewer's suggestions.

1) I would only ask that they rephrase the legend of Suppl. Fig. 19. For the bottom part of the figure, the legend states: "Bottom: Pollen species from algae and green plants showed minimal response to changes in solution pH." The statement is factually wrong and, again, goes far beyond what authors have demonstrated. The words "algae" and "green plants" should be changed since algae do not have pollen or spores, and "green plants" is meaningless. In the case of Lycopodium, spores were tested not pollen.

Reply: We thank the Reviewer for the constructive comment and agree that the figure legend should be edited. In the revised manuscript (**pg. 45, lines 1028-1030**), these points have been edited as follows:

"Bottom: Pollen and spore species from the tested non-eudicot plants maintained similar size and shape in different pH environments. Data are reported as mean \pm s.d. from $n = 500$ particles per condition. Source data are provided as a Source Data file."

2) Finally, the authors DO NOT have the data to justify the sweeping statement that all no-eudicots pollen and spores cannot form gels.

Reply: We appreciate the Reviewer's thoughtful comment. We added more constrained information on the related remarks as follows:

Pg. 1, lines 25-28

"Our investigation of pollen grains and spores from across the plant kingdom further showed that microgel formation occurs with tested pollen species from eudicot plants while tested pollen species from non-eudicot plants did not form microgels under the conditions used in this manuscript."

Pg. 7, lines 159-161

*"To extend our findings, we tested pollen grains and spores from other clades and discovered that those from flowering monocots (Cattail) and gymnosperms (Pine) as well as spore-bearing lycophytes (Lycopodium) did not form microgel particles (**Fig. 4**)."*

Pg. 7, lines 165-167

“Taken together, these results support that pollen grains from the tested eudicot plants have suitable material properties to facilitate microgel conversion using our nature-inspired strategy.”

Reviewer 3 Comments

The authors induce pectin de-esterification in pollen grains to “mimic the activity of key enzymes involved in pollen tube growth”. The authors conclude that “subtle, chemically programmed variations in the mechanical properties of both the exine and intine layers can cause the swelling of pollen particles akin to the shape transformations that occur during the orchestrated sequence of pollen hydration, germination, and tube growth”. The authors observed that as a result of changes in the cell wall, and specifically pectin de-esterification, the cell wall swells by pH and ionic treatment, a process that was seen to be reversible.

Reply: We thank the Reviewer for taking the time to carefully review our manuscript and for providing many excellent comments to help us revise and improve the manuscript. Upon reading the Reviewer’s comments, we identified one important point that could have led to misunderstanding of some of the key points in our manuscript in general. This possibility appears to have resulted from a fundamental misunderstanding of plant cell walls versus pollen shells, which represent distinct features of two very different systems.

Below, we provide detailed responses to each point raised by Reviewer #3 and explain how we have improved the revised manuscript in our attempt to respond to each of his/her comments. Where applicable, we have also provided additional Review-Only Material (ROM) to explain our approach and findings.

While there is in principle nothing wrong with a purely curiosity driven approach, the authors do suggest that they 1) address a biological knowledge gap with their study, and 2) provide the basics for a possible application (micro-gel particles). In my eyes, the present manuscript does not provide satisfactory information for either, and it fails to even properly identify the knowledge gap, which leaves the pollen biologist wanting.

Reply: We appreciate the Reviewer’s remark and concern. As addressed in the abstract, the biological knowledge gap that we address in the current manuscript is focused on how tuning pollen structure (through a facile chemical process) can alter pollen’s mechanical, chemical, and adhesion properties. The ability of pollen to control its microstructure depending on the environment during harmomegathy, reverse-harmomegathy, or germination¹, is guided by an organized sequence of enzymatically controlled reactions to modify structural components within certain parts of the pollen wall^{2,3}. Although this enzymatic process might be a better-known phenomenon (albeit still the subject of intense, ongoing investigations in labs worldwide) from a biological perspective, engineering pollen-inspired materials by recourse to and expanding on these remodeling processes is highly novel and results in high-performance material capabilities that exceed natural limits, as identified in this manuscript for the first time.

To better explain the significance of our work, we have revised the introduction of manuscript as follows:

Pg. 2, lines 43-55

*“Common features of pollen grains across various plant species include a microcapsule structure, function-driven shape, and ornamental architecture¹⁵. We selected pollen grains from sunflower plants (*Helianthus annuus*), which have spiky appendages and a tripartite structure (**Fig. 1a** and **Supplementary Fig. 1**). The outermost layer (“exine”) is made up of sporopollenin, which is a strong, crosslinked biopolymer¹⁰, while the inner layer (“intine”) is composed of elastic, load-bearing cellulose/hemicellulose microfibrils and pectin^{16,17} (**Fig. 1b**). The aperture gap in the exine layer is integral to pollen tube growth and is neighbored by pollen wall regions with distinct material properties. As part of remodeling processes, pectin methylesterase (PME) enzyme plays a key role in controlling wall elasticity by converting pectin into pectate¹⁸, exposing carboxylic acid functional groups and imparting greater surface charge density that modulates the intine structural arrangement¹⁸. Biologically, this enzymatic activity is spatially controlled within the pollen wall structure^{19,20}, and inspires biomimetic strategies to engineer pollen structures with tunable material properties.”*

We have carefully examined the existing references and our approach to better explain the concerns of the referee. As for the possible application of microgel particles, we have now included detailed information about our ongoing research in our reply to the Reviewer’s major comment 4, and also updated appropriate sections in our revised manuscript.

As for the biochemical aspect of the manuscript, pectin gel swelling is not entirely novel since changes in mechanical properties and swelling of cell walls upon changes in pectin esterification is already well known. I am not an expert in biochemistry, however, and may have overlooked important aspects that warrant this study to be of high profile with relation to the biochemistry.

Reply: We would like to clarify here how our work is distinct from plant cell walls that are typically studied by plant tissue swelling and what is significant about our work.

Plant cells always have a strong cell wall surrounding them. When they take up water by osmosis, they start to swell but the cell wall prevents them from bursting. Plant cells become “turgid” when they are put in dilute solutions. Turgid means swollen and hard. The pressure inside the cell rises, and eventually the internal pressure of the cell is so high that no more water can enter the cell. This liquid or hydrostatic pressure works against osmosis. Turgidity is very important to plants because this is what make the green parts of the plant “stand up” in the sunlight.

As noted above, the plant cell wall prevents plant cells from bursting due to osmosis-induced swelling. In the case of animal cells which are not surrounded by tough cell walls, they swell and eventually burst under osmotic pressure. The swelling of this tough cell wall, accommodated by pectin gels, is one of the mechanical strategies to reduce the

stress concentration throughout the cell wall by softening the cell wall for improved flexibility and improved resilience.

By contrast, the pollen wall performs two major roles in order to maximize the success rate of reproduction. First, it should protect the generative or sperm cells in various environmental conditions, until the right stigma environments are settled. Thus, the hydrophobic exine layer blocks water penetration into the pollen wall, going through harmomegathy. Second, the pollen wall should accommodate germination which requires significant water intake and tubular growth to facilitate the reproduction process. Due to the stiff and hydrophobic exine layer, water intake is significantly constrained at the beginning. As a result, mechanical and chemical degradation of the exine by releasing lipid and pollen cements associated with increased porosity should be a key change occurring during germination. In the meantime, pectin gel of the intine layer increases the inflation capability of the pollen grain, allowing significant water intake and rapid tubular growth. Thus, in fact, both exine and intine layers play important roles in this germination process. In our study, we mimicked this germination process, inducing chemical and structural changes of the exine and intine layers, for transformation of pollen grains into inflatable soft microgels.

First, we would like to address that the plant cell wall from tissue and plant pollen are fundamentally different, and thus the biological implication of revealing each pectin-involving mechanism is also very different. The composition of plant cell walls mainly consists of cellulose, pectic and hemicellulosic derivatives, which are similar to pollen's inner layer, intine. However, pollen includes an inner intine layer as well as the outer exine layer, which is structurally much tougher and more complex than any plant cell wall. Also, plant cell walls form tissues, whereas pollen particles exist discretely (**ROM Fig. 1**). Along this line, broadly studied topics cover the growth of the plant cell wall or softening of ripe fruit mediated by pectin, which is directly relevant to the life cycle of plant tissue⁴.

ROM Fig. 1. Difference between plant cell wall and pollen. **a**, Top: Fluorescence image of root of *Arabidopsis thaliana* with green fluorescent protein labeling cell membrane and red fluorescent protein marking nuclei. Bottom: Cellular structure of plant cell wall. **b**, Top: Fluorescence image of *Cucurbita* pollen with autofluorescence. Bottom: KOH-treated pollen particle structure in order to show the pollen shell.

Second, the definition of ‘gel’ that the Reviewer is referring to and ‘microgel’ from our manuscript are conceptually different. A general term of ‘pectin gel’ refers to the high-methoxy pectin under acidic condition or low-methoxy pectin forming egg-box model, whereby divalent ions crosslinks or bridges the polysaccharide chains, thus showing the increased viscosity^{5,6}. These types of gel are well-known as pectin is widely used in food as a gelling agent or stabilizer. However, ‘microgel’ refers to a gel formed from a network of microscopic polymer or macromolecular colloids that swell in a good solvent, whereby sometimes feature stimuli (*e.g.*, temperature, pH, ions)-responsive behaviors^{7,8}.

To the best of our knowledge, the current manuscript is the first paper reporting that the chemical treatment controls the de-esterification of pectin in order to produce a soft material including the exine layer.

Unfortunately, I haven't been convinced yet by the computational model approach either. The main reason is that the justification for the conditions used are inadequately presented and consequently I may simply not have understood (despite having experience in FE modeling). Concretely, the modeling approach and the boundary conditions used would require additional explanations to reassure me that the interpretations are of relevance to the biological system. This is crucial

since the authors proceed to draw important conclusions from their finite element model such as the one stated in the beginning of page 8:

“...that the swelling of pollen particles occurs when the ratio of the mechanical strength of the chemically tuned exine and intine layers is within a certain, optimal range. This chemomechanical balance appears to be carefully regulated and suggests that there might be a chemorheological regulatory pathway at the individual particle level underpinning whether pollen hydration leads to successful germination or incipient drying”.

In the following, detailed concerns are listed which the authors may wish to consider. (Annoyingly, the text didn't have line numbers and hence locations are approximate).

Reply: We appreciate your attention to our computational model approach and are able to confidently answer and clarify the following points.

Major concerns:

1) **Finite element model:** Even though some similarities between experimental and finite element data are drawn through matching a number of data points, I am not convinced that the working physics of the phenomena in model and experiments are actually the same. If a direct relation exists, is not clear enough in the present version of the manuscript. The study aims at investigating the contribution of the mechanical properties of the exine and intine layers of pollen grain in relation to the swelling and de-swelling of the grain. In the finite element model used to simulate the swelling process, the authors implement an internal pressure in the lumen of the grain to simulate pollen grain 'swelling', similar to inflating a balloon. The authors mention that “hydration and swelling of the hyperelastic pollen shells associated with osmotic pressure effects were simulated based on a hypothetical internal pressure”. The pollen grain's increase in size in the model does therefore seem to occur due to a pressure-induced stretching of both enveloping layers, the intine and the exine, and which essentially results in an increase of the volume of the internal cavity. In the biological sample this would presumably be achieved through osmotically driven water uptake into the cytoplasm and/or vacuole. The chemical treatment that is used in the experiments, on the other hand, is not cited to change the internal turgor pressure or volume of the internal cavity, but to cause a swelling of the wall material composing the inner layer of the envelope, the intine. It is unclear how the approach of inflating the structure by increasing the pressure in the cavity simulates the process of wall swelling in the inner of the two enveloping layers. Shouldn't this chemical process lead to a volume increase in the intine?

To justify their modeling approach, the authors refer to a publication by van der Sman (2015, reference 47). However, it is not exactly clear to me how this works and more detailed explanation is needed. van der Sman considered the hydration of the cellular tissue from two different sources: the vacuole water resulting in the turgor and the water retained in the hydrated cell wall (van der Sman considered the hydrated shell to consist of both the cell wall and the cytoplasm). According to the paper by van der Sman, “a theoretical analysis of the contributions of both the

turgor and the hydrated cell wall to the total water holding capacity of cellular tissue” was performed. In this publication, the turgor was not a “hypothetical” element presumed but a real pressure applied in the cell cavity due to osmotic pressure of the solutes in the vacuole (i.e., turgor). As van der Sman mentions, membrane integrity is required and results in water retention inside the cell (symplastic water) and turgor. In addition, in formulation of the stress in the shell, a term for osmotic pressure is also added to account for the hydration of the cell wall material (e.g., eq. 3 of van der Sman). van der Sman further concludes that “This implies that even for thin cell walls, with RWC E 5%, it will still be important to include the water retention in the apoplast for correct prediction of mechanical behavior”. Unless I overlooked it, no such term accounting for water content of the wall material is mentioned in the present manuscript. Instead, the authors seem to base their model on an inflation of the spherical shell by a (“hypothetical”) turgor and seem to suggest that this is effectively the same as swelling of the shell wall. I am unable to confirm that this can be concluded from van der Sman’s paper, or that it reflects the physics of the present experiments (swelling of the wall material).

Reply: We sincerely appreciate the Reviewer’s comments and the opportunity to explain our results here in greater detail. In this work, chemomechanical boundary conditions of particle swelling were reasonably simplified through a conventional solid mechanics approach, with continuum FE models that simulate the hydrogel-like intine layer as hyper-elastic solid. This approach has been well-accepted for modelling of smart hydrogel systems (e.g, hydraulic hydrogel actuators or natural seed capsule actuator from *Delosperma nakurense*) in a simpler manner^{9,10}. Furthermore, the purpose of our modelling in the current manuscript is to understand how the interplay of exine and intine layers influences inflation of pollen microgels, mimicking germination of pollen grains. Thus, we would like to clarify in the following response: 1) what happens to pollen grains after chemical treatment, 2) how pollen microgels are inflated, and 3) then how we mimicked the inflation of pollen microgels using the continuum FE models, ignoring hydration and volume change of the intine layer.

- 1) For pollen grains, the intine layer is originally hydrophobic due to highly esterified pectin, whereas the exine layer consists of sporopollenin covered or filled with lipid and pollenkitt (pollen cement). Thus, the pollen shell is inherently hydrophobic and not swellable when it is intact.
- 2) After the chemical treatment, the highly esterified pectin becomes de-esterified and hydrophilic while the exine layer becomes porous, losing pollen cement and lipid. In the meantime, the cytoplasm is also fully removed, thus the internal cavity is fully empty (**ROM Fig. 2**). Also, since the cytoplasm with cells was released through three apertures, the continuous intine layer was ruptured. Thus the bi-layered pollen shell was fully discretized through the entire shell thickness at the apertures.

ROM Fig. 2. Schematic illustration of pollen grains and pollen microgels before/after chemical treatment.

- 3) As a result, a pollen microgel has a hollow shell structure with three apertures, consisting of the outer exine and inner intine hydrogel layers. Thus, the volume increase of this pollen microgel system is caused by the swollen intine layer interplaying with the relatively stiffer exine. Moreover, three apertures are open to the environment, allowing rapid water intake into internal cavity of the hollow microgel shell due to the inflation of the pollen shell (**ROM Fig. 3**).

ROM Fig. 3. Schematic illustration of a pollen microgel before and after inflation from pH 2 to pH 7

- 4) In this study, we aimed to understand how the hollow pollen shell, consisting of stiff exine layer and swellaible intine layer, could deform under pH change depending on the duration of the second KOH treatment (from 0 h to 12 h). Since we used a conventional solid mechanics approach using nonlinear continuum FE models, chemical boundary conditions (e.g., solute concentrations, water contents, charge balance, mass transfer, etc.) in the real case require assumptions in order to make this chemical/mechanical coupling problem solvable with reasonably simplified mechanical boundary conditions (e.g., internal pressure) as shown in **Supplementary Figure 13**,
 - **Real case:** since the intine layer is swellaible depending on its osmotic pressure induced by ionization or de-ionization of pectin chains in response to pH changes, the induced tensile circumferential stresses ($\sigma_{\theta\theta}$, $\sigma_{\phi\phi}$) of the swollen intine layer with a resultant volume change of the intine layer inflate the pollen microgel. Subsequently, the increased cavity volume of the pollen

microgel is filled with water coming from the opened three apertures very rapidly to reach equilibrium. However, in this case, typical continuum FE models cannot deal with those chemical boundary conditions for intine where hydration and swelling occur through mass transfer under driving force by induced chemical potential depending on charge balance and solute concentration.

- **Equivalent model case:** instead of setting the chemical boundary conditions associated with swelling of the intine layer, we introduced the hypothetical hydrostatic pressure which can be directly applied to the internal surface of the intine layer. By using this hypothetical internal pressure, we could capture the inflation of pollen microgel using our continuum FE models. In order to make this boundary condition reasonable, we assumed that the deformation of pollen microgel is significantly large ($\gg 10\%$) and highly non-linear due to hyperelastic behavior and non-linear pollen geometry, ignoring hydration and volume increase of the intine layer associated with swelling. Thus, our models could predict the inflation of pollen microgels, showing good agreement with experimental observation.

Supplementary Figure 13 | Geometrical configuration of pollen shells used in the simulations, including an inner intine shell, outer exine shell, and spikes. d, Schematic illustration of pollen shell expansion that is induced by an osmotic pressure (π_{mix}) associated with hydration and swelling of the intine shell (top) and an equivalent internal pressure ($-p_{int}$) that is applied to the internal surface of the pollen shell (bottom). The inflation of the hyperelastic pollen shells due to hydrated and swollen intine was simulated using a hypothetical internal pressure, as proposed by van der Sman⁴⁷.

- 5) As the reviewer pointed out, van der Sman investigated the inflation and hydration of the elastic hydrogel shells due to internal pressure, p_{int} (van der Sman *et al.* 2015, ¹¹). In this context, we referred to a publication by his work on the internal osmotic pressure which induces the inflated hydrogel shell. However, since the pollen microgel system is a hollow hydrogel shell, it is significantly different from a typical plant cell fully enclosed with the cell wall and is different from the case modeled by van der Sman. Those cellular tissues can swell by increasing the osmotic pressure along with increase of solute content of the vacuole. On the other hand, our pollen microgel system can be inflated by pH responsive swelling behavior of the intine layer.
- 6) Also, in our follow-up work, we have been developing multi-phase models of the intine layers in conjunction with FE for a more accurate constitutive law of pollen microgels, considering hydration and swelling of pollen microgel in much more detail. We will address hydration, osmotic pressure, ionic concentration effects, etc. in our future works extensively, mainly focusing the improved modelling of pollen microgels.

One source of confusion may be the use of terminology. On page 22 the authors write "To simplify simulations, we disregarded ... volume changes in the intine due to swelling effects. Instead, intine swelling was imposed by tensile circumferential stresses ($\sigma_{\theta\theta}$, $\sigma_{\phi\phi}$) due to the osmotic pressure across its thin gel-like matrix." In this two-sentence passage, the term 'swelling' seems to refer to two different processes. In the first sentence the authors seem to refer to the increase in volume of the intine material due to the molecular repulsion (as in Figure 2), whereas in the second sentence the 'swelling' seems to refer to the increase in size of the entire pollen grain due to pressure from the internal cavity and resulting tension of the pollen envelope. If I interpret this correctly, I urge the authors to use two different terms for the two different phenomena, for example 'inflation' or 'turgor-driven expansion' for the latter. I could have this wrong and better explanations (and a clearer image in Supp Figure 13d) would be helpful. Further, when the authors mention that "intine swelling was imposed by tensile circumferential stresses", it initially gives the impression that these stresses were input as initial conditions in the model where in reality they seem to result from the stretch of the material due to turgor.

Reply: We agree that we used the term, "swelling", for both the intine layer and the microgel system, which caused the confusion. As the Reviewer pointed out, the term, "swelling" in the first sentence indicate the volume increase of intine due to osmotic pressure, whereas the swelling in the second sentence is the inflation of the pollen microgel as a result of swollen intine. Moreover, we agree that the sentence, "intine swelling was imposed by tensile circumferential stresses", caused some confusion about the initial boundary conditions of our models. We have updated those in the method section of our revised manuscript as follows:

Pg. 22, lines 627-637

"Boundary conditions for modeling pollen swelling/de-swelling To investigate pollen swelling and de-swelling behaviors as well as related deformation mechanisms, we employed a three-dimensional model based on finite element analysis (FEA) using a

*commercial software package ABAQUS 2017 (Dassault Systèmes SIMULIA, Johnston, RI). The model was generated by Python scripts and then run in parallel in 32 cores. To simplify simulations, we disregarded solvent flow into and out of the pollen shell walls during swelling, as well as volumetric changes in the intine due to swelling effects. Instead, intine swelling was imposed by tensile circumferential stresses ($\sigma_{\theta\theta}$, $\sigma_{\varphi\varphi}$) due to the osmotic pressure exerted across its thin gel-like matrix. Hydration and swelling of the hyperelastic pollen shells associated with osmotic pressure effects were simulated based on a hypothetical internal pressure in our FE models, as illustrated in **Supplementary Fig. 13d**.”*

Crucially, the authors do not mention how the thickness of the wall is affected in this experimental process. As can be seen from Fig. 1 of van der Sman, the wall can get thinner when stretched by an internal turgor. It needs to be clarified how the changes in pollen wall thickness upon wall swelling are related between the model and the experiment and I strongly suggest that the authors add transmission electron micrographs of cross-sections of rapidly frozen and fixed pollen grains to demonstrate the actually volume and thickness change in the intine resulting from the chemical treatment.

Reply: We appreciate the Reviewer’s suggestion. We observed the thickness of the exine and intine layers in dry condition after the chemical treatment using scanning electron microscope. Since pollen cement was removed from the exine layer, the reduced thickness and increased nanopores were clearly observed as the chemical treatment was prolonged (**Supplementary Figure 10**). In the meantime, through optical microscope, we confirmed that the intine layer was significantly hydrated and swollen in wet condition, increasing its total thickness. We fully agree that the precise thickness measurement of the hydrated intine layer can be done by transmission electron microscope as the reviewer suggested. However, this method allows us to observe the local thickness of a selected pollen microgel fragment in 2D. Since biological samples have some degree of structural inhomogeneity, we should do the thickness measurement more carefully.

Instead, we are working on nanoCT imaging of pollen microgels, under various pH and ionic conditions. Through this experiment, we hope we can clarify the real wall thickness of the hollow pollen shell in 3D. As the reviewer pointed out, we expect that we can capture the swelling behavior and underlying fundamentals of pollen microgels in a better way through this more accurate structural observation.

Moreover, during through our FE simulations, we also considered the thickness effect of the exine and intine layers as parametric simulations. The deformation mechanism was essentially independent of the thickness of the intine layer (e.g., for 1x, 2x and 4x intine thickness), whereas the magnitude of applied hypothetical internal pressure was increased to achieve the same level of inflation of pollen microgels with significantly increased simulation time, due to more prominent bulk behavior of the intine layer along with the increased thickness. In reality, along with swelling, the stiffness of hydrogel should be significantly reduced due to the increased water content within the hydrogel. Our new multi-phase models will capture the thickness effect of the intine layer in this context more accurately.

Finally, given the authors' intention to correlate their finding with biological processes, a discussion of the actual germination process would be worthwhile (e.g. Heslop-Harrison Y, Heslop-Harrison J (1992) Germination of monocolpate angiosperm pollen: evolution of actin cytoskeleton and wall during hydration, activation and tube emergence. *Ann Bot* 69:385–394; Ma´rquez J, Seoane-Camba JA, Sua´rez-Cervera M (1997) Allergenic and antigenic proteins released in the apertural sporoderm during the activation process in grass pollen grains. *Sex Plant Reprod* 10:269–278)

Reply: We thank the Reviewer’s attention to these references. As the Reviewer pointed out, we found that our processed pollen microgel samples exhibit key similarities to germinated pollen grains as follows:

- 1) During this germination process, lipid components from the outer layer of pollen grains are gradually dispersed, allowing increased water uptake^{12,13}. Through this process, apertures are opened, which accelerates water uptake, leading to germination with significant swelling of pollen grains¹³. Meanwhile, in the exine layer, pollen cements which occupy the cavity of sporopollenin network in exine are continuously released.
- 2) Since pollen grains are already open to the external environment due to the defatting process and pollen shell extraction, the hydrated intine can facilitate hydration of pollen grains even though the exine is still rigid. In this case, the exine only allows aperture opening or unfolding of pollen with minimal volume change (e.g., 0 h KOH-treated pollen). The swelling of intine or water uptake of pollen is significantly constrained by the rigid exine layer. Thus, germination is efficiently ceased or suspended until pollen allows more hydration from the environment.

Supplementary Figure 29 | Pollen hydration/dehydration cycle of angiosperms with corresponding FEA simulation images. The top sequence shows a dry pollen grain undergoing the water uptake process during germination. In situations where complete hydration is not possible, germination might be aborted, and the apertures close again as shown in the bottom sequence¹³.

- 3) Through the second KOH treatment, the exine layer exhibited significant structural changes, becoming more porous in the following figure. During germination, the exine also releases pollen cement or lipid components, increasing cavity size and hydrophilicity in order to accommodate fast water uptake¹³. Moreover, through the AFM measurements, we confirmed that the exine layer became less stiff.

Supplementary Figure 10 | Scanning electron microscope (SEM) images of defatted sunflower pollen grains and pollen shells before and after 10 w/v% KOH treatment. Top: SEM image of entire pollen microgels after defatting without or followed by 10 w/v% KOH incubation (2nd KOH treatment) for 0–12 h. Longer KOH treatment resulted in greater opening of the apertures. Bottom: Surface morphology of exine layers at higher magnification. Exines of the defatted pollen grain exhibited a dense and smooth surface morphology with a few microscale pores around the spikes, while the exine surfaces of KOH-treated pollen were rough and porous with exposed sporopollenin skeleton. The increased porosity of the exine surface was attributed to the release of pollen cement (“pollenkitt”) due to KOH treatment^{13,28}.

We also have added the information into the revised manuscript as follows:

Pg. 9, lines 199-209

“Understanding and tuning the biomechanics of natural materials holds promise for materials design and application. These findings are relevant to understanding how the morphological evolution of microgel particles might play a role in ensuring plant reproduction in nature¹³. The ease of aperture opening in pollen particles induces rapid water uptake from the stigma surface. Meanwhile, the subsequent stretching and expansion of pollen grains require significant water uptake and considerable structural change with high energy consumption, to promote germination. As a result, insufficient hydration of pollen due to the inappropriate stigma conditions ceases germination, leading to harmomegathy with minimum energy loss (Fig. S29). Thus, our findings provide insight into the design principles of pollen grains in the context of harmomegathy (dehydration) and germination (hydration), while explaining how our microgel formation and tuning approach can exceed the performance limits of these natural processes.”

2) In definition of the finite element boundary conditions: Why wasn't Abaqus/Standard used instead of /Explicit? Abaqus/Explicit is generally reserved for addressing problems of highly dynamic nature (e.g., impact). Here, the swelling process can be safely regarded as a quasi-static process and could be modeled as such using a Static step in Abaqus/Standard. While Abaqus/Explicit can also be used to solve such problems, the choice should really be justified. The reason is that ensuring the correctness of the values and patterns resulting from Explicit Dynamics can be challenging. In the same vein, how does use of such small time steps (in order of microseconds) correspond to the quasi static process of swelling?

Reply: We sincerely appreciate the Reviewer's great expertise in the FE modeling and fully agree with the Reviewer. Abaqus/Standard should be used for a static or quasi-static process since it provides more reliable solutions. As the reviewer pointed out, the swelling process of pollen microgel happened experimentally within a few seconds, implying that it definitely should be a quasi-static process rather than a dynamic process. However, due to the following reasons, we concluded that Abaqus/explicit should be more proper for our simulations than Abaqus/Standard.

1. We started our simulations using Abaqus/Standard. Unfortunately, the simulation failed when the swelling ratio of pollen microgel reached up to ~ 1.4 , where local deformation exceeded $\epsilon > 2$. We realized that even if our pollen microgel swelling is a quasi-static process, it induces significantly large deformation ($\epsilon > 2$) due to both material and geometric nonlinearity, particularly around the aperture regions. Therefore, mesh refining around the apertures was essential, unreasonably increasing the total number of elements and consequently extending the total simulation time up to a few days. Since the maximum swelling ratio of pollen microgel was ~ 1.8 , we concluded that Abaqus/Standard was not feasible to deal with this kind of large and non-linear deformation⁹. If we want to further model the swelling ratio > 1.4 using Abaqus/ Standard, we need to add a considerably greater number of elements around apertures to avoid convergence problems, which should be beyond our current computing limit as well as Abaqus's capability.
2. The quasi-static solution by Abaqus/explicit can improve the convergence of the model with a large deformation. Also, the quasi-static analysis with Abaqus/explicit is often used to solve large models and/or complicated contact problem which require fewer system resources than the implicit Abaqus/standards procedure¹⁴⁻¹⁶. Thus, we chose Abaqus/Explicit to solve our models.
3. As the reviewer commented, the quasi-static solutions by the Abaqus/Explicit should be carefully evaluated in terms of their correctness. The difference between the fully static solution (Abaqus/Standard) and quasi-static solution (Abaqus/Explicit) originates from kinematic effects.

ROM Fig. 4. Kinetic and strain energy density evolution of pollen microgel swelling as a function of swelling time using Abaqus/Explicit

As shown in **ROM Fig. 4**, we carefully monitored the kinematic energy density as well as strain energy density of pollen microgel during its swelling simulation. We clearly found that the kinematic energy occupies a very small portion compared with the strain energy, indicating the external work was mainly transformed into the static deformation strain energy of pollens. Thus, we ensured that the obtained quasi-static solutions were reasonable.

- Regarding the time steps, it is a pseudo time rather than the real physical time of the swelling event. Although in Abaqus/Explicit a longer step time reduces the kinematic energy, making quasi-static solutions closer to static, it requires too considerable computing resource. Hence, we carefully chose the step time of our simulations in times of the order of few microseconds, considering our large models (a total of 145,938 elements). As shown in **ROM Fig. 4**, with this microscale time step, the kinematic effects were almost negligible. Thus, we confirmed that we firmly obtained reasonable quasi-static solutions using Abaqus/Explicit under feasible model set-up conditions.

3) Page 3: How can FTIR confirm that 'structural integrity' of the wall is maintained? What does structural integrity mean in this context? That polymers remain polymers or that they maintain a given network structure (i.e. linkages)? What exactly does FTIR allow to conclude? Please provide a bit of information to the non-expert.

Reply: We agree with the Reviewer that the structural integrity might be misleading. In FTIR analysis, the structural characteristics of pollens depending on the treatment conditions were investigated. First, due to the existence of cytoplasm, the FTIR absorption spectrum of defatted pollen was more complicated and convoluted than those of KOH-treated pollen specimens. On the other hand, regardless of the different treatment time (0–12 h), the characteristic absorbance peaks of all KOH-treated pollen

grains appeared almost identical. Therefore, we presumed that one of the key components, pectin should be a distinctive feature through the FTIR spectra, depending on its de-esterification. The two important characteristic peaks of pectin and de-esterified pectin are ~ 1740 and $\sim 1620\text{ cm}^{-1}$ due to C=O stretching of esterified carboxyl groups ($-\text{COOCH}_3$) and free carboxyl groups ($-\text{COOH}$), respectively, along with the increase in $\sim 1410\text{ cm}^{-1}$ peak when the pectin is de-esterified^{17-19, 20}. The peak shift may occur due to the existence of potassium ions and the influence of other cell wall components²¹⁻²³. All pectin peaks were more clearly observed for KOH-treated pollen grains than defatted pollen, especially the peak at $\sim 1620\text{ cm}^{-1}$ for defatted pollen was not prominent. However, up to 12 h incubation with KOH, any significant changes of chemical structures of pollen grains weren't observed through FT-IR even though the existence of de-esterified carboxyl group for pectin was clearly confirmed. In order to make this point clearer, we edited the FTIR description in the revised manuscript as follows:

Pg. 4, lines 80-84

“Fourier transform infrared (FTIR) spectroscopy experiments also verified that all KOH-treated pollen grains appear nearly identical, irrespectively of the treatment time, whereas only the defatted sample exhibited more complex and convoluted spectral features along with peaks corresponding to pectin molecules due to residual cytoplasmic contents (Supplementary Fig. 4).”

4) Page 4: “can result in the transformation of hard pollen grains into pliable, soft microgel particles”. It would be helpful to define microgel particle. The authors seem to repeatedly suggest that, upon treatment, pollen grains from some species form “microgel” particles while others do not. No clear definition is provided what properties are required for a particle to count as a microgel particle. The pollen grains clearly still have a lumen and a stiff outer wall (exine). What makes them microgel particles that might be useful for applications in any manner?

Reply: We thank the Reviewer for this excellent question, and agree that the definition of microgel particles should be stated more clearly in the manuscript. A microgel is a gel formed from a network of microscopic polymer or macromolecular colloids that swell in a good solvent, whereby sometimes feature stimuli (*e.g.*, temperature, pH, ions)-responsive behaviors^{7,8}. In our study, the KOH-treated, tested pollen species from eudicot plants formed gel (as observed in the reversed vials in **Supplementary Figure 3c**) and were highly responsive to pH changes, which is the similar attribute to conventional microgels, although the unique morphology of the pollen grains remained after KOH treatment.

As the pollen grains are known to be practically indestructible along with demonstrating excellent biocompatibility, these microgel particles can be applied to various conventional material processes. Those individual microgel particles can be used as smart drug carriers which are responsive to pH or ion strength changes. More importantly, those soft pollen microgel particles can be processed for fabrication of thin films or sponges as shown in **ROM Fig. 5**. As the pollen microgels are responsive to external stimuli including pH and ionic strength, we believe these systems can be utilized as biosensors, purification as chelating agents or smart tissue scaffolds with tunable porous

structures. We are extensively expanding our fundamental research on pollen microgel systems into various applications as our future works. Moreover, natural polymer-based microgels are not thermally stable and have irregular particle shapes and heterogeneous particle sizes²⁴⁻²⁶. This limitation hampers the utility of existing natural printing materials used for freeform 3D printing. As the pollen microgel particles are uniform, thermally and mechanically stable, and stimuli-responsive material with strong material interfaces, they can be excellent source of natural materials for 3D/4D printing.

[REDACTED]

ROM Fig. 5. Our preliminary studies supporting the competitive advantages of plant pollen as a next-generation material. We have completed proof-of-concept experiments demonstrating that pollen-based microparticles from certain plant species are useful materials for various 3D printing strategies, paper, and sponge.

We have indicated this information in the revised manuscript as follows:

Pg. 9, lines 210-213

“Furthermore, the process offers new pathways to produce highly uniform microgel particles from pollen sources. Such particles could be useful in a wide range of applications where excellent quality control is essential as, for example, in high-performance sensors and actuators as well as in vivo drug delivery.”

5) Methods section, page 16: In the “microgel formation” section it is mentioned that the sample was left at 80 degree Celsius. Heat is known to alter the plant cell wall material. Did the authors verify how temperature affects their results?

Reply: We thank the Reviewer for this excellent question and have studied the effect of the temperature during the KOH incubation stages. Indeed, the temperature is one of the important factors for the microgel formation. In this study, we intentionally induce a biomimetic germination process by 1) removing pollen cements from the exine layer and 2) de-esterify pectin molecules of the intine layer in order to hydrate and inflate hydrophobic pollen grains. In particular, to de-esterify pectin molecules, high temperature (>50 °C) and strong alkali conditions are often required. We also found that the processing time were extended up to a few weeks at lower temperature (such as 25 °C) to obtain the same swelling behavior of produced pollen microgel particles. We also found that high temperature degraded pectin significantly after 24 h treatment, when pollen microgels couldn't maintain their structures with significantly reduced mechanical stability. Thus, we limited the processing time up to 12 h.

Minor comments:

6. Page 2, line 4: 'transfers viable cellular material' should be more specific. The transfer consists in the gametes, or sperm cells, from the male gametophyte (=pollen) to the female gametophyte.

Reply: In the revised manuscript, the sentence is revised to be more specific as follows:

Pg. 2, lines 33-35

“Pollen is a remarkable natural material that plays a critical role in plant reproduction and transfers viable cellular material (i.e., male gametes or sperm cells) between different reproductive parts of plants^{6,7}.”

7. Page 2, bottom half. 'hollow' does not seem to express that the pollen grain is filled by a liquid (not a gas). Maybe simply leave 'hollow' out of the sentence? Similarly, further down, 'holes' may leave the wrong impression of actual empty spaces which is not the case since an aperture still has an intine. I suggest replacing by 'gap in the exine'.

Reply: As explained above in major point 1), the pollen microgel has a hollow shell structure with three apertures entirely opened to the outer environment (also can refer to **Fig. 1e**). Therefore, we think that the terms of “hollow” and “holes” in this context are correct.

8. Page 3: “To address this challenge”: what challenge? No challenge is specified as of yet except for an allusion to potential of getting inspired by design of pollen grain.

Reply: We thank the Reviewer for the comment and agree that the sentence starts with vague transition. In the revised manuscript, the sentences have been edited as follows:

Pg. 3, lines 56-57

“Motivated by these intricate biological features, we developed a nature-inspired strategy to de-esterify pectin molecules throughout the entire pollen wall structure.”

9. Page 3, bottom half. 'Coinciding with an increase in gel-like properties...'. Until here, the manuscript has not mentioned any proof for gel-like properties to actually arise, hence it is somewhat misplaced to casually mention them here.

Reply: We thank the Reviewer for this suggestion and agree that the gel-like properties should be addressed before CLSM experiments in the context. In the revised manuscript, we have added a **Supplementary Figure 3c** to show the response of pollen gels to gravity, and incorporated it as follows:

Pg. 3, lines 71-77

*“These results were complemented by immunolabeling experiments with JIM7 antibody, which recognizes highly esterified epitopes of pectin²¹ (**Supplementary Fig. 3b**). In addition, the pollen dispersions exhibited more gel-like character, as indicated by increased resistance to gravity due to de-esterification of pectin (**Supplementary Fig. 3c**). Coinciding with an increase in gel-like properties, confocal laser scanning microscopy (CLSM) experiments revealed that chemical processing caused the pollen particles to swell and join together while individual particles remained structurally intact²³ (**Fig. 1e**).”*

Supplementary Figure 3 | c, Photographs of pollen dispersion in upright and reversed vials, showing the response of pollen gels to gravity.

10. Page 4, first line: 'soft microgel'. Before this line, there has been no mention of or proof for the fact that the particles have become soft or pliable (although this is presented later in the manuscript). I am wondering whether reorganizing the narrative would help the reader to follow.

“The aperture opening dissipates little energy, whereas stretching is an energy-consuming process.”: not clear based on what this statement is made. Is it based on elastic strain energy variations?

Reply: We thank the Reviewer for this comment and along with the comment 9 above, we have added the additional information to show the gel-like character of pollen microgel particles in **Supplementary Figure 3c**.

As for the second point, it is correct. In fact, the inflation of pollen microgel particles consist of two deformation modes. The first deformation mode occurs by opening three apertures. From the evolution of elastic strain energy, we confirmed that this deformation did not require much strain energy, localizing the deformation around the tips of apertures. On the other hand, the second deformation mode was induced by stretching the pollen shell along with the swelling of the intine layer. This process requires high strain energy, leading to the large global deformation throughout the pollen shell. Therefore, we have clarified this sentence in the revised manuscript as follows:

Pg. 7, lines 148-155

“The aperture opening does not require significant strain energy with highly localized deformation around the apertures, whereas subsequent stretching is an energy-consuming process that involves large-scale global deformation throughout the pollen shell. Stress and strain contours of 6 h KOH-treated pollen microgel particles ($M_{E/I} = 1.6$) indicate that the three apertures are initially opened, accompanied by a small volume change until the critical swelling ratio λ_{open} is reached, followed by a dramatic change in microgel particle diameter at the maximum swelling ratio, λ_{max} (Supplementary Videos 8 and 10).”

11. Page 6: What is the “initial swelling ratio, λ_{open} ”? Please define.

Reply: The initial swelling ratio, λ_{open} indicates the inflation of pollens mainly due to the opening of three apertures. When the intine layer is hydrated and swollen, the first deformation mode is opening those three apertures. This deformation mode doesn't require too much strain energy, and occur very rapidly, allowing water intake into the internal pollen cavity.

We have added this definition in the revised manuscript as follows:

Pg. 6-7, lines 139-148

“Numerical simulations also revealed the morphological evolution of a microgel particle during its structural expansion (Figs. 3b,c and Supplementary Figs. 14-15). We defined three key swelling ratios of pollen-derived microgel particles: λ_0 , λ_{open} , and λ_{max} . λ_0 is the initial swelling ratio ($\lambda_0=1$) when the total volume of the microgel particle is minimized. Once pectin hydration begins, rapid aperture opening occurs and intine swelling-induced pressure and resulting strain are highly localized at the tips of the three apertures. λ_{open} is the critical swelling ratio of a pollen microgel until its inflation is mainly accommodated by opening of three apertures. When a critical swelling ratio λ_{open} is reached, the deformation mode of a microgel particle transfers from aperture opening to stretching. Further expansion of the microgel particle induces large deformation of both the intine and exine layers until the maximum swelling ratio, λ_{max} , is reached.”

12. Page 7, third line: you may wish to add '... did not form microgel particles UNDER THE CONDITIONS USED HERE'.

Reply: We thank the Reviewer for this valuable comment and in the revised manuscript, the relevant remarks are added as follows:

Pg. 1, lines 25-28

“Our investigation of pollen grains and spores from across the plant kingdom further showed that microgel formation occurs with tested pollen species from eudicot plants while tested pollen species from non-eudicot plants did not form microgels.”

Pg. 7, lines 159-161

“To extend our findings, we tested pollen grains and spores from other clades and discovered that those from flowering monocots (Cattail) and gymnosperms (Pine) as well as spore-bearing lycophytes (Lycopodium) did not form microgel particles (Fig. 4).”

13. Page 7: “we discovered that this distinction alone is insufficient to explain hydration-induced pollen swelling.” This claim requires further discussion.

Reply: The underlying folding mechanisms of pollens have been investigated during harmomegathy where hydration and dehydration occur depending on the environmental

conditions around a pollen grain^{27,28}. As Katifori et al. reported in their paper, the unfolding/folding of pollen shells can accommodate this harmomegathy process with minimal energy cost²⁸. However, in our current study, we focused on the subsequent “germination” process after harmomegathy is halted. During this germination process, lipid components from the outer layer of pollen grains are gradually dispersed, allowing increased water uptake^{12,13}. Through this process, apertures are opened, which accelerates water uptake, leading to germination with significant swelling of pollen grains¹³. It is well-known that de-esterification of pectin in the intine layer is one of the key biochemical reactions during germination. However, none of studies have reported the role of the exine layer for germination. Through this study, we found that “interplay” of exine and intine layers only could induce inflation of pollen microgels, mimicking germination of pollen. We have added this point in the revised manuscript as follows.

Pg. 8, lines 176-187

“While experimental and computational studies of dehydration-induced harmomegathy have long recognized that the exine and intine layers of natural pollen grains have distinct mechanical properties¹¹, we discovered that this distinction alone is insufficient to explain hydration-induced pollen swelling. Strikingly, through direct experimental investigation with SEM and AFM and supporting computational modeling based on a bi-layered pollen shell structure, our findings reveal that the interplay of exine and intine layers plays a key role in dictating pollen swelling. The ratio of the stiffness values of the chemically tuned exine and intine layers must be within a certain, optimal range to trigger inflation of pollen-derived microgel particles in our experiments. This chemomechanical balance appears to be carefully regulated and suggests there is a chemorheological regulatory pathway at the individual particle level underpinning whether pollen hydration leads to successful germination or incipient drying¹³ (Supplementary Fig. 29).”

14. Page 7, last line: “mechanical strength” or stiffness?

Reply: We thank the Reviewer for pointing this out as it should be stiffness. We have corrected the term in the revised manuscript as follows:

Pg. 8, lines 182-184

“The ratio of the stiffness values of the chemically tuned exine and intine layers must be within a certain, optimal range to trigger inflation of pollen-derived microgel particles in our experiments.”

15. Figure 1C and E: the legend must indicate which primary antibody was used. What is “hydrostatic tensile stress”? Is it principal stress due to hydrostatic pressure?

Reply: We thank the Reviewer for the comment. The monoclonal JIM5 antibody was used as the primary antibody in Figure 1c (as well as Supplementary Fig. 3a), and Figure 1e is without any antibody treatment as the pollen is autofluorescence. In the revised manuscript, we added the antibody information in the figure legend as follows:

Pg. 10, lines 223-224

“Figure 1 / c, Immunofluorescence microscopy detection of de-esterified pectin within pollen shells using JIM5 as the primary antibody. d,...”

As for the second point, hydrostatic tensile stress indicates induced isotropic stress of the pollen shell due to applied hydrostatic pressure. Through this stress field, we could investigate where applied pressure was localized. Due to the symmetric geometry of the spherical pollen shell, the stress contour shows strong correlation with the max principal strain contour, explaining why large deformation occurs at a particular region of the pollen shell.

16. Page 19: Was AFM used or another indentation device? I am asking since the text mentions “instrumented indentation”. This term is generally used for cantilever-less indentation systems.

Reply: We apologize for the lack of clarity in the description. We used AFM for the stiffness measurement of pollen grains or microgel particles. In the revised manuscript, we have corrected the term to avoid this confusion as follows:

Pg. 19, lines 515-516

“This depth-sensing AFM indentation approach enables the quantitative determination of the Young’s modulus of the shell material^{35,36}.”

17. What was the AFM tip shape?

Reply: The AFM tips that we used were shaped as polygon-based pyramid with a half-cone angle of 20°. In the revised manuscript, we have added this information as follows:

Pg. 19-20, lines 531-537

“For all measurements, the NX-10 AFM instrument (Park Systems, Suwon, South Korea) was used with two AFM probes: (i) an aluminum reflex-coated silicon cantilever PPP-NCHR (Nanosensors, Neuchâtel, Switzerland) with a typical spring constant of ~42 N/m and a tip end radius of 5 nm; and (ii) a diamond cantilever TD26135 (Micro Star Technology, Huntsville, TX) with a spring constant of ~150 N/m and a tip end radius of 5 nm. The tips were shaped as polygon-based pyramid with a half-cone angle of 20°.”

18. Page 19: Was the immunolabel not preceded by a chemical fixation step? The protocol speaks of 'labeled sections' which would mean there would be a resin embedding step as well. There doesn't seem to be any mention of that (unless it is elsewhere in the methods). Please provide exhaustive description of the method.

Reply: In the immunolabeling study, we did not use a chemical fixation step and resin embedding step. To avoid the confusion, we would like to revise the manuscript as follows:

Pg. 19, lines 507-510

“Antibody-labeled pollen particles were examined immediately with a confocal laser scanning microscope (Zeiss LSM 710) and without any antifade reagents. Pollen grains without primary and/or secondary antibodies were used as controls.”

19. Page 19: To compare fluorescence intensity between samples it is crucial that they were prepared in parallel, that the microscope settings were identical for acquisition and that image processing was done in identical manner. Please confirm that this was the case.

Reply: We indeed confirm that the identical imaging acquisition settings (i.e., fluorescence light exposure time, strength) were used in the respective immunolabeling studies (JIM5 or JIM7). In the revised manuscript, we have added the information as follows:

Pg. 18, lines 472-473

“For the immunolabeling studies, all imaging conditions were identical in order to compare the fluorescence intensity of each sample.”

20. Legend of Figure 1 and elsewhere: Please specify that the SEM images are pseudo-colored.

Reply: We thank the Reviewer for the suggestion and in the revised manuscript, the following information is added where it is applicable:

Pg. 10, lines 218-220

“*Figure 1 | Engineering dynamic responsiveness in pollen particles. a, Steps involved in the extraction of pollen grains from sunflower plants, shown here with cross-sectional SEM images. The pollen structures are pseudo-colored.*”

Pg. 27, lines 834-837

“*Supplementary Figure 1 | Scanning electron microscope (SEM) images and schematic illustration of sunflower pollen grains. a-b, SEM images show distinctive features of pollen grains, including the exine, apertures, and intine. The pollen structures are pseudo-colored.*”

21. Why does particle size decrease between 6-12 hours (Figure 1D)?

Reply: The decrease of the size of pollen microgels from 6 to 12 hours is attributed to the gradually degradation of pectin contents under alkaline conditions³¹. In the Diaz *et al*,

pectin degraded at $\text{pH} > 8$ via β -elimination reaction. In Figure 1d, the diameter of the pollen gel was determined in $\text{pH} 7$ when pectin was in deprotonation state. Therefore, the pollens were in the swollen state due to the repulsive electrostatic force between $-\text{COO}^-$ groups. It is reasonable that the degradation of pectin will result in a lower swelling ratio.

22. Figure 2: I realize that the drawings are conceptual and simplified. However, pectin is a highly branched molecule (rather than a single strain as indicated). Maybe this could be symbolized at least conceptually?

Reply: We thank the Reviewer for the constructive comment and in the revised manuscript, the branched pectin structures are represented as follows in **Fig. 1** and **2**:

Figure 1 | Engineering dynamic responsiveness in pollen particles. b, Schematic illustration of stimuli-responsive pollen capsule behavior at multiple length scales. Left: Pollen structural components. Middle: Cellulose microfibrils organized by hemicellulose and pectin in the intine layer. Right: Hydrolytic conversion of pectin to pectate within the intine layer.

Figure 2 | Tunable pectate interactions enable rapid, stimuli-responsive material properties. a, Schematic illustration of increased pH inducing repulsion in pectin structure. **d,** Schematic illustration of cation-induced attraction and EDTA chelating agent-induced repulsion in pectin structure.

23. Figure 2a: The figure legend should specify whether the repulsion is made due to increase or decrease in pH .

Reply: We thank the Reviewer for the comment and in the revised manuscript, the figure legend has been revised as follows:

Pg. 11, lines 232-234

“Figure 2 | Tunable pectate interactions enable rapid, stimuli-responsive material properties. a, Schematic illustration of pH-dependent effects on pectin structure and corresponding intermolecular repulsion events.”

24. Figure 2c: How is “area swelling ratio” determined?

Reply: We thank the Reviewer for the detailed comment and in the revised manuscript, the definition of ‘area swelling ratio’ has been added to the figure legend as follows:

Pg. 11, lines 239-242

“Figure 2 | f, Quantitative comparison of ion-induced pollen de-swelling and swelling behaviors. Mean \pm s.d. are reported ($n = 5$ particles) in c and f, and the area swelling ratio was normalized by the initial area at pH 2 and pH 7 in c and f, respectively. Source data are provided as a Source Data file.”

25. Figure 2d: The legend is not descriptive of the events shown in figure.

Reply: We thank the Reviewer for this comment and in the revised manuscript, the figure legend has been revised as follows:

Pg. 11, lines 236-238

“Figure 2 | d, Schematic illustration of cation-induced attraction and EDTA chelating agent-induced repulsion between pectin molecules. e,...”

26. Figure 3 and FEA model: Did the presence of spikes actually influence the modeling results? Did they serve any other purpose than looking prettier when put side-by-side with micrographs? (This is fine, but it should be mentioned somewhere whether or not they have an effect or are purely meant to make the model look similar to the biological sample).

Reply: We thank the Reviewer for the interesting question and comment. We tried to mimic the pollen shell structure with spikes as close as possible for our modelling. However, there is another reason for us to introduce spikes on the sunflower pollen shell surface. As Katifori *et al.*³² reported in their paper, the unfolding/folding of pollen shells can accommodate the harmomegathy process with minimal energy cost. In case of sunflower pollen, it has spiky shells without any foldable shell design (*e.g.*, wrinkles or folded structure) except for apertures. Also, the spikes restrict extensive folding of the pollen shell during harmomegathy due to structural hindrance. Thus, we could set the minimal volume of those sunflower pollen microgel particles when their apertures were fully closed.

27. Supplementary Figure 2: I understand that these are conceptual drawings, but filling the entire pollen grain lumen with 'genetic material' simply ignores the cellular structure. Not the entire protoplast consists of genetic material! To not change the drawing, the light green material would need to be identified as 'Protoplast of the vegetative cell and generative cell (or sperm cells - depending on species)'. To draw this more accurately, at least the nucleus of the vegetative cell as well as the generative cell or sperm cells should be drawn.

Reply: We appreciate the Reviewer for this thoughtful comment and in the revised manuscript, the corresponding figures have been modified as follows:

Supplementary Figure 1 | Scanning electron microscope (SEM) images and schematic illustration of sunflower pollen grains. a-b, SEM images show distinctive features of pollen grains, including the exine, apertures, and intine. The pollen structures are *pseudo*-colored. **c,** Schematic illustration of pollen cement, exine, intine, and genetic material.

Supplementary Figure 2 | Schematic illustration of pollen microgel fabrication process. Microgel fabrication process involves the following four processing steps (as indicated by arrows): (1) Treatment with organic solvent to remove excess lipid coating; (2) Pollen incubation in alkaline conditions for 2 h at 80 °C under stirring condition (“Pollen shell extraction”); (3) Extended incubation in alkaline conditions at 80 °C for up to 12 h (“Pollen microgel formation”); and (4) Gelation of pollen microgels induced by water rinsing.

28. Supplementary Figure 3: Is this epifluorescence or confocal laser scanning microscopy? If the latter, are these optical sections or max projections? Please confirm that imaging was done under identical conditions for all samples.

Reply: We apologize for the missing information in the figure. The same confocal laser scanning microscopy described in the Methods section has been used, and the imaging conditions were identical for all samples after the first adjustment. Also, all presented images are from the middle optical sections while most of the pollens display the largest 2D diameter. We understand that this is an important information for the readers to follow; thus, in the revised manuscript, following sentences were added to the Methods section:

Pg. 18, lines 471-473

“Plane mode scanning was performed with a pixel dwell of 1.0 μ s. For the immunolabeling studies, all imaging conditions were identical in order to compare the fluorescence intensity of each sample.”

29. Supplementary Figure 5: Please specify that the gray tone images are brightfield micrographs and identify the optics used (DIC?)

Reply: We thank the Reviewer's comment and confirm that the grey tone images were from DIC. The following information has been added to the figure legend of the revised manuscript:

Pg. 31, lines 890-892

“The white arrows indicate the aperture openings on the pollen microgel particles. The corresponding bottom images were obtained in differential interference contrast (DIC) mode.”

Supplementary References

- 1 Katifori, E., Alben, S., Cerda, E., Nelson, D. R. & Dumais, J. Foldable structures and the natural design of pollen grains. *Proceedings of the National Academy of Sciences* **107**, 7635-7639 (2010).
- 2 Heslop-Harrison, J. Pollen Germination and Pollen-Tube Growth. *Int Rev Cytol* **107**, 1-78, doi:Doi 10.1016/S0074-7696(08)61072-4 (1987).
- 3 Bosch, M. & Hepler, P. K. Pectin methylesterases and pectin dynamics in pollen tubes. *The Plant Cell* **17**, 3219-3226 (2005).
- 4 Jarvis, M. C. Structure and properties of pectin gels in plant cell walls. *Plant, Cell & Environment* **7**, 153-164 (1984).
- 5 Willats, W. G., Knox, J. P. & Mikkelsen, J. D. Pectin: new insights into an old polymer are starting to gel. *Trends in Food Science & Technology* **17**, 97-104 (2006).
- 6 Liang, R.-h. *et al.* Extraction, characterization and spontaneous gel-forming property of pectin from creeping fig (*Ficus pumila* Linn.) seeds. *Carbohydrate polymers* **87**, 76-83 (2012).
- 7 Saunders, B. R. & Vincent, B. Microgel particles as model colloids: theory, properties and applications. *Advances in colloid and interface science* **80**, 1-25 (1999).
- 8 Plamper, F. A. & Richtering, W. Functional microgels and microgel systems. *Accounts of chemical research* **50**, 131-140 (2017).
- 9 Yuk, H. *et al.* Hydraulic hydrogel actuators and robots optically and sonically camouflaged in water. *Nature communications* **8**, 14230 (2017).
- 10 Guiducci, L., Fratzl, P., Bréchet, Y. J. & Dunlop, J. W. Pressurized honeycombs as soft-actuators: a theoretical study. *Journal of The Royal Society Interface* **11**, 20140458 (2014).
- 11 van der Sman, R. G. Hyperelastic models for hydration of cellular tissue. *Soft Matter* **11**, 7579-7591 (2015).
- 12 Hesse, M. Pollenkitt and viscin threads: their role in cementing pollen grains. *Grana* **20**, 145-152 (1981).
- 13 Heslop-Harrison, J. in *International review of cytology* Vol. 107 1-78 (Elsevier, 1987).
- 14 Moseley, P. *et al.* Modeling, Design, and Development of Soft Pneumatic Actuators with Finite Element Method *Advanced Engineering Materials* **18**, 978-988, doi:10.1002/adem.201500503 (2016).
- 15 Egan, B., McCarthy, M. A., Frizzell, R. M., Gray, P. J. & McCarthy, C. T. Modelling bearing failure in countersunk composite joints under quasi-static

- loading using 3D explicit finite element analysis. *Composite Structures* **108**, 963-977, doi:<https://doi.org/10.1016/j.compstruct.2013.10.033> (2014).
- 16 ABAQUS. *ABAQUS 2017 User documentation*. (Dassault Systemes Simulia Corp., 2017).
- 17 Lim, Z. X. & Cheong, K. Y. Effects of drying temperature and ethanol concentration on bipolar switching characteristics of natural Aloe vera-based memory devices. *Physical Chemistry Chemical Physics* **17**, 26833-26853, doi:10.1039/C5CP04622J (2015).
- 18 Monsoor, M. A., Kalapathy, U. & Proctor, A. Improved Method for Determination of Pectin Degree of Esterification by Diffuse Reflectance Fourier Transform Infrared Spectroscopy. *Journal of Agricultural and Food Chemistry* **49**, 2756-2760, doi:10.1021/jf0009448 (2001).
- 19 Urias-Orona, V. *et al.* A novel pectin material: extraction, characterization and gelling properties. *Int J Mol Sci* **11**, 3686-3695, doi:10.3390/ijms11103686 (2010).
- 20 Fajardo, A. R., Lopes, L. C., Pereira, A. G., Rubira, A. F. & Muniz, E. C. Polyelectrolyte complexes based on pectin-NH₂ and chondroitin sulfate. *Carbohydrate polymers* **87**, 1950-1955 (2012).
- 21 Buta, E. *et al.* FT-IR Characterization of Pollen Biochemistry, Viability, and Germination Capacity in Saintpaulia H. Wendl. Genotypes. *Journal of Spectroscopy* **2015**, 7, doi:10.1155/2015/706370 (2015).
- 22 Kyomugasho, C., Christiaens, S., Shpigelman, A., Van Loey, A. M. & Hendrickx, M. E. FT-IR spectroscopy, a reliable method for routine analysis of the degree of methylesterification of pectin in different fruit- and vegetable-based matrices. *Food Chemistry* **176**, 82-90, doi:<https://doi.org/10.1016/j.foodchem.2014.12.033> (2015).
- 23 Wellner, N., Kačuráková, M., Malovíková, A., Wilson, R. H. & Belton, P. S. FT-IR study of pectate and pectinate gels formed by divalent cations. *Carbohydrate Research* **308**, 123-131, doi:[https://doi.org/10.1016/S0008-6215\(98\)00065-2](https://doi.org/10.1016/S0008-6215(98)00065-2) (1998).
- 24 Bhattacharjee, T. *et al.* Writing in the granular gel medium. *Science advances* **1**, e1500655 (2015).
- 25 Hinton, T. J. *et al.* Three-dimensional printing of complex biological structures by freeform reversible embedding of suspended hydrogels. *Science advances* **1**, e1500758 (2015).
- 26 Hinton, T. J., Hudson, A., Pusch, K., Lee, A. & Feinberg, A. W. 3D printing PDMS elastomer in a hydrophilic support bath via freeform reversible embedding. *ACS biomaterials science & engineering* **2**, 1781-1786 (2016).
- 27 Payne, W. W. Observations of harmomegathy in pollen of Anthophyta. *Grana* **12**, 93-98 (1972).
- 28 Katifori, E., Alben, S., Cerda, E., Nelson, D. R. & Dumais, J. Foldable structures and the natural design of pollen grains. *Proc. Natl. Acad. Sci. U. S. A.* **107**, 7635-7639 (2010).
- 29 Parre, E. & Geitmann, A. More than a leak sealant. The mechanical properties of callose in pollen tubes. *Plant Physiol.* **137**, 274-286 (2005).
- 30 Qu, Z. & Meredith, J. C. The atypically high modulus of pollen exine. *J. R. Soc. Interface* **15** (2018).

- 31 Diaz, J. V., Anthon, G. E. & Barrett, D. M. Nonenzymatic degradation of citrus pectin and pectate during prolonged heating: effects of pH, temperature, and degree of methyl esterification. *Journal of agricultural and food chemistry* **55**, 5131-5136 (2007).
- 32 Katifori, E., Alben, S., Cerda, E., Nelson, D. R. & Dumais, J. Foldable structures and the natural design of pollen grains. *Proc. Natl. Acad. Sci. U. S. A.* **107**, 7635-7639 (2010).

Reviewers' Comments:

Reviewer #3:

Remarks to the Author:

The authors have taken great care to provide additional explanations and many of the previously highlighted points have been clarified. While this has improved the manuscript incrementally, my main concern remains: the validity of the finite element model. The additional explanations provided by the authors make it clear that my initial interpretation was accurate: Deformation of the outer envelope is accomplished by applying a hypothetical turgor pressure from the inside. While it could be argued that all modeling approaches are based on both educated assumptions and simplifications, in this case this simplification is such that the result becomes meaningless in terms of the outcome. Even if a swelling of the virtual pollen grain is achieved through this pressure application, the simulation can not be used to validate any of the structural concepts. It does not simulate the material swelling of the intine, nor the change in volume or the actual pressure that this process exerts on the exine.

The inconsistencies are expressed in the authors own words as they state

"In the meantime, through optical microscope, we confirmed that the intine layer was significantly hydrated and swollen in wet condition, increasing its total thickness."

and

"Thus, the volume increase of this pollen microgel system is caused by the swollen intine layer interplaying with the relatively stiffer exine."

but then they state:

"To simplify simulations, we disregarded solvent flow into and out of the pollen shell walls during swelling, as well as volumetric changes in the intine due to swelling effects. Instead, intine swelling was imposed by tensile circumferential stresses...due to the osmotic pressure exerted across its thin gel-like matrix. Hydration and swelling of the hyperelastic pollen shells associated with osmotic pressure effects were simulated based on a hypothetical internal pressure in our FE models, as illustrated in Supplementary Fig. 13d."

In a nutshell, the model simply does not have any predictive power that helps us understand the biological, physical or chemical processes at hand. I can therefore not support publication of that aspect of the manuscript.

That said, the experimental and chemical part of the manuscript is actually very nice and novel and that part alone could represent an full manuscript, although probably better suited for a different journal.

Reviewer #4:

Remarks to the Author:

I have been asked to look specifically at the modelling aspect of this paper, and determine whether the framework is valid. I agree with Reviewer 3; the model as set out does not sufficiently capture the underlying mechanisms required to have predictive power. A critical element is the swelling of the intine which has been neglected - imposing a hypothetical internal turgor pressure is not equivalent. I also do not think the reference used to justify this assumption (van der Sman) is doing the same thing - there the internal pressure is representing the inflation of the internal

cell vacuole. The van der Sman paper does include the effect of swelling of the gel, but I think that is via a different element in the model.

Editorial Note: Based on the advice from reviewers an additional reviewer with specific expertise was added; this reviewer is numbered Reviewer #4

Response to Reviewers for NCOMMS-19-11769B

Reviewer 3 Comments

The authors have taken great care to provide additional explanations and many of the previously highlighted points have been clarified. While this has improved the manuscript incrementally, my main concern remains: the validity of the finite element model. The additional explanations provided by the authors make it clear that my initial interpretation was accurate: Deformation of the outer envelope is accomplished by applying a hypothetical turgor pressure from the inside. While it could be argued that all modeling approaches are based on both educated assumptions and simplifications, in this case this simplification is such that the result becomes meaningless in terms of the outcome. Even if a swelling of the virtual pollen grain is achieved through this pressure application, the simulation can not be used to validate any of the structural concepts. It does not simulate the material swelling of the intine, nor the change in volume or the actual pressure that this process exerts on the exine. In a nutshell, the model simply does not have any predictive power that helps us understand the biological, physical or chemical processes at hand. I can therefore not support publication of that aspect of the manuscript.

That said, the experimental and chemical part of the manuscript is actually very nice and novel and that part alone could represent a full manuscript, although probably better suited for a different journal.

Reply: We sincerely thank the Reviewer for carefully reviewing our manuscript and for providing many excellent comments to help us revise and improve the manuscript. We were very encouraged by the Reviewer's positive comments that the experimental and chemical results of the manuscript are significant. We also agree that the modeling aspects can be improved and we have made significant efforts in this direction by removing the simplified hypothetical hydrostatic pressure and replacing it with a more detailed accounting of the swelling behavior of the intine layer.

To this end, we have developed a multiphysics model of intine layer swelling as a function of environmental conditions (*e.g.*, ion effects) that works in conjunction with the FEA modeling. This development has been part of our ongoing research in the field and we decided to include it in the revised manuscript for a more thorough treatment of this swelling aspect. The following points briefly summarize why we added these elements and how they improve our modeling overall.

- 1) As the Reviewer pointed out, the hyperelastic exine layer is inflated by swelling of the intine layer. In our previous version of the manuscript, we captured the exine inflation by applying a hypothetical hydrostatic pressure to the inner surface of the intine layer. This approach captured the basic concept of swelling-induced inflation, however, we agree that it can be further refined to better describe intine swelling based on actual changes in the system such as ionic changes.

2) Thus, in our revised manuscript, we introduced a multiphysics model of the intine layer, capturing its swelling behavior by accounting for chemo-electro-mechanical coupled field effects. Through this multiphysics model of the intine layer swelling, the resulting swelling-induced mechanical pressure was analytically calculated depending on factors such as solution pH and ion type/concentration. Since the intine swelling pressure equals the equilibrium pressure for inflating the exine layer, we used the intine swelling pressure as the boundary condition for exine inflation in our FEA modeling. Below, we compare the previous and new modeling approaches used in the previous versions of the manuscript and in the revised manuscript (**ROM Table 1**).

ROM Table 1. Comparison of our old and new numerical modeling approaches

	Past numerical modeling in previous manuscript versions	Extended numerical modeling in revised manuscript
Platform	FEA-based solid mechanics model	Multiphysics model in conjunction with FEA modeling
Materials	Exine: hyperelastic (FEA) Intine: hyperelastic (FEA)	Exine: hyperelastic (FEA) Intine: hydrogel (Multiphysics)
Boundary conditions	Exine: tightly bound with the intine layer Intine: hypothetical mechanical pressure applied to the inner surface of the intine	Exine: swelling-induced osmotic pressure exerted on the inner surface of the exine layer (predicted by intine swelling) Intine: electrochemical potential depending on pH and ionic conditions

3) Thus, two key factors that affect the swelling/deswelling behavior of the pollen microgel particle system were addressed through our extended numerical modeling: 1) intine swelling by the multiphysics model: the intine uptakes ionic solution from the surrounding medium to generate a considerable osmotic pressure, which is the driving force for inflating the outer exine; and 2) exine inflation by the FEA model: the stiffer exine along with the three apertures exerts an inhomogeneous constraint on intine swelling. The interplay between exine and intine layers plays a key role in controlling swelling behavior of pollen microgel particles, and our findings show strong agreement between experiment and simulation.

Pg. 5-7, lines 117-151 (Main)

Moreover, atomic force microscopy (AFM) measurements revealed that prolonged alkaline treatment caused a significant decrease in the Young's modulus of the exine layer (Supplementary Figs. 11-12). For example, the ratio of the exine to intine Young's modulus values (λ) decreased from ~ 3 (defatted pollen) to ~ 1.5 (12 h KOH-treated pollen).

To further evaluate how changes in the chemomechanical properties of pollen substructure layers affect the swelling/de-swelling behavior of pollen microgel particles, we also conducted computational simulations based on a multiphysics model that incorporated finite element analysis (FEA)²⁹⁻³¹. Specifically, the model simulated the extent of pollen intine swelling in different ionic solutions in line with the aforementioned experiments and accounted for the effects of chemo-electro-mechanical coupled fields on inflation/deflation of the exine layer (Supplementary Fig. 13). The pollen intine swelling pressure (p) was analytically calculated as a function of changes in environmental conditions (i.e., ion types and concentrations) and the computed values used as boundary conditions for analyzing inflation of the exine layer (i.e., exine inflation pressure, p_e) to evaluate the swelling/deswelling behavior of pollen microgels, and we systematically studied λ ratios from 0.15 to 8 (Supplementary Table 1). A stiffer exine ($\lambda > 2$) would be expected to impose a rigid boundary condition upon the hydrated intine, thereby constraining the potential for intine swelling (Fig. 3a). On the other hand, when $\lambda < 2$, both experiments and simulations demonstrated a steep increase in particle diameter ($\lambda = 1.6$, Supplementary Video 7). Moreover, the de-swelling of pollen microgel particles that was experimentally observed in the presence of multivalent cations (cf. Fig. 2) was also captured in the simulations for $\lambda < 1$, and arises from the intine becoming stiffer than the exine due to the chelation reaction as shown in Fig. 2d ($\lambda = 0.7$, Supplementary Video 8). Together with the gelation of de-esterified pectin molecules within the intine layer, these findings reveal that a softened exine layer plays an important role in modulating the mechanical properties of microgel particles by allowing greater swelling of the hydrated intine. Thus, the interplay of mechanical responses in the exine and intine layers dictates the morphological behavior of the microgel particles.

Taking into account the mechanical behavior of an individual microgel particle, its representative volume element (RVE) in the FEA was defined as a structure that consisted of a hyperelastic exine layer only with one aperture and one-third symmetry (Supplementary Fig. 13). We captured the large elastic deformation and strain energy density of a microgel particle by using FEA to simulate pollen exine inflation. Numerical simulations also revealed the morphological evolution of a microgel particle during its structural expansion (Figs. 3b,c and Supplementary Figs. 14-15)."

Cited References

- 29 Hong, W., Zhao, X. & Suo, Z. Large deformation and electrochemistry of polyelectrolyte gels. *J. Mech. Phys. Solids* **58**, 558-577 (2010).
- 30 Hong, W., Zhao, X., Zhou, J. & Suo, Z. A theory of coupled diffusion and large deformation in polymeric gels. *J. Mech. Phys. Solids* **56**, 1779-1793 (2008).
- 31 Liu, X. *et al.* Ingestible hydrogel device. *Nat. Commun.* **10** (2019).

Consequently, **Supplementary Fig. 13**, **Supplementary Table 1**, and **Fig. 3** have been revised as follows:

Supplementary Figure 13 | Multiphysics model of swelling of pollen microgel particles.

a, Schematic illustration of pollen microgel swelling that is composed of the interplay of intine (hydrogel) swelling and exine (rubber) inflation. The swelling-induced mechanical pressure exerted on the intine layer (,) due to osmotic pressure effects (Π) decreases as intine swelling proceeds. The intine swelling pressure (,) is equal to the equilibrium pressure required to inflate the exine layer (,). The swelling ratio of the intine layer, λ_i , is equal to the inflation ratio of the exine layer, λ_e , due to the intine layer being tightly bound to the exine layer. The hydration and swelling of pollen microgels due to osmotic pressure-related effects were predicted based on the multiphysics model of the intine layer (, =) and the pressure for exine inflation was applied to the internal surface of the exine layer using the boundary condition = , as proposed in Ref. 31. **b**, Schematics of the intine shell placed in an ionic solution with the computational domain and boundary conditions for numerical simulations. **c**, Geometrical configuration of pollen microgel particle shells used in the simulations, including exine layer and spikes with mesh grid concentrated on two tips of the aperture while avoiding excessive mesh

distortion during the aperture opening process. Note that the pollen microgel particle is modeled using one-third symmetry. **d**, Deformed contour of a pollen grain simulated with one-third symmetry to effectively use computational resources. **e**, Measurement setup for the swollen diameter of a deformed pollen microgel particle structure. Note that the deformation is irregular and the red circle marks the maximum deformed diameter.

Supplementary Table 1 | Ratios of exine to intine Young's modulus values for the parametric study on the swelling behavior of pollen microgel particles. Based on the modulus data of the exine (E_x) and intine (E_i) layers that were obtained in the AFM measurements, we calculated the Young's modulus ratio (E_x/E_i) depending on the treatment conditions of the microgel particles during material processing. We performed a parametric study by varying E_x/E_i from 0.15 to 8, as indicated in the table below. Verification and validation of the simulation models were implemented with respect to the swelling ratio based on a reference model for E_x/E_i of 1.6 (grey highlighted column in the table). When the applied pressure was 8.3 MPa, a simulated swelling ratio of 1.80 (associated with a swollen diameter of 50.4 μm) was observed, showing good agreement with swelling data for 6 h KOH-treated pollen at pH 10 condition. Other conditions, such as E_x/E_i of ≥ 2.0 or ≤ 0.5 represented pollen particles treated for shorter KOH incubation times (0 to 3 h) or exposed to different concentrations of divalent cations, respectively.

Modulus of the exine (E_x)	400	400	400	250	400	500	750	1000	1250	1750	2000
Modulus of the intine (E_i)	2609	2003	567	250	250	250	250	250	250	250	250
Modulus ratio (E_x/E_i)	0.15	0.2	0.7	1	1.6	2	3	4	5	7	8
Internal pressure (MPa)	1.6	2	8	8.3	8.3	8.3	8.3	8.3	8.3	8.3	8.3

Figure 3 | Mechanical response of pollen microgel particles. **a**, Swelling diameter of pollen microgel particle as a function of the ratio of the Young's modulus values of the exine to intine layers, M_{EI} . **b**, Predicted evolution of the strain energy density during pollen expansion as a function of the swelling ratio (λ) for three typical Young's modulus

ratio values ($\lambda = 0.15, 1.6$ and 3). **c**, Hydrostatic tensile stress and maximum principal strain contours of the pollen microgel particles ($\lambda = 0.15, 1.6$ and 3) for three critical swelling ratios (λ_0 , and) (labeled FEA), along with representative optical micrographs of pollen microgel particles in various chemical environments (*i.e.*, ionic changes) that triggered similar morphological evolutions. $\lambda = 0.15$ corresponds to pollen microgel particles immersed in 100 mM CaCl₂; $\lambda = 1.6$ to 6 h KOH-treated pollen microgels incubated in pH 10 solution; $\lambda = 3$ to defatted pollen grains. Source data are provided as a Source Data file.

Also, detailed descriptions of the extended modeling approach have been added in the revised Methods section as follows:

Pgs. 21-26, lines 552-737 (Methods)

Numerical Modeling for Swelling/Deswelling of Pollen Microgel Particles

For pollen grains, the intine layer is naturally hydrophobic due to highly esterified pectin, whereas the exine layer consists of sporopollenin that contains lipids and pollenkitt (pollen cement). Thus, the pollen shell is inherently hydrophobic and not swellable in its natural form. After the chemical treatment, the highly esterified pectin becomes de-esterified and hydrophilic while the exine layer becomes porous (due to removal of pollen cement). In addition, due to chemical treatment, the cytoplasm is also fully removed and thus the internal cavity is fully empty (**Supplementary Fig. 2**). Also, since the cytoplasmic contents, including cells and cellular debris, have to be released through the three apertures, the once-continuous intine layer is also ruptured. Thus, in the modeling, the two-layer pollen shell was fully discretized through the entire layer thickness at the apertures as shown in **Supplementary Fig. 2**. As a result, a pollen microgel particle has a hollow shell structure with three apertures, consisting of the outer exine and inner intine layers. Thus, the volumetric increase in a pollen microgel particle is caused by the interplay of the swollen intine layer and relatively stiffer exine layer. In particular, we carefully defined the swelling/deswelling behavior of the hydrogel-like intine layer due to water absorption and desorption depending on the osmotic pressure and inflation/deflation of the hyperelastic exine due to swelling-induced mechanical pressure of the underlying intine layer. Moreover, the three apertures are open, which allows rapid water intake into the internal cavity of the hollow particle that is concomitant with inflation of the pollen exine. Taken together, two key factors affect the swelling/deswelling behavior of the pollen microgel particle system: 1) intine uptakes ionic solution from the surrounding medium to generate a considerable osmotic pressure, which is the driving force for inflating the outer exine; and 2) the stiffer exine along with the three apertures exerts an inhomogeneous constraint on intine swelling. Therefore, the swelling-induced mechanical pressure exerted at the outer surface of the intine layer (σ ,) due to osmotic pressure decreases as swelling proceeds. The intine swelling pressure (σ ,) equals the equilibrium pressure that is required to inflate the exine layer (σ ,). The swelling ratio of the intine, λ , is equal to the inflation ratio of exine, λ , since the intine layer is tightly bound to the exine layer (**Supplementary Fig. 13a**). Details of the overall approach behind the multiphysics and computational models are described in Refs. 29-31.

Constitutive law for intine layer based on a multiphysics model Based on the experimental results, the intine layer of pollen microgel particles is shown to behave as a stimuli-responsive hydrogel, which provides the driving force for inflation and deflation of the outer exine layer in pollen microgel particles. For simplification, the pollen intine immersed in bath solution with mobile ions are represented approximately by spherical shell structures without apertures, as shown in **Supplementary Fig. 13b**. The equilibrium of pollen intine swelling was modeled in the radial direction due to spherical symmetry. Therefore, the present computational domain consists of (i) the pollen intine layer represented by a shell-like structure, (ii) the surrounding buffer medium, including both internal and external solutions, and (iii) the interface over the pollen intine and solution domains.

For theoretical formulation of the model in a Lagrangian scheme, several assumptions were made as follows: (i) the pollen intine system is maintained under isothermal conditions, such that the dissociation constant K is independent of temperature; (ii) no chemical reaction occurs that generates extra ions (ion concentration is constant); and (iii) the pollen intine is placed in an unstirred solution and thus the effect of convection on ionic diffusion is negligible. In line with the experiments, all four diffusive species in the system, including the pollen intine and surrounding solution, are considered in the present multiphysics model, namely hydrogen ions (H^+), hydroxide ions (OH^-), chloride ions (Cl^-), and respective cations (*i.e.*, K^+ or Ca^{2+}). According to the law of mass conservation, the Nernst-Planck equation is used for characterization of ionic diffusion, as given below,

$$\nabla \cdot \left[-D_k \nabla C_k - \frac{z_k F}{RT} C_k \nabla \psi \right] = 0, \quad (k = H^+, Cl^-, OH^-, K^+ \text{ or } Ca^{2+}) \quad (2)$$

where \mathbf{X} is the material coordinate, and C_k (mM) are the valence number and molar concentration of the ionic species k , ψ (V) is the electric potential, and R , T and F are the universal gas constant (8.314 J/(mol·K)), the room temperature (298 K), and Faraday constant (9.6487×10^4 C/mol), respectively. In addition, $\mathbf{C} = \mathbf{F}^T \mathbf{F}$, where $\mathbf{F} = \partial \mathbf{x} / \partial \mathbf{X} = (\lambda_r, \lambda_\theta, \lambda_\phi)$ is the deformation gradient.

For the distributive electric potential, the Poisson equation is given as follows,

$$\nabla_{\mathbf{x}} \cdot (J \varepsilon_0 \varepsilon_r \mathbf{C}^{-1} \nabla_{\mathbf{x}} \psi) = -F (z_f C_f + \sum_k C_k z_k) \quad (3)$$

where $J = \det \mathbf{F}$ is the change in volume of the pollen intine layer due to swelling, ε_0 is the vacuum dielectric permittivity (8.854×10^{-12} C²/N·m²), ε_r is the relative dielectric permittivity, z_f is the valence number of the fixed charge, and C_f is the fixed charge density given below⁴³,

$$C_f = \frac{C_m^0 JK}{JK + C_{H^+}} \quad (4)$$

where C_0 is the initial molar concentration of ionizable groups within the pollen intine in the dry state, K (mM) is the dissociation constant, and C_+ is the molar concentration of hydronium ions.

For the effect of mechanical equilibrium on pollen intine swelling, the mechanical governing equation with nonlinear deformation is employed as³⁰,

$$\nabla_{\mathbf{x}} \left(\mathbf{G}(\mathbf{F} - \mathbf{F}^{-T}) - \mathbf{J}^{-1} \mathbf{F}^{-T} \right) = 0 \quad (5)$$

where G (GPa) is the pollen intine shear modulus associated with the Young's modulus E and the Poisson's ratio ν via $\mathbf{G} = \mathbf{E}/[2(\mathbf{1} + \nu)]$, and Π is the osmotic pressure given by²⁹,

$$\Pi = -\frac{k_B T}{v_s} \left(\frac{C_s}{1 + \nu_s C_s} - \frac{1}{1 + \nu_s C_s} + \chi_H \left(\frac{1}{1 + \nu_s C_s} \right)^2 \right) - k_B T \left(\frac{1}{C_s v_s} C_k - \sum_k \bar{C}_k \right) \quad (6)$$

where the overbar ($\bar{}$) denotes the ion concentration in the solution, k_B is the temperature in the unit of energy, v_s is the volume per solvent species, C_s is the number of solvent species divided by the volume of the dry polymer, and χ_H is the Flory-Huggins parameter to characterize the interactions between solvent molecules and the polymer network inside the pollen intine. From Equation (6), it is found that the osmotic pressure inside the pollen intine Π is due to mixing of the polymer and solvent molecules, as well as due to the imbalance of ions in the external solution and the pollen intine, including mobile ions and fixed charges.

The swelling-induced Cauchy stress is given by

$$\boldsymbol{\sigma} = (\sigma_r, \sigma_\theta, \sigma_\phi) = \frac{1}{J} \mathbf{G}(\mathbf{F}\mathbf{F}^T - \mathbf{I}) - \Pi \mathbf{I} \quad (7)$$

where \mathbf{I} is a unit matrix and $\mathbf{I} = \mathbf{I}$ for a spherical symmetric system.

The stress in the radial direction is then obtained by substituting the deformation gradient and osmotic pressure into Equation (7), as follows:

$$\sigma_r = \frac{1}{J} G(\lambda_r^2 - 1) + \frac{k_B T}{v_s} \left(\ln\left(\frac{C_s}{1 + \nu_s C_s}\right) + \frac{1}{1 + \nu_s C_s} + \chi_H \left(\frac{1}{1 + \nu_s C_s}\right)^2 \right) - k_B T \left(\frac{1}{C_s v_s} C_k - \sum_k \bar{C}_k \right) \quad (8)$$

As shown in **Supplementary Fig. 13b**, for the spherical symmetric problem, the Neumann type of the electrochemical boundary conditions are imposed at the center of the pollen, located at $\theta = 0$, namely,

$$\frac{\partial C_k}{\partial X} = 0, \quad \frac{\partial \psi}{\partial X} = 0 \quad (9)$$

and Dirichlet boundary conditions are imposed at the edge of the solution domain,

$$C_k = \bar{C}_k, \quad \psi = 0 \quad (10)$$

A free swelling process is conducted to identify the ionic concentrations at equilibrium, and an equilibrium state over the spherical shell-solution interface is given as

$$G(\mathbf{F} - \mathbf{F}^{-T}) - \Pi \mathbf{F}^{-T} = 0 \quad (11)$$

The driving pressure is then achieved by inputting the equilibrium ionic concentrations into Equation (8)³¹. In order to perform numerical simulations of pollen intine swelling in the ionic solution, all the inputs required by the multiphysics model are tabulated in **Supplementary Table 2**. Among the inputs required in the present model, only the initial fixed charge density in the dry state ρ_0 was not directly obtained in the experiment. Thus, by an approach to the inverse problem, $\rho_0 = 1.7848 \times 10^4$ mM was identified through the present model associated with the experimental data, where the swelling of the deformed intine layer ($\sigma = 250$ MPa) was $\lambda = 1.5946$ at $\text{pH} = 7$. Moreover, the constant $\rho_0 = 1.7848 \times 10^4$ mM was further validated with the FEA modeling of the exine layer where intine swelling was constrained by the external exine layer, thus free swelling of the intine layer became swelling-induced mechanical pressure exerted to the internal surface of the exine layer. Since ρ_0 is an initial material property of the pollen intine, and is independent of the pollen intine swelling process, the well-identified value of ρ_0 could provide consistent modeling prediction which showed good agreement with experimental observations.

Constitutive law for exine layer The maximum swelling diameter of pollen microgel particles reached up to 1.8 times the size of their original diameter, as seen for the case of 6 h KOH-treated pollen samples. Large deformation around the tips of the three apertures can occur due to the high-stress concentration factor at the tip of a crack⁴⁴. Moreover, the swelling/de-swelling behavior of pollen microgel particles was reversible. Thus, the particles clearly showed rubber-like hyperelastic behavior. Therefore, we built three-dimensional computational models for simulating the micromechanics of **the exine layer in** pollen particles based on the classic hyperelastic neo-Hookean model⁴⁵. In the context of pollen microgel particles, about the paucity of reliable data on material properties of exine layer and the lack of detailed studies of **its** constitutive response led us to choose the simple but efficient neo-Hookean model, rather than complex models with many adjustable parameters.

The strain energy density of the neo-Hookean model is given by

$$U = \mu_0 \left(\left(\frac{1}{3} \lambda_1 \right)^2 + \left(\frac{1}{3} \lambda_2 \right)^2 + \left(\frac{1}{3} \lambda_3 \right)^2 - 3 \right) + \frac{1}{\lambda_1} (\lambda - 1)^2 \quad (12)$$

where U is the elastic strain energy per unit reference volume; λ_1 , λ_2 , and λ_3 are the three principal stretches; and λ is the total volume ratio, defined as $J = \lambda_1 \lambda_2 \lambda_3$ ^{45,46}. Two material constants, μ_0 and λ_1 , were chosen to simulate the **inflation** and **deflation** behavior of the **exine** layer under large elastic strain. μ_0 is related to the initial shear modulus G as follows,

$$\mu_0 = \frac{1}{2} \quad (13)$$

$$\lambda_1 = \frac{2}{3} \quad (14)$$

where K is the bulk modulus, and G and μ are related to the Young's modulus E , which retrieved from AFM force-based nanoindentation tests performed on the pollen particles (**Supplementary Figs. 11-12**), by the formulas, $G = E/2(1 + \nu)$ and $\mu = E/3(1 - 2\nu)$ with a Poisson's ratio, ν . Thus, we can express the two material constants, μ_0 and μ_1 , in terms of **exine** material parameters, E and ν , as follows,

$$\mu_0 = \frac{E}{4(1 + \nu)} \quad (15)$$

$$\mu_1 = \frac{6(1 - 2\nu)}{E} \quad (16)$$

Considering the presence of pores in the exine layer, a value of 0.4 for the Poisson's ratio was selected to account for exine compressibility.

FEA modeling for inflation/deflation of exine layer The geometry of pollen microgel particles was determined by direct experimental measurements using a scanning electron microscope (SEM). The average thickness of the exine layer was determined to be 0.58 μm , and the mean outer diameter of the particles was 28 μm . Spikes associated with the exine layer were defined as cones with a height of 5 μm and a base diameter of 3 μm . The aperture length was described relative to a central angle of 90°, and the tip radius of curvature of the aperture was set as 1.0 μm . This latter value was selected in order to avoid convergence issues related to large deformation around the aperture during finite element simulations of inflation of pollen exine (**Supplementary Fig. 13c**). A one-third symmetry structure with only one aperture and 38 spikes was constructed as the exine for one microgel particle, where symmetric boundary conditions were applied at the perimeter (**Supplementary Fig. 13d**). Furthermore, the density of exine was taken as 1 g/cm^3 . Considering the small size and low density of the pollen grains in solvents, gravitational effects were not considered in the model. The outer surface of the exine layer was tightly attached to the spikes, and an inflation pressure (P_{int}) generated by osmotic pressure of the intine (P_{ext}) was exerted on the inner surface of the exine layer. The effects of mesh size on the deformation capacity were studied by an adaptive mesh strategy, where the largest mesh size was 0.5 μm . Therefore, various mesh sizes were employed for small deformation at regions distant from the aperture and were decreased to a minimum value of 0.02 μm for large deformation ($\epsilon > 2$) around the tip of the aperture. Meanwhile, the artificial strain energy of the whole model associated with hourglass was less than 2% of the total internal energy. It should be noted that a severely inhomogeneous deformation was observed during the swelling process so the swelling diameter was measured by a red circle in the direction of maximum deformation, as indicated in **Supplementary Fig. 13e**. To investigate the inflation/deflation processes as well as related deformation mechanisms, we employed a three-dimensional model of the exine layer based on finite element analysis (FEA) using a commercial software package ABAQUS 2017® (Dassault Systèmes SIMULIA, Johnston, RI). The exine model was generated by Python scripts and then ran in parallel in 32 cores. Quasi-static solutions were calculated using ABAQUS/Explicit^{46,47} to simulate the inflation process of the pollen exine layer with different mechanical properties under an internal pressure while kinematic effects were ignored. The finite element mesh entailed solid eight-node brick elements with reduced integration (C3D8R), while the spike on the pollen exine surface was treated as a rigid surface without deformation throughout the entire simulation. A total of 75,938 elements were used in the pollen exine model. A pseudo step time of 30 μs for inflating and 30 μs for deflating the

exine structure was set in order to minimize the kinetic energy. The inflation pressure ($P_{int} = P_{ext}$) was predicted by the intine swelling model ($P_{int} = P_{ext}$) based on the relationship, $P_{int} = P_{ext}$; thus, the boundary condition applied to the internal surface of the exine layer was $P_{int} = P_{ext}$. A maximum internal pressure ($P_{int} = 8.3$ MPa) of 8.3 MPa, induced by the swollen intine layer, was predicted by the multiphysics model that corresponded to 6 h KOH-treated pollen microgels incubated in pH 10 solution; this was the treatment condition which exhibited maximum swelling among all pollen microgel particle specimens. The reference pressure was, therefore, set as 8.3 MPa ($P_{int} = 8.3$ MPa) and 0 MPa ($P_{int} = 0$ MPa) for conditions corresponding to pH 10 and pH 2 conditions, respectively, for all of the simulations, and the Young's modulus values and the predicted internal pressure values were systematically varied in the parametric studies, as indicated in **Supplementary Table 1**. The pressure boundary condition on the inner surface of the exine layer was applied with a defined load profile gradually increasing up to 8.3 MPa and then decreasing to zero to improve the convergence of the model in large deformation simulations.

Supplementary Table 2 has been added in the revised manuscript and describes the input parameters for the numerical simulations as follows:

Supplementary Table 2 | Input parameters for numerical simulations of the intine layer shell.

Parameter	Value
Relative dielectric permittivity ϵ_r	80
Young's modulus E (pH = 7)	250 MPa
Poisson's ratio ν	0.475
Flory-Huggins interaction parameter χ_H	0.1
Radius of the pollen intine at dry state r_0	13.42 μm
Thickness of the pollen intine at dry state δ	0.54 μm
Dissociation constant K	0.1 mM
External ionic concentration \bar{C}_{H^+}	$10^{(3-\text{pH})}$ mM
External ionic concentration \bar{C}_{OH^-}	$10^{(\text{pH}-1)}$ mM
External ionic concentration \bar{C}_k	Depending on the solution

In addition, related Supplementary Figs. 14-15 have been updated as follows:

Supplementary Figure 14 | Hydrostatic tensile stress of the sunflower pollen microgels over various Young's modulus ratios ($M_{E/I} = 0.15, 0.2, 0.7, 1, 1.6, 2, 3, 4, 5, 7$ and 8) at three critical swelling ratios (λ_0 , λ_{open} and λ_{max}). $M_{E/I} < 1$ corresponds to the experimental results of pollen microgel particles exposed to various ion species with different concentrations. $M_{E/I} > 1$ corresponds to the experimental results of varied pH values. Typically, $M_{E/I} = 0.15$, $M_{E/I} = 0.2$, and $M_{E/I} = 0.7$ represent the 100 mM CaCl_2 solution, 10 mM CaCl_2 , and 5 mM KCl solution conditions, respectively; $M_{E/I} = 1.6$ shows the 6 h KOH -treated pollen at pH 10 condition (as presented in Fig. 3a).

Supplementary Figure 15 | Maximum principal strain of the sunflower pollen microgel particles over various Young's modulus ratios ($M_{E/I} = 0.15, 0.2, 0.7, 1, 1.6,$

2, 3, 4, 5, 7 and 8) at three critical swelling ratios, (λ_c and λ_c). $\lambda_c < 1$ corresponds to the experimental results of pollen microgel particles exposed to various ion species with different concentrations. $\lambda_c > 1$ corresponds to the experimental results of varied pH values. Typically, $\lambda_c = 0.15$, $\lambda_c = 0.2$, and $\lambda_c = 0.7$ represent the 100 mM CaCl₂ solution, 10 mM CaCl₂, and 5 mM KCl solution conditions, respectively; $\lambda_c = 1.6$ shows the 6 h KOH-treated pollen at pH 10 condition (as presented in Fig. 3a).

Reviewer 4 Comments

I have been asked to look specifically at the modelling aspect of this paper, and determine whether the framework is valid. I agree with Reviewer 3; the model as set out does not sufficiently capture the underlying mechanisms required to have predictive power. A critical element is the swelling of the intine which has been neglected - imposing a hypothetical internal turgor pressure is not equivalent. I also do not think the reference used to justify this assumption (van der Sman) is doing the same thing - there the internal pressure is representing the inflation of the internal cell vacuole. The van der Sman paper does include the effect of swelling of the gel, but I think that is via a different element in the model.

Reply: We sincerely thank the Reviewer for taking the time to carefully review our manuscript and providing helpful feedback to improve the manuscript. We have extensively expanded the modeling approach by incorporating the chemo-electro-mechanical multiphysics model to theoretically obtain the pollen intine swelling pressure, rather than imposing a hypothetical pressure as we had done before. Also, we have removed several references found to be less relevant and added other pertinent ones that support our modeling approach. We hope that the revised manuscript is deemed worthy of publication and will hopefully reach a wide audience and inspire future readers to explore this exciting field.

Reviewers' Comments:

Reviewer #3:

Remarks to the Author:

The major modification in the new version of the manuscript is the model developed to simulate the swelling process of the pollen grain. To facilitate the comparison of the old and new models the pollen grain is considered with a hollow spherical geometry. Two shell layers comprising the wall of the spherical shell represent the intine (inner layer) and exine (outer layer) of the pollen grain. The swelling of the pollen grain is suggested to be the result of swelling of the intine layer leading to a pressure exerted on the internal surface of the exine layer, stretching and expanding the latter. In the previous model, the swelling process was simulated by exerting a hypothetical internal pressure on the inner surface of the intine layer, stretching both intine and exine layers leading to an increase in the radius of the sphere. In the new model, a multiphysics approach is adopted for the deformation of the intine in which parameters such as ionic flow and osmotic pressure are accounted for in determining the swelling of the intine layer. Subsequently, the stress arising due to swelling is calculated and considered to constitute the pressure exerted from intine onto the inner face of the exine layer. From this point onward, a classic finite element modeling approach is adopted in which the exine layer is deformed by a pressure applied to its inner surface.

Major considerations

Overall, the model presented in the current version is conceptually more relevant to the biophysical processes occurring during pollen grain swelling, when compared to the model in previous versions of the manuscript. However, given the multi-physics approach, it would be advisable to seek the opinion of an expert in electrochemical modeling of hydrogel swelling to fully assess the electrochemical aspects of the multiphysics model for the intine swelling (especially the use of equations 2-4). The equations in this part are presented in highly succinct manner and intermediate steps are not always presented. In some parts (e.g. lines 601-613), references are needed. More extensive explanations and data pertaining to the derivation of these equations should be provided here or in the supplemental materials.

Although this new approach is certainly more relevant than the previous, I remain not entirely convinced by the implementation of the new model: the multiphysics model of the intine swelling and the deformation of the exine seem uncoupled. While the authors mention the significance of the mechanical constraint that is imposed by the exine on the swelling intine (lines 134, 574-575), this aspect does not seem to be implemented in the multiphysics model. The intine seems to be able to swell freely and mechanically unconstrained. This is important as the constraint from the comparatively rigid exine can be expected to significantly limit the swelling of the intine and, as a result, the expansion of the whole pollen grain.

Finally, the descriptions and the solution to the presented multiphysics equations could be more expanded for clarity. For instance, it is not clear what code/software was used for the numerical solution of multiphysics equations.

Minor comments

- Figure 3a: Please specify the scaling of the x-axis. Also, some data points in Figure 3a seem different from the previous version and no clarification is provided in the rebuttal letter with regards to this change. While minor in magnitude, this uncommented alteration does not instill confidence.
- Are the presented deformations/stresses in Fig. 3c ($ME/I=1.6$ and 3) identical to those presented in the previous versions of the manuscript? Since the working mechanism of the new and old models are entirely different, it is interesting for the models to produce virtually identical outputs for the same ratios of ME/I . It would be helpful if the authors provided some explanation on this issue, albeit to satisfy the reviewers' need for reassurance that the modeling approach is actually revised.
- Line 134: What does "constraining the potential for intine swelling" mean? Does it reduce it or does it prevent it? Please rephrase for clarity.

- Supplementary Figure 13a: 1- It is not clear what the inward blue arrows on the rightmost schematic indicate. 2- Where are swelling and inflation ratios properly defined? I had made this comment in previous versions. The use of these terms in this figure legend is vague. Additionally, how do these ratios defined by λ correlate with λ parameters of the Neo-Hookean energy potential (if they do)?
- Supplementary Figure 13b: The authors mention internal and external solutions for the intine. Considering that the intine is ruptured upon the treatment, I wonder how the internal and external solutions are distinguished. Also, the drawing shows more than half of the grain and hence r_o and r_i are indicated too big it seems.
- Supplementary Figure 13c: Avoiding excessive mesh distortion is mentioned but it is unclear what control was set in place to enforce it prior to running the simulation. Was it the mesh size, aspect ratio or angles? Also, it is unclear how the geometries presented in (c) and (d) correlate.
- Supplementary Figures 13d,e, 29: What does the heatmap show - stress, strain? Please provide LUT.
- Figure 3c, Supp Figure 14 and likely elsewhere: As mentioned for the previous versions, the use of the term "hydrostatic tensile stress" is ambiguous. I am aware that the authors mean to refer to the stress arising in the shell due to internal hydrostatic pressure. However, I find the use of hydrostatic for stress generated in the solid shell odd. Other measures of stress such as max. principal stress may be chosen for the label.
- Line 684: "which WERE retrieved"?
- Line 723: How does a 30-microsecond step help to reduce the kinetic effects? Is it not so that a shorter step time increases the inertial effects, compared to a, say 1-second step? Here the authors could actually compare this with the real time scale of the swelling observed in the experiments. Further, the authors should provide a comment mentioning the parameter that helped to determine that this time step value was appropriate.

Reviewer #5:

Remarks to the Author:

In this manuscript, the authors report an approach to transform ultratough pollen grains into pliable, soft microgel by remodeling pollen shells and markedly alter the Young's modulus. The effect of internal pressure on the exine layer was also verified through the deformed shapes by using a continuum mechanics model. The paper is publishable but the authors may need to consider the following comments.

1) Ln. 656, "The driving pressure is then achieved by inputting the equilibrium ionic concentrations into..." If only equilibrium states are to be considered, no field theory (Eq. 2-7) is needed, and the authors are suggested to use simpler descriptions, e.g. the Donnan model, directly.

2) Ln. 697, "Considering the presence of pores in the exine layer, a value of 0.4 for the Poisson's ratio was

selected to account for exine compressibility." It seems that the moduli, especially the bulk modulus is highly affected by the value of Poisson's ratio. How did the authors obtain this particular number? Is it chosen to fit some data? How would it change the final results?

3) Ln. 725, "Quasi-static solutions were calculated using ABAQUS/Explicit..." Abaqus Explicit is known for dynamic simulations. How did the authors obtain the quasi-static solutions?

4) It seems that the effect of the intine inflation is treated also as quasi-static by assuming homogeneous swelling throughout the intine core. How is this assumption justified? Would the existence and opening of the apertures change the local swelling and consequently the deformation of the exine layer?

Response to Reviewers for NCOMMS-19-11769C

Reviewer #3 Comments

The major modification in the new version of the manuscript is the model developed to simulate the swelling process of the pollen grain. To facilitate the comparison of the old and new models the pollen grain is considered with a hollow spherical geometry. Two shell layers comprising the wall of the spherical shell represent the intine (inner layer) and exine (outer layer) of the pollen grain. The swelling of the pollen grain is suggested to be the result of swelling of the intine layer leading to a pressure exerted on the internal surface of the exine layer, stretching and expanding the latter. In the previous model, the swelling process was simulated by exerting a hypothetical internal pressure on the inner surface of the intine layer, stretching both intine and exine layers leading to an increase in the radius of the sphere. In the new model, a multiphysics approach is adopted for the deformation of the intine in which parameters such as ionic flow and osmotic pressure are accounted for in determining the swelling of the intine layer. Subsequently, the stress arising due to swelling is calculated and considered to constitute the pressure exerted from intine onto the inner face of the exine layer. From this point onward, a classic finite element modeling approach is adopted in which the exine layer is deformed by a pressure applied to its inner surface.

Reply: We sincerely thank the Reviewer for carefully reviewing our revised manuscript and for recognizing the model improvements that we have made. We have fully incorporated all additional suggestions made by the Reviewer, as described below.

Major considerations

Comment 1. Overall, the model presented in the current version is conceptually more relevant to the biophysical processes occurring during pollen grain swelling, when compared to the model in previous versions of the manuscript. However, given the multiphysics approach, it would be advisable to seek the opinion of an expert in electrochemical modeling of hydrogel swelling to fully assess the electrochemical aspects of the multiphysics model for the intine swelling (especially the use of equations 2-4). The equations in this part are presented in highly succinct manner and intermediate steps are not always presented. In some parts (e.g. lines 601-613), references are needed. More extensive explanations and data pertaining to the derivation of these equations should be provided here or in the supplemental materials.

Reply: We thank the Reviewer for recognizing the more conceptually relevant nature of the improved modeling approach and for the helpful suggestion to improve the clarity of its description. In the revised manuscript (Supplementary Methods), we have provided more detailed description of the equation derivations (including term meanings) and additional references as follows:

Supplementary Methods, Pg. 2-3

“For theoretical formulation of the model in Lagrangian scheme, several assumptions were made as follows: (i) the pollen intine system is maintained at isothermal condition, such that the dissociation constant K is independent of temperature, (ii) no chemical reaction occurs for generation of extra ions, and (iii) the pollen intine is placed in an unstirred solution and thus the effect of convection on ionic diffusion is negligible¹. In line with the experiments, all four diffusive species in the system, including the pollen intine and surrounding solution, are considered in the present multiphysics model, namely hydrogen ions (H^+), hydroxide ions (OH^-), chloride ions (Cl^-), and respective cations (i.e., K^+ or Ca^{2+}). According to the law of mass conservation, the Nernst-Planck equation is used for characterization of ionic diffusion, as given below²,

$$\nabla_{\mathbf{x}} \cdot \mathbf{J}_k(\mathbf{X}) = 0, \quad (k = H^+, Cl^-, K^+, \text{ or } Ca^{2+}) \quad (2)$$

where \mathbf{X} is the material coordinate, and $\mathbf{J}_k(\mathbf{X})$ is the nominal flux of each mobile ion in the solution, defined as,

$$\mathbf{J}_k(\mathbf{X}) = \mathbf{C}^{-1} [\nabla_{\mathbf{x}} C_k + z_k C_k F_r \nabla_{\mathbf{x}} \Psi / (R_0 T)] \quad (3)$$

, where z_k and C_k (mM) are the valence number and molar concentration of the ionic species k , Ψ (V) is the electric potential, and R_0 , T and F_r are the universal gas constant (8.314 J/(mol·K)), the room temperature (298 K), and Faraday constant (9.6487×10^4 C/mol), respectively. In addition, $\mathbf{C} = \mathbf{F}^T \mathbf{F}$, where $\mathbf{F} = \partial \mathbf{x} / \partial \mathbf{X} = (\lambda_r, \lambda_\theta, \lambda_\phi)$ is the deformation gradient. $\lambda_\theta = u_r / r$ is the deformation gradient in the circumferential direction, where u_r is the displacement of spherical pollen intine in radial direction, and, in turn, is equivalent to the swelling ratio (λ) in the present study.

For the distributive electric potential, the Poisson equation is given as follows²,

$$\nabla_{\mathbf{x}} \cdot (\mathbf{J} \varepsilon_0 \varepsilon_r \mathbf{C}^{-1} \nabla_{\mathbf{x}} \Psi) = -F (z_f C_f + \sum_k C_k z_k) \quad (4)$$

where $J = \det \mathbf{F}$ is the change in volume of the pollen intine layer due to swelling, ε_0 is the vacuum dielectric permittivity (8.854×10^{-12} C²/N·m²), ε_r is the relative dielectric permittivity, z_f is the valence number of the fixed charge, and C_f is the fixed charge density given below³,

$$C_f = \frac{C_m^0 JK}{JK + C_{H^+}} \quad (5)$$

where C_m^0 is the initial molar concentration of ionizable groups within the pollen intine in the dry state, K (mM) is the dissociation constant, and C_{H^+} is the molar concentration of hydronium ions. It is noteworthy that the fixed charge density in the multiphysics model isn't constant, being varied depending on ionic concentrations and swelling ratios.

By the law of momentum conservation, the governing equation for mechanical equilibrium with nonlinear deformation is given as follows,

$$\nabla_{\mathbf{x}} \cdot \mathbf{P} = 0 \quad (6)$$

,where $\mathbf{P} = \mathbf{P}_e + \mathbf{P}_{os}$ is the nominal stress tensor contributed by two components, namely the elastic stress \mathbf{P}_e and nominal osmotic pressure \mathbf{P}_{os} , and thus it is formulated by

$$\mathbf{P} = G(\mathbf{F} - \mathbf{F}^{-T}) - J\Pi\mathbf{F}^{-T} \quad (7)''$$

References

- 1 Nikonenko, V., Lebedev, K., Manzanares, J. & Pourcelly, G. Modelling the transport of carbonic acid anions through anion-exchange membranes. *Electrochimica Acta* **48**, 3639-3650, doi:10.1016/S0013-4686(03)00485-7 (2003).
- 2 Li, H., Luo, R., Birgersson, E. & Lam, K. Y. A chemo-electro-mechanical model for simulation of responsive deformation of glucose-sensitive hydrogels with the effect of enzyme catalysis. *Journal of the Mechanics and Physics of Solids* **57**, 369-382, doi:10.1016/j.jmps.2008.10.007 (2009).
- 3 Lai, F. & Li, H. Transient modeling of the reversible response of the hydrogel to the change in the ionic strength of solutions. *Mech. Mater.* **43**, 287-298 (2011).

Comment 2. Although this new approach is certainly more relevant than the previous, I remain not entirely convinced by the implementation of the new model: the multiphysics model of the intine swelling and the deformation of the exine seem uncoupled. While the authors mention the significance of the mechanical constraint that is imposed by the exine on the swelling intine (lines 134, 574-575), this aspect does not seem to be implemented in the multiphysics model. The intine seems to be able to swell freely and mechanically unconstrained. This is important as the constraint from the comparatively rigid exine can be expected to significantly limit the swelling of the intine and, as a result, the expansion of the whole pollen grain.

Reply: Thank you for the Reviewer's valuable comment. We want to clarify how we coupled these two approaches as follows:

- 1) *Why did we introduce the multiphysics model in addition to FEA for our simulations?*
Each modeling approach has particular strengths to capture different aspects of the pollen microgel simulation needs. The FEA method is the most efficient and widely used method particularly, for solving complex two- and three- dimensional structural problems. While FEA approaches have been used to simulate the swelling behavior of various hydrogels in the past decade¹⁻⁴, this approach requires the assumption of a constant fixed charge during ion diffusion. In this regard, the multiphysics theory is useful because it includes the effect of mobile ionic diffusion in an electric field on mechanical deformation in response to the

change of environmental pH or ionic strength, while varying fixed charges accordingly. Since the pollen microgel systems are significantly influenced by pH and ionic strength, we believe that the multiphysics model can accurately capture those phenomena. At the same time, the multiphysics approach alone can only solve a system with a simple geometry. As a result, we had to combine these two approaches together for the pollen system, which has a complicated geometry due to the bi-layered microstructure with apertures while also taking into account ionic diffusion effects.

2) *How did we resolve the compatibility issue due to this dual approach?*

We developed a method by combing two deformation modes of pollen intine and exine through shared boundary conditions (radial force of the outer intine layer = hydrostatic tensile force exerted on the internal surface of exine) at the interface of those two adjoining layers. The proposed approach follows a similar approach to that taken in a recent study by the Zhao group⁵. The overall modeling process is presented in **ROM Fig. 1**.

- Forces induce the expansion of pollen microgels due to swelling of the intine layer. The force magnitudes correlate with pH or ionic strength changes and can be calculated by the multiphysics model of the intine.
- Deformation of exine (i.e., radial displacement of the outer exine surface) in the FEA simulations due to intine swelling can be directly compared with the experimental results from the observation of the external surface of the pollen microgels under various environmental pH or ionic strength/type conditions. It is noteworthy that the mechanical constraint should be naturally imposed on the intine due to the stiffness of the exine layer, reducing the equilibrium swelling diameter of the intine. However, the exine does not alter the intrinsic swelling capability (the capability of water absorption) of the intine since osmotic pressure is due to mixing of the polymer and solvent molecules, as well as due to the imbalance of ions in the external solution and the pollen intine.
- However, in fact, we could not directly predict forces from the multiphysics model due to the existence of one fitting parameter, the initial fixed charge. Thus, we should determine the parameters by comparing the simulation results with the experiments (**ROM Fig. 1A**). First, we chose one swelling condition of the pollen microgels as a reference state (e.g., 6 hr-KOH treated pollen microgels from pH 2 to pH 10) and determined the parameter. Since the experimental observation found that the intine layer was already fully esterified after 3 hr-KOH treatment, we assumed that the swelling behavior of the intine for all microgels should be identical under the same environmental conditions. In order to increase the accuracy of our simulations, we repeated this fitting process by changing the reference state with different pollen microgels (e.g., 3 hr-KOH pollen or 12 hr-KOH pollen). We confirmed that regardless of the KOH treatment time or environmental changes, the initial fixed charge was almost identical, validating our approach.
- After determination of the initial fixed charge, we redid the whole modeling process by varying all environmental conditions as well as microgel properties, and obtained the stress/strain contours for further analyses (**ROM Fig. 1B**).

ROM Fig. 1. Flowchart of the modeling process by the dual approach combining multiphysics and FEA modeling. A. Flowchart of the modeling process for the determination of the initial fixed charge for the intine multiphysics model. B. Flowchart of the modeling process for stress/strain analyses of pollen microgels, after obtaining the constant value of the initial fixed charge by repeating the process.

References

- 1 Marcombe, R. *et al.* A theory of constrained swelling of a pH-sensitive hydrogel. *Soft Matter* **6**, doi:10.1039/b917211d (2010).
- 2 Hong, W., Liu, Z. & Suo, Z. Inhomogeneous swelling of a gel in equilibrium with a solvent and mechanical load. *International Journal of Solids and Structures* **46**, 3282-3289, doi:10.1016/j.ijsolstr.2009.04.022 (2009).
- 3 Chester, S. A., Di Leo, C. V. & Anand, L. A finite element implementation of a coupled diffusion-deformation theory for elastomeric gels. *International Journal of Solids and Structures* **52**, 1-18, doi:10.1016/j.ijsolstr.2014.08.015 (2015).
- 4 Zheng, S. & Liu, Z. Constitutive model of salt concentration-sensitive hydrogel. *Mechanics of Materials* **136**, doi:10.1016/j.mechmat.2019.103092 (2019).
- 5 Liu, X. *et al.* Ingestible hydrogel device. *Nat Commun* **10**, 493, doi:10.1038/s41467-019-08355-2 (2019).

Comment 3. Finally, the descriptions and the solution to the presented multiphysics equations could be more expanded for clarity. For instance, it is not clear what code/software was used for the numerical solution of multiphysics equations.

Reply: Please refer to our answer to Comment 1 for the expanded description of the multiphysics questions. Moreover, the multiphysics modeling was conducted by the commercial software COMSOL Multiphysics 5.3 (COMSOL Inc., Stockholm, Sweden). The partial differential equations (PDEs) interfaces were used for the ionic transport and pollen intine deformation, as well as the electrostatics module for the electric distribution in the domain. All the dependent variables in PDEs and electrostatics modules were computed by the fully coupled solver. We have added this information in the revised manuscript (Supplementary Methods) as follows:

Supplementary Methods, Pg. 4

“...The simulation was conducted by the commercial software COMSOL Multiphysics 5.3 (COMSOL Inc., Stockholm, Sweden). The partial differential equations (PDEs) interfaces were used for the ionic transport and pollen intine deformation, as well as the electrostatics module for the electric distribution in the domain. All the dependent variables in PDEs and electrostatics module were computed by the fully coupled solver....”

Minor comments

Comment 4. Figure 3a: Please specify the scaling of the x-axis. Also, some data points in Figure 3a seem different from the previous version and no clarification is provided in the rebuttal letter with regards to this change. While minor in magnitude, this uncommented alteration does not instil confidence.

Reply: We thank the Reviewer for his/her attention to the details of Figure 3a. We clarified the raised points by the Reviewer.

1. The linear-log plot of Figure 3a has a linear scale on the y-axis, and a logarithmic scale on the x-axis covering the ratio of Young's modulus values of the exine to the intine from 0.1 to 10.
2. We could not measure the Young's modulus of the intine and exine layers when the ionic concentrations were varied. Thus, we could not accurately display any data points from experiments since the ratio of Young's modulus values was not identified. However, using our multiphysics model, we could calculate the Young's moduli of the intine at different ionic concentrations, assuming that the Young's modulus of the exine layer remained constant. Thus, we could accurately plot those data points under different ionic solution conditions, as compared with experimental results in the revised Figure 3a.

Comment 5. Are the presented deformations/stresses in Fig. 3c (ME/I=1.6 and 3) identical to those presented in the previous versions of the manuscript? Since the working

mechanism of the new and old models are entirely different, it is interesting for the models to produce virtually identical outputs for the same ratios of M_E/I . It would be helpful if the authors provided some explanation on this issue, albeit to satisfy the reviewers' need for reassurance that the modeling approach is actually revised.

Reply: Thank you for your helpful suggestion. We would like to clarify a few key points regarding this comment as follows:

1. As the Reviewer pointed out, the working mechanism of the intine layer swelling in the new and old models are different. We assumed a hyperelastic and passive intine in our previous FEA model, whereas the intine is swellable in response to various environment stimuli in our new multiphysics model. However, the deformation mechanism of the exine in both models remains the same since the inner surface of exine was subjected to the forces induced by either intine inflation (our previous FEM model) or swelling (our new multiphysics model). The magnitude of such forces should be varied depending on the constitutive law of the intine layer.
2. Since **Fig. 3c** presents the stress/strain contours of the exine layer after inflation, regardless of the magnitude of forces, overall, the results should be the same as we observed. The deformation mechanism of the exine should depend on 1) its geometry, and 2) its stiffness in response to the applied force.
3. As we discussed in the manuscript, the role of the exine is important since, without its contribution, pollen grains are just rigid and passive microspheres even though the underlying intine layer is swellable and responsive. That is why researchers believed that pollen is indestructible before our current work. The native pollen grains cannot be used as ingredients for macroscopic applications (e.g., tissue scaffolds, membrane, etc.). Our work proved that, by softening the exine layer, pollen microgels become processible and responsive soft matter materials.
4. In the same manner, to initiate the germination process of pollen grains in nature, the exine layer should allow the intine layer to be swollen by reducing its stiffness. Fig. 3c shows how the exine facilitates the expansion of pollen.

Comment 6. Line 134: What does “constraining the potential for intine swelling” mean? Does it reduce it or does it prevent it? Please rephrase for clarity.

Reply: The exine layer should reduce the potential for intine swelling, and in turn, the equilibrium swelling diameter of the intine in the presence of exine should be smaller than that of intine in the case of free swelling. We have rephrased this description in the revised manuscript as follows:

Pg. 4

*“A stiffer exine ($M_{E/I} > 2$) would be expected to impose a rigid boundary condition upon the hydrated intine, thereby **reducing** the potential for intine swelling (**Fig. 3a**).”*

Comment 7. Supplementary Figure 13a: 1- It is not clear what the inward blue arrows on the rightmost schematic indicate. 2- Where are swelling and inflation ratios properly defined? I had made this comment in previous versions. The use of these terms in this figure legend is vague. Additionally, how do these ratios defined by lambda correlate with lambda parameters of the Neo-Hookean energy potential (if they do)?

Reply: We thank the Reviewer for these suggestions and have clarified these points as follows:

1. We agree that the description of inward blue arrows should be improved. The blue arrows indicate the mechanical constraint applied to the intine layer due to the exine layer, reducing the swelling diameter of the intine layer compared to that of the intine for free swelling. Since the existence of blue arrows are potentially confusing, we have removed the arrows from Supplementary Figure 13a.
2. The swelling ratios of the pollen intine were defined as $\lambda_i = u_r/r$, where u_r is the deformation in the radial direction. To describe the inflation of the exine in accordance with intine swelling, we defined the inflation ratio of the exine in the same manner ($\lambda_e = u_r/r$).
3. The swelling ratio, λ_i or λ_e , in our current model is different from the deformation ratio (λ) of the Neo-Hookean energy potential in solid mechanics. The deformation ratio (λ) of the Neo-Hookean energy potential in solid mechanics is a “**deformation tensor**” in all directions for the Neo-Hookean model.

We have indicated this information in the revised manuscript (Supplementary Methods) as follows:

Supplementary Methods, Pg. 2

“... $\lambda_\theta = u_r/r$ is the deformation gradient in the circumferential direction, where u_r is the displacement of spherical pollen intine in radial direction, and, in turn, is equivalent to the swelling ratio (λ) in the present study...”

Comment 8. Supplementary Figure 13b: The authors mention internal and external solutions for the intine. Considering that the intine is ruptured upon the treatment, I wonder how the internal and external solutions are distinguished. Also, the drawing shows more than half of the grain and hence r_o and r_i are indicated too big it seems.

Reply: Thank you for the careful comments regarding **Supplementary Figure 13b**. In our multiphysics model, we did not distinguish the internal and external solutions because the intine shell is an open system with three apertures as the Reviewer pointed out. Thus, the internal solution is identical to the external solution. We classified the two solutions based on their positions. We have clarified this information in the revised manuscript (Supplementary Methods) as follows:

Supplementary Methods, Pg. 2

“Therefore, the present computational domain consists of (i) the pollen intine layer represented by a shell-like structure, (ii) the surrounding buffer medium, including both internal and external solutions, and (iii) the interface over the pollen intine and solution domains. Since the pollen grain is an open system through three apertures, we assumed that all the conditions of the internal solution were identical to those of the external solution.”

Comment 9. Supplementary Figure 13c: Avoiding excessive mesh distortion is mentioned but it is unclear what control was set in place to enforce it prior to running the simulation. Was it the mesh size, aspect ratio or angles? Also, it is unclear how the geometries presented in (c) and (d) correlate.

Reply: Thank you for your kind comments. Since a large and nonlinear deformation is induced at the tips of an aperture, we carefully introduced much finer meshes in those regions. **Supplementary Fig. 13c** shows the 1) mesh profile, and 2) correlation between the spherical and Cartesian coordinates of the pollen system. **Supplementary Fig. 13d** is the one-third symmetry model of the exine in the view of the xy plane. We have modified the information in the revised manuscript (Supplementary Figures) as follows:

Supplementary Figures, Pg. 20

“Supplement Figure 13 c, *Geometrical configuration of pollen microgel particle shells used in the simulations, including exine layer and spikes. Much finer meshes were placed on two tips of the aperture in order to avoid excessive mesh distortion during the aperture opening process.*”

Comment 10. Supplementary Figures 13d,e, 29: What does the heatmap show - stress, strain? Please provide LUT.

Reply: Thank you for the detailed comment. We decided that Supplementary Fig. 13d and e do not need to show any stress/strain contours because they are designed to explain 1) the geometry of our FEA model and 2) the way to measure the swelling diameter of a deformed pollen grain. We have updated **Supplementary Figures 13d and e** in the revised manuscript as follows:

Comment 11. Figure 3c, Supp Figure 14 and likely elsewhere: As mentioned for the previous versions, the use of the term “hydrostatic tensile stress” is ambiguous. I am aware that the authors mean to refer to the stress arising in the shell due to internal hydrostatic pressure. However, I find the use of hydrostatic for stress generated in the solid shell odd. Other measures of stress such as max. principal stress may be chosen for the label.

Reply: Thank you for the suggestion. As the Reviewer pointed out, it is not common to show the hydrostatic pressure as stress contours. The reason why we chose to depict the hydrostatic pressure that way is because we wanted to show how the pressure exerted from the intine was distributed in the exine layer. Nevertheless, we agree that the ‘Max. principal stress’ is more commonly used in describing the stress distribution in structural analysis. Thus, we have changed the current pressure contours into max. principal stress contours. Accordingly, we have replaced all ‘hydrostatic tensile stress’ to ‘Max. principal stress’ in **Fig. 3c** and **Supplementary Fig. 14**.

Comment 12. Line 684: “which WERE retrieved”?

Reply: Thank you for the comment. The Young’s modulus E was retrieved from indentation tests. We have clarified this information in the revised manuscript (Supplementary Methods) as follows:

Supplementary Methods, Pg. 5

“...where K_b is the bulk modulus, and G and K_b are related to the Young's modulus E by the formulas, $G = E/2(1 + \nu)$ and $K_b = E/3(1 - 2\nu)$ with a Poisson's ratio, ν . The Young's modulus was obtained from AFM force-based nanoindentation tests performed on the pollen particles (Supplementary Figs. 11-12).”

Comment 13. Line 723: How does a 30-microsecond step help to reduce the kinetic effects? Is it not so that a shorter step time increases the inertial effects, compared to a, say 1-second step? Here the authors could actually compare this with the real time scale of the swelling observed in the experiments. Further, the authors should provide a comment mentioning the parameter that helped to determine that this time step value was appropriate.

Reply: We appreciate the Reviewer’s comments about the FE modeling. Indeed, a 30-microsecond step is enough to ensure a quasi-static solution in ABAQUS/Explicit as explained below:

1. Because the kinetic effects are determined by the wave speed of a material, the general recommendation for quasi-static analysis in ABAQUS/Explicit is to limit the impact velocity v to less than 1% of the wave speed of the material, according to the ABAQUS documentation⁶. In our FEA model, we estimated the wave speed of the pollen exine

material to be $w_{exine} = \sqrt{E_e/\rho} \approx 600m/s$ according to its Young's moduli, E_e , of 400MPa and density of $1g/cm^3$. At the same time, we calculated the impact velocity, $v = D/t = 0.8 \times 28\mu m/30\mu s \approx 0.7 m/s$, according to i) the step time $t=30 \mu s$ and ii) global deflection displacement, $D = r_0(\lambda_{max} - 1)$, where the original pollen diameter (r_0) is $28 \mu m$ and the maximum swelling ratio (λ_{max}) of pollen is about 1.8. Therefore, the quasi-static requirement is fully satisfied since $v/w_{exine} \ll 1\%$. It is noteworthy that the scale of deformation is on the order of micrometers so the required time steps for quasi-static analysis in ABAQUS/ Explicit should be on the order of micrometers as well.

2. In fact, the time steps in the FEA simulation represent a pseudo time rather than real physical time of the swelling event. Although, in Abaqus/Explicit, a longer step time reduces the kinematic energy, making quasi-static solutions closer to static, it requires considerable computing resources. Hence, we carefully chose the step time of our simulations in times on the order of microseconds. As shown in **ROM Fig. 2**, with this microscale time step, the kinematic effects were almost negligible. Thus, we confirmed that we firmly obtained reasonable quasi-static solutions using Abaqus/Explicit under feasible model set-up conditions.

ROM Fig. 2. Kinetic and strain energy density evolution of pollen microgel swelling as a function of swelling time using Abaqus/Explicit.

References

- 6 ABAQUS. *ABAQUS 2017 User documentation*. (Dassault Systemes Simulia Corp., 2017).

Reviewer #5 Comments

In this manuscript, the authors report an approach to transform ultratough pollen grains into pliable, soft microgel by remodeling pollen shells and markedly alter the Young's modulus. The effect of internal pressure on the exine layer was also verified through the deformed shapes by using a continuum mechanics model. The paper is publishable but the authors may need to consider the following comments.

Reply: We are pleased to know that the Reviewer recommends publication of our work in *Nature Communications*. We sincerely thank the Reviewer for taking the time to carefully review our revised manuscript and for providing helpful feedback to improve the manuscript. Below, we provide our point-by-point responses to Reviewer's comments.

Comment 1. Ln. 656, "The driving pressure is then achieved by inputting the equilibrium ionic concentrations into..." If only equilibrium states are to be considered, no field theory (Eq. 2-7) is needed, and the authors are suggested to use simpler descriptions, e.g. the Donnan model, directly.

Reply: We sincerely appreciate the Reviewer's great expertise in the multiphysics modeling. Indeed, the Donnan model can predict a simple concentration relation of ionic species located at both sides of a membrane, but it is no way for the model to include the effect of i) the abundant fixed-charge groups (e.g. carboxylic or hydroxyl groups) of pectin in the intine layer and ii) other coupled fields (e.g. electric field) besides chemical potential, in order to describe the complex swelling mechanism of intine. As observed experimentally, those fixed-charge groups play a significant role for the responsive pollen swelling, implying that they are an important variable associated with variation of both pollen deformation and environmental changes (e.g., pH or ion concentration). Therefore, the Donnan model should be overly simplified for the pollen intine system due to the present complicated multiphysical mechanism, even if the equilibrium states are only considered for modeling.

Comment 2. Ln. 697, "Considering the presence of pores in the exine layer, a value of 0.4 for the Poisson's ratio was selected to account for exine compressibility." It seems that the moduli, especially the bulk modulus is highly affected by the value of Poisson's ratio. How did the authors obtain this particular number? Is it chosen to fit some data? How would it change the final results?

Reply: Thank you for this valuable comment. We also agree that the value of Poisson's ratio affects the bulk moduli of the exine.

1. Unfortunately, due to the complicated geometry of the exine layer (thin membrane with pores and spikes), we couldn't directly measure its Poisson's ratio. Since the deformation of pollen shells is reversible through multiple stimuli experiments, we are confident that the pollen exine is elastic, accommodating large deformation. Thus, we assume that the pollen exine is hyper-elastic.

2. For hyperelastic materials, compressibility or bulk modulus can be expressed in terms of Poisson's ratio (ν). In case of $\nu = 0.5$, a hyperelastic material is incompressible. In many studies on modeling of hyperelastic systems, the value of Poisson's ratio is assumed to be constant ($\nu < 0.5$) and is known to be continuously decreased from 0.5 when a system transforms from an incompressible material to a more compressible material.
3. Based on the SEM morphology of the exine layer (**Supplementary Figure 12**), we confirmed that it is porous. Thus, we assumed that the Poisson's ratio of the pollen exine is 0.4, which is smaller than the value of 0.5 for an incompressible hyperelastic material, considering that the exine is porous, thus compressible.
4. In the meantime, we did parametric study by varying the Poisson's ratio of intines from 0.499 to 0.3, and confirmed that there wasn't any significant difference for the final deformation of the pollen microgels regardless of various Poisson's ratios. Since the porosity of the exine layer is also dependent on the KOH treatment time, we chose the average value of the Poisson's ratios from our parametric study, $\nu = 0.4$. Considering the limited knowledge and complicated variation of porosity in the exine layer, we believe that this assumption, $\nu = 0.4$, should be reasonable, allowing us to capture the critical deformation behavior of the exine layer.

Comment 3. Ln. 725, "Quasi-static solutions were calculated using ABAQUS/Explicit..."
Abaqus Explicit is known for dynamic simulations. How did the authors obtain the quasi-static solutions?

Reply: Please refer to our answer to **Comment 13 of the Reviewer #3**.

Comment 4. It seems that the effect of the intine inflation is treated also as quasi-static by assuming homogeneous swelling throughout the intine core. How is this assumption justified? Would the existence and opening of the apertures change the local swelling and consequently the deformation of the exine layer?

Reply: We sincerely thank the Reviewer for this valuable comment. We agree that, due to the existence of the three apertures in the pollen microgel system, both the intine and exine layers undergo non-linear and inhomogeneous deformation. We clarified our description of this issue as follows:

1. Since we described the intine swelling using the multiphysics model, we did not consider the geometric effect of intine swelling. That is why we introduced the FEA modeling in order to capture the geometric effect on pollen inflation in an integrated fashion.
2. The existence of the apertures in the intine layer was implemented in our multiphysics model by setting the same conditions for both internal and external solutions. Due to the apertures, the pollen grain is an open system. Thus, we did not need to consider internal turgor pressure induced by osmotic pressure associated with a chemical potential difference between the internal and external solutions in the closed intine system.
3. Regardless of the existence of apertures, the water absorption volume of the intine layer should remain the same because the osmotic pressure associated with intine swelling should be independent of the geometry.

4. Moreover, due to the apertures, deformation is highly concentrated around the tips of the apertures, showing the elevated values of both max. principal stress and strain in **Fig. 3c** of the revised manuscript. This deformation mode is intrinsically induced by geometry of the pollen shell (particularly, due to the apertures). If inhomogeneous swelling of the intine occurred, then the expansion or inflation of the exine is expected to be either promoted or suppressed, implying that the max. values of stress/strain for deformed exine should be changed. At the same time, the stress/strain distribution across the exine layer should be almost identical because it is solely influenced by the geometry of the pollen system.